# INDUCTIVE RELATION PREDICTION USING ANALOGY SUBGRAPH EMBEDDINGS

**Jiarui Jin**[1][†]**, Yangkun Wang**[1][†]**, Kounianhua Du**[1][†]**, Weinan Zhang**[1][‡]**, Quan Gan**[2][‡]**, Zheng Zhang**[2]**, Yong Yu**[1]**, David Wipf**[2]
[1]Shanghai Jiao Tong University, [2]Amazon
{jinjiarui97,espylacopa,774581965,wnzhang,yyu}@sjtu.edu.cn,
{quagan,zhaz,daviwipf}@amazon.com

## ABSTRACT

Prevailing methods for relation prediction in heterogeneous graphs including knowledge graphs aim at learning the latent representations (i.e., embeddings) of observed nodes and relations, and are thus limited to the transductive setting where the relation types must be known during training. In this paper, we propose **AN**alogy Sub**G**raph **E**mbedding **L**earning (GraphANGEL), a novel relation prediction framework that predicts relations between each node pair by checking whether the subgraphs containing the pair are similar to other subgraphs containing the considered relation. Each graph pattern explicitly represents a specific logical rule, which contributes to an inductive bias that facilitates generalization to unseen relation types and leads to more explainable predictive models. Our model consistently outperforms existing models in terms of heterogeneous graph based recommendation as well as knowledge graph completion. We also empirically demonstrate the capability of our model in generalizing to new relation types while producing explainable heat maps of attention scores across the discovered logics.

## 1 INTRODUCTION

Relation modeling aims to learn the relations between nodes, leading to advances in a wide range of applications, e.g., recommender systems (Koren et al., 2009), knowledge graphs (Bordes et al., 2013), and biology (Yasunaga et al., 2021). As most relational data in the real world is heterogeneous, a principal way is to organize it into a heterogeneous graph. The dominant paradigms for relation prediction can be categorized into matrix factorization techniques (Nickel et al., 2011; 2012), statistical relational learning approaches (Richardson & Domingos, 2006; Singla & Domingos, 2005), and neural-embedding-based methods (Bordes et al., 2013; Dettmers et al., 2018). Among these, neural-embedding-based methods which learn to encode relational information using low-dimensional representations of nodes and relations, have shown good scalability (Bordes et al., 2013) and inductive learning capability (Battaglia et al., 2018) in terms of validating unseen nodes.

Results of such methods show that graph neural networks (GNNs) are able to condense the neighborhood connectivity pattern of each node into a node-specific low-dimensional embedding and successfully exploit such local connectivity patterns and homophily. Recent advances further reveal the logical expressiveness of GNNs (Barceló et al., 2019) and support their inductive ability to generalize to unseen nodes (Teru et al., 2020; Zhang & Chen, 2019).

In contrast, a limited study has been conducted (Yang et al., 2014) on the inductive learning capability for unseen *relation* types. Such inductive ability, if successfully exploited, can directly improve logical expressiveness of GNNs and enable them to effectively capture the underlying logical semantics (e.g., logical rules). Note that this is more challenging than unseen nodes, since it is usually hard to define the "neighborhood" of a relation type topologically.

In this paper, we propose **AN**alogy Sub**G**raph **E**mbedding **L**earning (GraphANGEL), a new relation prediction paradigm that holds a strong inductive bias to generalize to unseen relation types. Given a pair of nodes to predict the existence of a specific relation between them, the core idea is to extract

---

[†]Work done during internship at Amazon Web Services Shanghai AI Lab.
[‡]Weinan Zhang and Quan Gan are the corresponding authors.

some analogy subgraphs containing the pair, and compare them against other subgraphs sharing similar shapes. We call these shapes *graph patterns*.

Taking Figure 1 as an example, the task is to predict whether `Person E lives` in `London`. We construct three patterns involving the relation `live`. The first is *target pattern* containing the source and target nodes, i.e., `Person E` and `London`, for which we are to predict the existence of the relation `live`. The second is *supporting pattern* that includes `live` as evidence *supporting* the existence. The third is *refuting pattern* that does not include `live` as a baseline for comparison. If an edge does exist between the two nodes, the subgraphs matching the target pattern should be more similar to those matching supporting patterns than the refuting patterns. Following the above intuition, we find a set of subgraphs that match each of the patterns. Then we compare the first set against the set matching supporting patterns, and also against the set matching refuting patterns, using a neural network. As shown in the bottom part of the figure, the subgraphs in the second set share the higher similarity with the one in the first set, so the prediction result in this case is that there exists a `live`.

Given a triplet $\langle s, r, t \rangle$, GraphANGEL consists of the following stages: (1) determining target patterns from $s, t$ as well as supporting and refuting patterns from $r$, (2) retrieving subgraphs matching each pattern, (3) computing the representations of each set and then the similarity between the subgraph set matching target pattern and the set matching supporting/refuting patterns, and the final prediction based on the similarities. For the first stage, our architecture design only involves the graph patterns in pair, 3-cycle, and 4-cycle shapes, which is efficient to match and already shows good results. For the second stage, we introduce efficient searching and sampling techniques to find the subgraphs matching the patterns. We use a GNN with attention in the third stage to combine the sets of subgraphs as well as their node features. The

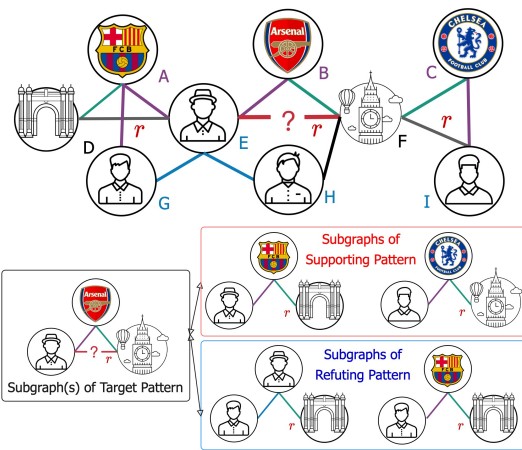

Figure 1: An illustrative example of motivation.

attention module can simultaneously produce an explainable heat map across the discovered patterns. Notably, none of the above stages requires explicitly learning a representation of $r$, the relation we are predicting, thus, GraphANGEL naturally generalizes to modeling unseen relation types.

We benchmark GraphANGEL on heterogeneous graph based recommendation and knowledge graph completion tasks with the state-of-the-art methods. For the evaluation of inductive capabilities, we construct several new inductive benchmarks by either removing or adding relations from knowledge graph datasets. Extensive experimental comparisons on these benchmarks exhibit the superiority of our method under both transductive and inductive settings.

## 2 BRIDGING LOGICAL EXPRESSIONS AND GRAPH PATTERNS

We begin with bridging logical expressions and graph patterns based on two intuitions/assumptions:

- A relation can often be inferred from other relations with a combination of simple logical rules that do not involve too many nodes.
- One can predict relation existence by finding whether the subgraphs containing the pair are similar to the subgraphs containing an edge with the same relation.

Take Figure 1 as an example where we wish to predict whether `Person E lives` in `London`. An example of the first intuition goes as follows: if `Person E watches` the soccer matches with club `Arsenal`, which is `based` in `London`, then `Person E` may also `live` in `London`. However, such logical rules (so called chain-like logic rules in (Yang et al., 2017)) may not always hold, requiring probabilistic inference. For instance, `Person G watches` the matches with `FC Barcelona`, which is `Based` in `Barcelona`, but he does not `live` in the same city. To make more accurate predictions, we need to combine other possible logical rules, such as "Friends often live in the same city" - in this case, `Person E` and `Person H` are `friends`, and `Person H lives`

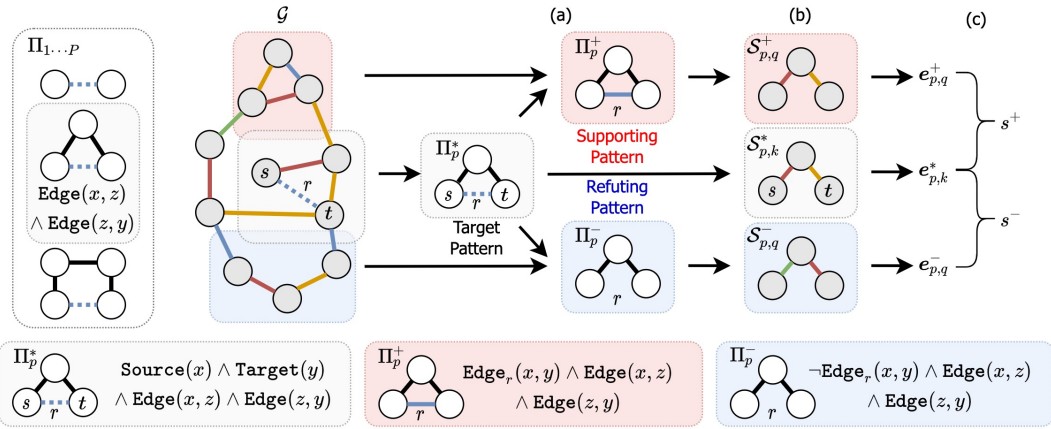

Figure 2: Illustration of GraphANGEL's relation prediction workflow, where different edge colors in the graph $\mathcal{G}$ represent different relation types, and dashed edges in $\mathcal{G}$ represent the triplet $\langle s, r, t \rangle$ we wish to predict. The left box shows the patterns considered in our implementation, where black edges mean matching edges irrespective of relation types. The bottom boxes show the logical function of the three patterns.

in `London` (some of them construct certain patterns are called tree-like logic rules (Yang & Song, 2019) or graph-like rules (Shi et al., 2020)).

The second intuition tells us that we can predict whether `Person E lives in London` by checking if the subgraphs containing the pair (`Person E`, `London`) (named *target* subgraphs, e.g. `Person E - Arsenal - London`) are similar to the subgraphs containing a `live` relation (named *supporting* subgraphs, e.g. `Person I - Chelsea - London`). Similarity computation is done by a neural network. To determine whether the two sets of subgraphs are similar, we additionally compare the target subgraphs against another set of subgraphs that do not contain the relation (named *refuting* subgraphs, e.g. `Person E - Person G - Barcelona`) as a baseline. A variety of options exist for selecting the refuting subgraphs, ranging from selecting those having the same topology regardless of the actual relation types, to those having both the same shape and relation types.

In addition, we do not include the edge of `live` in the supporting subgraphs to avoid information leakage. We also require the supporting subgraphs and refuting subgraphs to have the same shape as that of the target subgraphs in order to make similarity computation focus more on the node features and relation types rather than the topology, and that is why we call both supporting subgraphs and refuting subgraphs *analogy* subgraphs. This enables us to predict `live` relation with the pipeline above *without* learning an explicit embedding for `live` relation, unlike prior works. The reason is that the information of `live` relation can be implicitly expressed by the other relations found in the supporting and refuting subgraphs, e.g. `live` can be expressed by `watch` and `based`. This serves as the basis for our model's generalizability to relations that have no occurrence in the training set.

We can describe what kind of subgraphs we are looking for in the above example with logical expressions. The target subgraphs above will have the logical expression $\text{Source}(x) \wedge \text{Target}(y) \wedge \text{Edge}(x, z) \wedge \text{Edge}(z, y)$, where $\text{Source}(x)$ means if node $x$ is the source node $s$, $\text{Target}(y)$ means if node $y$ is the target node $t$, and $\text{Edge}(x, z)$ means if there exists an edge regardless of relation type between $x$ and $z$. Our task is to determine whether an edge of relation type $r$ exists between $s$ and $t$. The supporting subgraphs will have $\text{Edge}(x, z) \wedge \text{Edge}(z, y) \wedge \text{Edge}_r(x, y)$, meaning that the subgraph can be any 3-cycle except that there must be an edge with relation type $r$. The refuting subgraphs will have $\text{Edge}(x, z) \wedge \text{Edge}(z, y) \wedge \neg\text{Edge}_r(x, y)$, meaning that it can be any 2-path except that the starting node and ending node must not have an edge with relation $r$ in between. This leads to the concept of *graph patterns*, defined as follows:

**Definition 1** (Graph Pattern). *A graph pattern is a logical function that takes in a subgraph as input and returns a boolean value, consisting of logical operators ($\neg, \wedge, \vee$) as well as indicator operators determined by the existence of nodes and edges, the latter includes:*

- $\text{Source}(x)$ *and* $\text{Target}(x)$, *returning* `true` *iff $x$ is the source node and target node, respectively.*
- $\text{Edge}(x, y)$, *returning* `true` *iff there exists an edge between node $x$ and $y$.*
- $\text{Edge}_r(x, y)$, *returning* `true` *iff there exists an edge of relation $r$ between node $x$ and $y$.*

We call a subgraph $\mathcal{S}$ *matches* a graph pattern $\Pi$ if `true` is returned when $\mathcal{S}$ is applied to $\Pi$. Such operation is already well supported in graph databases (Francis et al., 2018), and in the following sections we give more efficient algorithms that perform matching with simple patterns.

Table 1: A summary of how to construct target, supporting, and refuting patterns.

| Base Pattern $\Pi_p$ | Target Pattern $\Pi_p^*$ | Supporting Pattern $\Pi_p^+$ | Refuting Pattern $\Pi_p^-$ |
|---|---|---|---|
| $\Pi_p$ | $\texttt{Source}(x) \wedge \texttt{Target}(y) \wedge \Pi_p$ | $\texttt{Edge}_r(x,y) \wedge \Pi_p$ | $\neg\texttt{Edge}_r(x,y) \wedge \Pi_p$ |

## 3 Modeling Relation with Analogy Subgraph Embeddings

For each graph $\mathcal{G} = (\mathcal{V}, \mathcal{E}, \mathcal{R})$ where $\mathcal{V}$ denotes node set, $\mathcal{E}$ denotes edge set and $\mathcal{R}$ is relation set, as Figure 2 shows, we outline how GraphANGEL works for each triplet $\langle s, r, t \rangle$ to predict the existence for the edge of type $r$ connecting source node $s$ and target node $t$ as follows. (a) We start by determining $P$ *target patterns* denoted as $\Pi_1^*, \cdots, \Pi_P^*$. For each $\Pi_p^*$ we also determine its corresponding *supporting pattern* $\Pi_p^+$ and *refuting pattern* $\Pi_p^-$. (b) We then sample a set of $K$ target subgraphs $\{\mathcal{S}_{p,k}^*\}_{k=1}^K$ matching $\Pi_p^*$, $Q$ supporting subgraphs $\{\mathcal{S}_{p,q}^+\}_{q=1}^Q$ matching $\Pi_p^+$, and $Q$ refuting subgraphs $\{\mathcal{S}_{p,q}^-\}_{q=1}^Q$ matching $\Pi_p^-$. (c) For each pattern $\Pi_p$, we next compute the representation of each sampled subgraph and obtain the set of target subgraph embeddings $\{e_{p,k}^*\}_{k=1}^K$, supporting subgraph embeddings $\{e_{p,q}^+\}_{q=1}^Q$ and refuting subgraph embeddings $\{e_{p,q}^-\}_{q=1}^Q$. (d) We finally compute a similarity score between the set $\{e_{p,k}^*\}_{p=1,k=1}^{P,K}$ and the set $\{e_{p,q}^+\}_{p=1,q=1}^{P,Q}$, as well as $\{e_{p,k}^*\}_{p=1,k=1}^{P,K}$ and $\{e_{p,q}^-\}_{p=1,q=1}^{P,Q}$, to get the relation prediction result.

We summarize the notations in Appendix A1, show the training algorithm in Algorithm 1, describe the details in the following subsections, and provide the overall time complexity in Appendix A3.

**Pattern Construction.** During training, we set for each $\Pi_p$ the target pattern $\Pi_p^*$, the supporting pattern $\Pi_p^+$, and the refuting pattern $\Pi_p^-$ using Table 1. Since pattern matching is NP-complete like subgraph matching (Lewis, 1983), we first designate before training a set of patterns $\{\Pi_p\}_{p=1}^P$ whose pattern matching can be computed in manageable time. In practical, we use *Pairs*, *3-cycles*, and *4-cycles*.

**Subgraph Retrieval.** Although general graph pattern matching is supported in graph databases (Francis et al., 2018), more efficient solutions exist for simpler graph patterns, especially when we only consider pairs, 3-cycle and 4-cycle shapes. Searching and retrieving the subgraphs matching patterns in Pair shape is trivial since it reduces to finding edges with a given relation, so the following discussion only involves 3-cycle and 4-cycle patterns. We pre-compute all subgraphs matching 3-cycle patterns and uniformly sample a number of subgraphs matching 4-cycle patterns, and store them into a buffer by our algorithms, summarized as follows, with pseudocode, correctness proofs and complexity analysis in Appendix A2.

---

**Algorithm 1:** GraphANGEL

**Input:** Graph $\mathcal{G}$, Patterns $\Pi_1, \ldots, \Pi_P$.

**for** *each tuple* $\langle s, r, t \rangle$ **do**
    **for** *each pattern* $\Pi_p$ **do**
        Construct $\Pi_p^*, \Pi_p^+, \Pi_p^-$.
        Retrieve $K$ subgraphs $\{\mathcal{S}_{p,k}^*\}_{k=1}^K$ matching $\Pi_p^*$.
        Retrieve $Q$ subgraphs $\{\mathcal{S}_{p,q}^+\}_{q=1}^Q$ matching $\Pi_p^+$.
        Retrieve $Q$ subgraphs $\{\mathcal{S}_{p,q}^-\}_{q=1}^Q$ matching $\Pi_p^-$.
        Compute $e_{p,k}^*, e_{p,q}^+, e_{p,q}^-$ via Eq. (1).
    **end**
    Compute $s^+, s^-$ via Eq. (2).
    Update parameters according to Eq. (3).
**end**

---

Then, we perform pattern matching for 3-cycles and 4-cycles for each subgraph as follows.

- *3-cycles.* We first partition the node set by the node degrees into two sets $\mathcal{V} = \mathcal{V}_1 \cup \mathcal{V}_2$. $\mathcal{V}_1$ contains the nodes whose degrees are less than $|\mathcal{E}|^{\frac{1}{2}}$, and $\mathcal{V}_2$ contains the rest. For each $u, v, w \in \mathcal{V}_1$, we check if they form a 3-cycle. Then, for each $u \in \mathcal{V}_2$, we enumerate all pairs of its neighbors and see if they are connected. The complexity to find all the subgraphs is $O(|\mathcal{E}|^{\frac{3}{2}})$.
- *4-cycles.* For each node, we find all 2-paths $(u, v, w)$ whose starting node $u$ has the largest degree, i.e., $d_u = \max(d_u, d_v, d_w)$, and then store them in $\mathcal{T}_{uw}$. Each pair of 2-paths that shares the same ending node forms a 4-cycle. In this regard, 4-cycles can be uniformly sampled by first sampling $u, w$ with the probability in proportion to $|\mathcal{T}_{uw}| \cdot |\mathcal{T}_{uw} - 1|$, and then choosing two distinct nodes $v, x$ from $\mathcal{T}_{uw}$. The complexity to sample $n_{\texttt{4-cycle}}$ subgraphs is $O(\max(n_{\texttt{4-cycle}}, |\mathcal{E}|^{\frac{3}{2}}))$.

Then, we perform pattern matching for 3-cycles and 4-cycles for each subgraph as follows.

- *Target patterns.* Matching 3-cycle target patterns reduces to finding common neighbors of $s$ and $t$, which takes $O(d_s + d_t)$ time where $d_s$ and $d_t$ are degrees of $s$ and $t$. Matching 4-cycle target pattern reduces to finding whether the neighbors of $s$ and $t$ are connected, taking $O(d_s d_t)$ time.

Table 2: Patterns $\Pi_p$ considered in our experiments.

| Task | Pair | 3-cycle (with type) | 4-cycle (with type) |
|---|---|---|---|
| Knowledge Graph Completion | `true` | $\texttt{Edge}(x,z) \wedge \texttt{Edge}(z,y)$ | $\texttt{Edge}(x,z) \wedge \texttt{Edge}(z,w) \wedge \texttt{Edge}(w,y)$ |
| Heterogeneous Graph Recommendation | `true` | $\texttt{Edge}_a(x,z) \wedge \texttt{Edge}_b(z,y)$ | $\texttt{Edge}_a(x,z) \wedge \texttt{Edge}_b(z,w) \wedge \texttt{Edge}_c(w,y)$ |

- *Supporting patterns*. Matching supporting patterns reduces to finding a subgraph containing an edge of type $r$ in the precomputed result. One can efficiently retrieve with a precomputed inverted map with relation $r$ as key and the actual subgraphs containing it as value.
- *Refuting patterns*. Matching refuting patterns reduces to random walks, followed by checking whether the starting node and the ending node has an edge of type $r$.

We further design a series of novel uniform sampling algorithms such that the time complexity of sampling $n$ refuting cases of $\Pi_{\texttt{3-cycle}}$ or $\Pi_{\texttt{4-cycle}}$ reduces to $O(|\mathcal{V}| + |\mathcal{E}| + n)$. See detailed descriptions in Appendix A2.

**Representation Computation.** We apply a neural network $\Phi(\cdot)$ over each subgraph $\mathcal{S}_{p,k}^*, \mathcal{S}_{p,q}^+$ and $\mathcal{S}_{p,q}^-$ to obtain graph-level representations $\boldsymbol{e}_{p,k}^*, \boldsymbol{e}_{p,q}^+, \boldsymbol{e}_{p,q}^-$, following

$$\boldsymbol{e}_{p,k}^* = \Phi\left(\mathcal{S}_{p,k}^*\right), \; \boldsymbol{e}_{p,q}^+ = \Phi\left(\mathcal{S}_{p,q}^+\right), \; \boldsymbol{e}_{p,q}^- = \Phi\left(\mathcal{S}_{p,q}^-\right). \tag{1}$$

In the implementation, we adopt single layer R-GCN (Schlichtkrull et al., 2018) followed by any readout function, e.g., $\texttt{Mean}(\cdot)$, $\texttt{Max}(\cdot)$ as $\Phi(\cdot)$. We also empirically study the effect of using other GNNs with different number of propagation layers as $\Phi(\cdot)$ in Appendix A6.5.

**Similarity Computation.** We deploy a neural network $\Psi(\cdot)$ to measure the similarity $s^+$ between the set of subgraphs matching $\Pi_1^*, \ldots, \Pi_P^*$ and the set of subgraphs matching $\Pi_1^+, \ldots, \Pi_P^+$; and the similarity $s^-$ between the set of subgraphs matching $\Pi_1^*, \ldots, \Pi_P^*$ and the set of subgraphs matching $\Pi_1^-, \ldots, \Pi_P^-$, which can be formulated as

$$\begin{aligned} s^+ &= \Psi\left(\{\boldsymbol{e}_{p,k}^*, p=1,\ldots,P; k=1,\ldots,K\}, \{\boldsymbol{e}_{p,q}^+ : p=1,\ldots,P; q=1,\ldots,Q\}\right), \\ s^- &= \Psi\left(\{\boldsymbol{e}_{p,k}^* : p=1,\ldots,P; k=1,\ldots,K\}, \{\boldsymbol{e}_{p,q}^- : p=1,\ldots,P; q=1,\ldots,Q\}\right). \end{aligned} \tag{2}$$

There are many choices of measuring the similarity between two sets. In the implementation, we adopt a co-attention mechanism (Lu et al., 2016) as $\Psi(\cdot)$, because even within the same pattern, the subgraphs having similar node features are more important than others.

We put the concrete formulation of $\Phi(\cdot)$ and $\Psi(\cdot)$ in the Appendix A3.3 and A3.4.

**Loss Function.** For each tuple $\langle s, r, t \rangle$, we have the binary label $y$ in the training dataset $\mathcal{D}$ to denote the relation existence. We here train the model by logistic loss with negative sampling:

$$L = - \sum_{\langle s,r,t \rangle \in \mathcal{D}} (y \log \hat{y} + (1-y) \log(1-\hat{y})), \tag{3}$$

where $\hat{y}$ is the final prediction calculated with normalized similarity as $\hat{y} = \frac{s^+}{s^+ + s^-}$.

**Inference.** We perform pattern matching to find the target, supporting, and refuting subgraphs on the testing graph, and compute the prediction score $\hat{y}$ exactly as what we do in training. We predict that an edge exists if $\hat{y}$ is larger than a threshold (as a hyper-parameter, $0.5$ in our implementation).

**Limitations.** There are two main limitations of GraphANGEL. The first one is that GraphANGEL would not work reliably if the assumptions (intuitions) in Section 2 are violated. Examples include when node $s$ and $t$ are disconnected or topologically far away when $\langle s, r, t \rangle$ is removed, so that one may not be able to find 3-cycle or 4-cycle target subgraphs. The prediction in this case can only rely on Pair patterns, i.e., comparing if the source-target pair is similar to the incident nodes of edges with relation $r$ or not. The second one is that current online subgraph sampling algorithms are slow. To make sampling efficient, the subgraph retrieval stage requires finding and storing all 3-cycles and 4-cycles (See Appendix A3.1 for the space complexity analysis). For static graphs, it is a one-time preprocessing step, although the results can be stored on external storage. We also provide the incremental searching and retrieving algorithms to address the dynamic graphs in the real-world scenario in Appendix A3.2. See further discussions on these limitations in Appendix A4.1.

Table 3: Result comparisons with baselines on heterogeneous graph recommendation task.

| Models | LastFM | | | Yelp | | | Amazon | | | Douban Book | | |
|---|---|---|---|---|---|---|---|---|---|---|---|---|
| | AUC | ACC | F1 | AUC | ACC | F1 | AUC | ACC | F1 | AUC | ACC | F1 |
| HetGNN | 0.7936 | 0.7258 | 0.7177 | 0.9083 | 0.8297 | 0.8205 | 0.7744 | 0.7108 | 0.7109 | 0.8737 | 0.7912 | 0.7915 |
| HAN | 0.8915 | 0.8337 | 0.8296 | 0.9156 | 0.8488 | 0.8426 | 0.8487 | 0.7682 | 0.7572 | 0.9244 | 0.8501 | 0.8458 |
| TAHIN | 0.8910 | 0.8463 | 0.8337 | 0.9067 | 0.8490 | 0.8393 | 0.8535 | 0.7718 | 0.7644 | 0.9253 | 0.8497 | 0.8373 |
| HGT | 0.8394 | 0.7939 | 0.7882 | 0.9006 | 0.8375 | 0.8334 | 0.7125 | 0.6482 | 0.6296 | 0.9132 | 0.8364 | 0.8222 |
| R-GCN | 0.8526 | 0.8393 | 0.8341 | 0.9098 | 0.8427 | 0.8323 | 0.8130 | 0.7408 | 0.7366 | 0.9203 | 0.8413 | 0.8271 |
| GraphANGEL$_{3-cycle}$ | 0.8934 | 0.8519 | 0.8465 | 0.9167 | 0.8498 | 0.8514 | 0.8601 | 0.7746 | 0.7746 | 0.9256 | 0.8512 | 0.8479 |
| GraphANGEL$_{4-cycle}$ | 0.8961 | 0.8514 | 0.8467 | 0.9201 | 0.8506 | 0.8521 | 0.8609 | 0.7752 | 0.7716 | 0.9242 | 0.8502 | 0.8378 |
| GraphANGEL | 0.8979 | 0.8524 | 0.8469 | 0.9231 | 0.8512 | 0.8533 | 0.8611 | 0.7790 | 0.7753 | 0.9311 | 0.8601 | 0.8543 |
| GraphANGEL* | **0.9001** | **0.8611** | **0.8589** | **0.9337** | **0.8701** | **0.8577** | **0.8700** | **0.7810** | **0.7813** | **0.9410** | **0.8640** | **0.8591** |

## 4 EXPERIMENT

### 4.1 EXPERIMENT SETUP AND COMPARED ALGORITHMS

**Recommendation on Heterogeneous Graph**. We evaluate our model on four heterogeneous graph benchmark datasets in various fields: LastFM (Hu et al., 2018a), Yelp (Hu et al., 2018b), Amazon (Ni et al., 2019), and Douban Book (Zheng et al., 2017). The baselines we compare against are: HetGNN (Zhang et al., 2019a), HAN (Wang et al., 2019b), TAHIN (Bi et al., 2020), HGT (Hu et al., 2020), and R-GCN (Schlichtkrull et al., 2018). As the recommendation task can naturally be regarded as the relation predictions between each user and item pair, for each triplet, we formulate the task as a binary classification task. We split each dataset into 60%, 20%, and 20% for training, validation and test sets, respectively. Following the setting of (Zhang et al., 2019a), we generate an equal number of negative triplets with the same relation type in the test set, and report Area Under ROC Curve (AUC), Accuracy (ACC), and F1 score. More details of the datasets and experimental configurations as well as the implementation details for baselines are reported in Appendix A5.

**Knowledge Graph Completion**. We compare different methods on two benchmark datasets: FB15k-237 (Toutanova & Chen, 2015) and WN18RR (Dettmers et al., 2018), which are constructed from Freebase (Bollacker et al., 2008) and WordNet (Miller, 1995), respectively. The baselines we compare against are: MLN (Singla & Domingos, 2005), TransE (Bordes et al., 2013), ConvE (Dettmers et al., 2018), ComplEx (Trouillon et al., 2016), pLogicNet (Qu & Tang, 2019), RotatE (Sun et al., 2019a), RNNLogic (Qu et al., 2020), ComplEx-N3 (Lacroix et al., 2018), GraIL (Teru et al., 2020) and QuatE (Zhang et al., 2019b). For each triplet, we mask the source or target node, and let each method predict the masked node. We use the filtered setting during evaluation on the standard training-validation-test split, randomly break ties for triplets with the same score (Sun et al., 2019b) and report Mean Rank (MR), Mean Reciprocal Rank (MRR), and Hit@K (K=1,3,10).

In each task, we implement GraphANGEL as we proposed in Section 3. Concretely, we imply GraphANGEL with $\Pi_p$ upon the logical patterns shown in Table 2. For further investigations on the influence of different graph patterns, we here introduce GraphANGEL$_{3-cycle}$, a variant of GraphANGEL without using patterns in 3-cycle shapes; and GraphANGEL$_{4-cycle}$, another variant without using patterns in 4-cycle shapes. For recommendation tasks on heterogeneous graphs, since the number of edge types is usually small, we can enumerate all the relation type combination in each pattern. This allows us to make the pattern $\Pi_p$ specific to relation types. We denote this variant as GraphANGEL*. As the main advantage of our model is that it can generalize to relation types unseen during training *without fine-tuning*, we both evaluate the overall performance of GraphANGEL in standard relation prediction tasks and its generalizability to unseen relation types against other state-of-the-art methods. We exam the robustness of GraphANGEL by adding the Gaussian noise into the heterogeneous graphs and report the results in Appendix A6.5. We also compare the training and inference time of GraphANGEL against baseline models and report results in Appendix A6.7.

### 4.2 RESULT ANALYSIS OF STANDARD TASKS

**Heterogeneous Graph Based Recommendation**. In recommendation scenarios, edges between user and item nodes are generally more likely to exist if they share neighboring users or items. In other words, users close in the graph may share similar interests and items close usually share similar attributes. Table 3 summarizes the performances of GraphANGEL and baselines on four different kinds of recommendation tasks. We observe that GraphANGEL significantly outperforms the baselines across all datasets in terms of AUC, ACC, and F1 metrics. Almost all prevailing baseline methods on heterogeneous graph are based on sampling through metapath. One explanation is that

Table 4: Result comparisons with baselines on knowledge graph completion task.

| Models | FB15k-237 | | | | | WN18RR | | | | |
|---|---|---|---|---|---|---|---|---|---|---|
| | MR | MRR | Hit@1 | Hit@3 | Hit@10 | MR | MRR | Hit@1 | Hit@3 | Hit@10 |
| pLogicNet | 173 | 0.332 | 0.237 | 0.367 | 0.524 | 3408 | 0.441 | 0.398 | 0.446 | 0.537 |
| TransE | 181 | 0.326 | 0.229 | 0.363 | 0.521 | 3410 | 0.223 | 0.235 | 0.401 | 0.531 |
| ConvE | 244 | 0.325 | 0.237 | 0.356 | 0.501 | 4187 | 0.430 | 0.400 | 0.440 | 0.520 |
| ComplEx | 339 | 0.247 | 0.158 | 0.275 | 0.428 | 5261 | 0.440 | 0.410 | 0.460 | 0.510 |
| MLN | 1980 | 0.098 | 0.067 | 0.103 | 0.160 | 11549 | 0.259 | 0.191 | 0.322 | 0.361 |
| RotatE | 177 | 0.338 | 0.241 | 0.375 | 0.533 | 3340 | 0.476 | 0.428 | 0.492 | 0.571 |
| RNNLogic | 232 | 0.344 | 0.252 | 0.380 | 0.530 | 4615 | 0.483 | 0.446 | 0.497 | 0.558 |
| ComplEx-N3 | 159 | 0.370 | 0.272 | 0.400 | 0.561 | 3452 | 0.491 | 0.440 | 0.500 | 0.581 |
| GraIL | 205 | 0.322 | 0.223 | 0.361 | 0.520 | 3539 | 0.401 | 0.352 | 0.438 | 0.501 |
| QuatE | **87** | 0.348 | 0.248 | 0.382 | 0.550 | **2314** | 0.488 | 0.438 | 0.508 | 0.582 |
| GraphANGEL$_{3-cycle}$ | 159 | 0.366 | 0.270 | 0.398 | 0.560 | 2919 | 0.492 | 0.463 | 0.497 | 0.590 |
| GraphANGEL$_{4-cycle}$ | 165 | 0.351 | 0.239 | 0.381 | 0.548 | 2914 | 0.493 | 0.465 | 0.502 | 0.587 |
| GraphANGEL | 151 | **0.374** | **0.275** | **0.408** | **0.564** | 2834 | **0.504** | **0.470** | **0.515** | **0.598** |
| | ±3 | ±0.003 | ±0.002 | ±0.004 | ±0.004 | ±25 | ±0.003 | ±0.002 | ±0.004 | ±0.004 |

given a specific pattern, these metapaths can be roughly regarded as target patterns, but without constructing supporting and refuting patterns in Table 1.

**Knowledge Graph Completion**. In knowledge graphs, the connection between two nodes is determined by both logic and node attributes. Table 4 summarizes all experimental results. As can be seen, GraphANGEL outperforms all baselines across all datasets. One explanation is that most knowledge graph embedding techniques focus on mining the hidden information in each tuple, which is similar to only considering the patterns in Pair shapes in Table 1. However, other patterns contain information involving multiple relations, which enables to model the logics.

In order to better illustrate our performance of these relations with few occurrences in the training set, we solely report the results of testing each model on the 20% relations with few occurrence in Table A4 (see Appendix A6.1 for details). From the comparison between Tables 4 and A4, we can observe that with few shots of relations, GraphANGEL can have a better generalization ability. One reason is that embeddings of the relations with few occurrences cannot be trained with plenty of data samples, resulting in low expressive power of the relations. In contrast, GraphANGEL does not learn the embeddings directly, but learns to represent the relations of the related logics. However, it is still more challenging to model these relations that lead to a drop in performance.

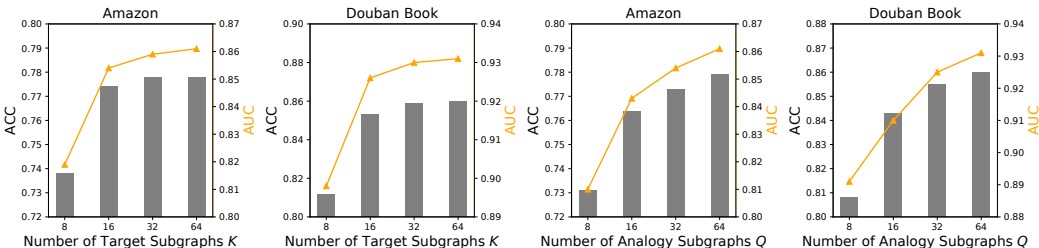

Figure 3: Performance change of GraphANGEL with different number of subgraphs in terms of ACC and AUC.

## 4.3 EFFECT OF DIFFERENT PATTERNS

We systematically investigate the effect of three shapes of patterns used in GraphANGEL. For each dataset, we evaluate the effect of the patterns in 3-cycle and 4-cycle shapes by the performance of GraphANGEL$_{3-cycle}$ and GraphANGEL$_{4-cycle}$. Since different shapes of patterns represent different composition logical rules, as shown in Table 1, patterns in Pair shape are the most general but include the least structure information, while those in 4-cycle shape are rich in the structure but less common. Hence, these patterns have their unique power in representing logics. Although it is hard to determine whether 3-cycle or 4-cycle shaped patterns is more powerful, as shown in Tables 3, 4 and A4, GraphANGEL with patterns in all shapes achieves the best performance. Besides the pattern type, we also investigate how the number of sampled subgraphs affects the performance. Taking Amazon and Douban Book datasets as examples, we show the performance of GraphANGEL under different $K$ and $Q$ in terms of ACC and AUC in Figure 3. One explanation is that the subgraphs following target patterns are constricted within the neighborhood of source and target nodes, the number of which is much smaller than subgraphs following supporting and refuting patterns.

Table 5: Result comparisons with baselines on generalization setting by randomly removing 20% relations. See Appendix A6.2 for full version and Appendix A6.2 for results of dropping 5%, 10%, 15%. The numbers in brackets show the descent degree.

| Models | FB15k-237 | | | WN18RR | | |
|---|---|---|---|---|---|---|
| | Hit@1 | Hit@3 | Hit@10 | Hit@1 | Hit@3 | Hit@10 |
| pLogicNet* | 0.112(52.7%↓) | 0.179(51.2%↓) | 0.257(51.0%↓) | 0.141(64.6%↓) | 0.222(50.2%↓) | 0.267(50.3%↓) |
| TransE* | 0.101(55.9%↓) | 0.163(55.1%↓) | 0.246(52.8%↓) | 0.072(46.7%↓) | 0.200(50.1%↓) | 0.260(51.0%↓) |
| ConvE* | 0.104(56.1%↓) | 0.178(50.0%↓) | 0.247(50.7%↓) | 0.201(49.8%↓) | 0.223(49.3%↓) | 0.268(48.5%↓) |
| ComplEx* | 0.078(50.6%↓) | 0.142(48.4%↓) | 0.226(47.2%↓) | 0.214(47.8%↓) | 0.236(48.7%↓) | 0.267(47.6%↓) |
| MLN* | 0.031(53.7%↓) | 0.049(52.4%↓) | 0.070(56.3%↓) | 0.092(51.8%↓) | 0.154(52.2%↓) | 0.178(50.7%↓) |
| RotatE* | 0.121(49.8%↓) | 0.187(50.1%↓) | 0.271(49.1%↓) | 0.238(44.3%↓) | 0.260(47.1%↓) | 0.296(48.2%↓) |
| RNNLogic* | 0.124(50.7%↓) | 0.172(54.7%↓) | 0.240(54.6%↓) | 0.244(45.2%↓) | 0.260(47.6%↓) | 0.281(49.7%↓) |
| ComplEx-N3* | 0.142(47.2%↓) | 0.208(49.6%↓) | 0.289(48.5%↓) | 0.250(43.2%↓) | 0.269(46.2%↓) | 0.311(46.4%↓) |
| GraIL* | 0.125(43.9%↓) | 0.185(48.8%↓) | 0.263(49.4%↓) | 0.195(44.7%↓) | 0.222(49.3%↓) | 0.267(46.8%↓) |
| QuatE* | 0.127(48.7%↓) | 0.190(50.3%↓) | 0.282(48.7%↓) | 0.248(43.3%↓) | 0.255(49.8%↓) | 0.308(47.0%↓) |
| GraphANGEL$_{3-cycle}$ | 0.168(37.6%↓) | 0.230(42.2%↓) | 0.333(40.5%↓) | 0.277(40.2%↓) | 0.291(**41.4%**↓) | 0.329(44.3%↓) |
| GraphANGEL$_{4-cycle}$ | 0.147(38.7%↓) | 0.222(41.7%↓) | 0.328(40.2%↓) | 0.278(40.2%↓) | 0.291(42.1%↓) | 0.326(44.4%↓) |
| GraphANGEL | **0.173**(37.2%↓) | **0.238**(**41.5%**↓) | **0.337**(**40.1%**↓) | **0.284**(**39.5%**↓) | **0.299**(41.8%↓) | **0.334**(**44.1%**↓) |

## 4.4 RESULT ANALYSIS OF GENERALIZATION STUDY

We further evaluate these models in a scenario where generalizing from existing relations to unseen relations is required. Concretely, we use the same datasets with knowledge graph completion and randomly split $\mathcal{R}$ into two partitions $\mathcal{R}_{\text{seen}}$ and $\mathcal{R}_{\text{unseen}}$. Each model is trained and validated only with the relations in $\mathcal{R}_{\text{seen}}$. During testing, the training and validation triplets in $\mathcal{R}_{\text{unseen}}$ are added back to the original graph. We report results on test triples with relations in $\mathcal{R}_{\text{unseen}}$ only.

We cannot directly use the baselines above for unseen relations since those relation embeddings are never trained. Therefore, we combine them with EmbedRule, which estimates the relation embedding by finding a number of most common relation sequences that cooccur with the unseen relation, and composing those embeddings thereafter (Yang et al., 2014). We superscript the name with an asterisk for models enhanced by EmbedRule (e.g. TransE*).

Results in Table 5 illustrate that our model is significantly less affected than other models when we drop 20% relations from the training and validation sets. We additionally report the results of dropping or adding 5%, 10%, 15% relations in Appendix A6.2. These results on FB15k-237 are summarized in Figure 4, where we see that our model obtains the better generalization ability.

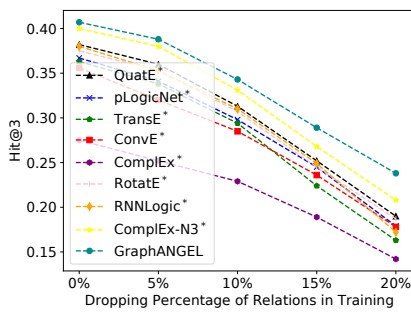

Figure 4: Illustrations of generalization ability for GraphANGEL against baselines.

## 4.5 RESULT ANALYSIS OF EXPLAINABLE ATTENTION MAP

Besides the performance, we further show that our model can produce explainable heat maps of attention scores across the discovered logic. We here provide an illustration on the recommendation task based on Douban Book graph, where we are required to predict the relation existence between each user and book pair. In Douban Book, we can define the graph patterns based on the node types, such as 4-cycle shaped patterns: User − Book − Author − Book denoted as $(b, a)$, User − Book − Year − Book denoted as $(b, y)$, and User − User − User − Book denoted as $(u, u)$. In Figure 5, the rows represent the supporting subgraphs while the columns represent the target subgraphs. Each cell represents the similarity between a target subgraph (at the top) and a supporting subgraph (at the bottom). The color of each cell shows the attention weight for corresponding pair of supporting and target subgraphs. We can

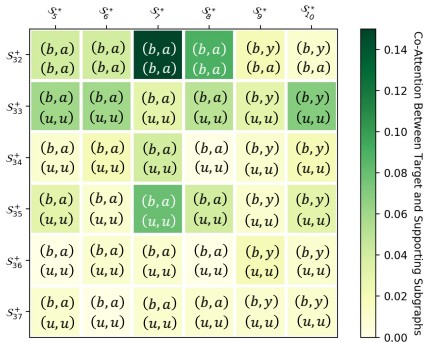

Figure 5: Illustrations of generated heat map of attention scores. See Appendix A6.3 for the full version with subgraph structure.

observe that the deep color of the cell located at target and supporting subgraphs following $(b, a)$ patterns, which indicates that the logic User ∧ Book ∧ Author ∧ Book ⇒ User ∧ Book has high confidence and can be strong evidence to support the relation prediction.

## 5 RELATED WORK

Heterogeneous graphs such as knowledge graphs (Bordes et al., 2013; Schlichtkrull et al., 2018) and social networks (Zhang et al., 2019a; Jin et al., 2020) that encode facts about the world surrounding us, have motivated work on automatically predicting new statements based on known ones. Roughly speaking, existing approaches can be summarized into three main branches, namely matrix factorization techniques (Nickel et al., 2011; 2012), statistical relational learning approaches (Richardson & Domingos, 2006; Singla & Domingos, 2005), and neural-embedding-based methods (Bordes et al., 2013; Dettmers et al., 2018). Our work focuses on the study of neural-embedding-based models as they embed nodes and relations into low-dimensional spaces (Bordes et al., 2013; Schlichtkrull et al., 2018; Trouillon et al., 2016; Yan et al., 2019; Teru et al., 2020), showing good scalability and strong generalization ability. One dominant paradigm (Ji et al., 2015; Lin et al., 2015) in this branch is constructed based on translation (Bordes et al., 2013) or rotation (Sun et al., 2019a) assumptions. The key idea behind this kind of models is that for a positive instance, the source node should be as close as possible to the target node through the relation, serving as a translation or rotation.

Although there are multiple successful stories in this line of works, these models are all trained on individual instances, regardless of their local neighborhood structures. As stated in (Schlichtkrull et al., 2018), explicitly modeling local structure can be an important supplement to help recover missing relations. Inspired by the success of GNNs in modeling structured neighborhood information, another line of literature (Schlichtkrull et al., 2018; Zhang & Chen, 2018) learns the relation embedding based on its neighborhood subgraph using graph convolution layers. With a similar approach, recent work (Zhang & Chen, 2019; Teru et al., 2020) illustrates the inductive capabilities of generalization to unseen nodes. However, these approaches fail to generalize to unseen relation types, as it is indirect to introduce the "neighborhood" for a relation type upon the graph structure.

Meanwhile, our work also relates to previous researches (Qu et al., 2019; Qu & Tang, 2019; Zhang et al., 2020; Qi et al., 2018; Zheng et al., 2019) studying to effectively combine GNNs techniques with symbolic logic rule-based approaches (Giarratano & Riley, 1998; Jackson, 1998; Lafferty et al., 2001; Taskar et al., 2012; Richardson & Domingos, 2006; Singla & Domingos, 2005). However, these methods, as originally proposed, are transductive in nature. Unlike our method, they still require learning relation type specific embeddings, whereas we treat the relation prediction as a graph pattern matching problem, independent of any particular relation type identity. There are also other work (Yang et al., 2014) that designs a logical rule extraction approach based on the potential generalization ability of transition-based methods. As this method can incorporate with various neural-embedding-based models, this set of models constitute our baselines in the inductive setting. Notably, comparing to the message passing neural network (Gilmer et al., 2017), the current prevailing GNN framework, as illustrated in Figure A2, GraphANGEL is able to avoid the GNNs' limitation of using neighbor nodes by using nodes satisfying certain logical patterns (See detailed discussion in Appendix A4.1).

Another popular link prediction approach is to directly infer the likelihood of relation existence using a local neighborhood or a local subgraph (Schlichtkrull et al., 2018; Zhang & Chen, 2018; Hu et al., 2020). A significant difference between those works and our approach is in the formulation of the loss function. Let $f_r$ be any function that maps a vector to a scalar score parametrized by $r$ and $\ell_r$ be a scalar function, then the prior works' loss function often take the form $L = \sum_{\langle s,r,t \rangle \in \mathcal{D}} \ell(f_r(\boldsymbol{h}^*), y)$, meaning that the score computation explicitly depends on the embedding of relation to be predicted. Ours, in contrast, has the form $L = \sum_{\langle s,r,t \rangle \in \mathcal{D}} \ell(d(\boldsymbol{h}^*, \boldsymbol{h}^+), d(\boldsymbol{h}^*, \boldsymbol{h}^-), y)$ where $\ell$ is a scalar function and $d$ is a distance function between two vectors, which are embeddings of a set of subgraphs. Consequently, our loss function $\ell$ does not require representation of $r$.

## 6 CONCLUSION

We propose a novel relation prediction framework that predicts the relations between each node pair based on the subgraph containing the pair and other subgraphs with identical graph patterns, and has a strong inductive bias for the generalization to unseen relation types. With these graph patterns, we introduce several graph pattern searching and sampling techniques, which can efficiently find subgraphs matching the patterns in triangle and quadrangle shapes. In the future, we plan to further extend GraphANGEL to more complex structures (i.e., compositional logical rules).

**Acknowledgments.** The Shanghai Jiao Tong University Team is supported by Shanghai Municipal Science and Technology Major Project (2021SHZDZX0102) and National Natural Science Foundation of China (62076161, 62177033). We would also like to thank Wu Wen Jun Honorary Doctoral Scholarship from AI Institute, Shanghai Jiao Tong University.

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

## A1  ILLUSTRATIONS OF NOTATIONS

Figure A1: An illustrated example of notations in GraphANGEL.

In this paper, we begin by bridging the logical expression and the graph pattern. Taking Figure A1 as example, we aim to predict the relation `live` between the source and target nodes (i.e., `Person E` and `London`). We consider the 3-cycle graph pattern. Formally, we use $\Pi_p$ where $p = 1, 2, \ldots, P$ to denote a certain graph pattern. For each $\Pi_p$, we use $\Pi_p^*$ to denote the corresponding target pattern, $\Pi_p^+$ to denote the corresponding supporting pattern, $\Pi_p^-$ to denote the corresponding refuting pattern. Concretely, as shown in Table 1, we can define $\Pi_p = \text{Edge}(x, z) \land \text{Edge}(z, y)$, $\Pi_p^* = \text{Edge}(x, z) \land \text{Edge}(z, y) \land \text{Source}(x) \land \text{Target}(y)$, $\Pi_p^+ = \text{Edge}(x, z) \land \text{Edge}(z, y) \land \text{Edge}_{\text{live}}(x, y)$, $\Pi_p^- = \text{Edge}(x, z) \land \text{Edge}(z, y) \land \neg\text{Edge}_{\text{live}}(x, y)$. According to these patterns, we can retrieve the corresponding subgraphs on the graph (as shown in Figure 1) and construct the corresponding subgraph sets. Figure A1 illustrates a target subgraph representing the logical expression `Watch(Person E, Arsenal)`$\land$`Base(Arsenal, London)`$\land$`Source(PersonE)` $\land$ `Target(London)`. We use $\mathcal{S}_{p,k}^*$ to denote the $k$-th sampled target subgraph following the target pattern $\Pi_p^*$ where $k = 1, 2, \ldots, K$, $\mathcal{S}_{p,q}^+$ to denote the $q$-th sampled supporting subgraph following the supporting pattern $\Pi_p^+$, $\mathcal{S}_{p,q}^-$ to dentoe the $q$-th sampled refuting subgraph following the refuting pattern $\Pi_p^-$ where $q = 1, 2, \ldots, Q$.

For each $\Pi_p$, after applying pattern retrieval and sampling introduced in Appendix A2, we build a set of target subgraphs denoted as $\{\mathcal{S}_{p,k}^*\}_{k=1}^K$, a set of supporting subgraphs denoted as $\{\mathcal{S}_{p,q}^+\}_{q=1}^Q$, a set of refuting subgraphs denoted as $\{\mathcal{S}_{p,q}^-\}_{q=1}^Q$. We then learn a representation for each subgraph, and obtain a set of corresponding target subgraph embeddings denoted as $\{e_{p,k}^*\}_{k=1}^K$, a set of corresponding supporting subgraph embeddings denoted as $\{e_{p,q}^+\}_{q=1}^Q$, a set of corresponding refuting subgraph embeddings denoted as $\{e_{p,q}^-\}_{q=1}^Q$.

## A2  PSEUDOCODE, CORRECTNESS PROOFS, AND COMPLEXITY ANALYSIS FOR 3-CYCLE AND 4-CYCLE PATTERN RETRIEVAL AND SAMPLING

In this section, we first summarize the complexity of retrieving and sampling subgraphs following the graph patterns in 3-cycle and 4-cycle with the following theorem:

**Theorem 1.** (Time Complexity of Retrieval and Sampling) *Given a graph $\mathcal{G} = (\mathcal{V}, \mathcal{E})$ and a graph pattern $\Pi_{\text{3-cycle}}$ in 3-cycle and a graph pattern $\Pi_{\text{4-cycle}}$ in 4-cycle, the time complexity of retrieving all the (supporting) subgraphs satisfying $\Pi_{\text{3-cycle}}$ is $O(|\mathcal{E}|^{\frac{3}{2}})$, of retrieving all the (supporting) subgraphs satisfying $\Pi_{\text{4-cycle}}$ is $O(\max(|\mathcal{E}|^{\frac{3}{2}}, N_{\text{4-cycle}}))$ where $N_{\text{4-cycle}}$ is the number of quadratic cycles in $\mathcal{G}$ and also the trivial lower bound of the time complexity. For uniform sampling algorithms, the time complexity of sampling $n_{\text{4-cycle}}$ supporting cases of $\Pi_{\text{4-cycle}}$ is $O(|\mathcal{E}|^{\frac{3}{2}} + n_{\text{4-cycle}})$. As there are usually more refuting cases than supporting ones, the time complexity of sampling $n$ refuting cases of $\Pi_{\text{3-cycle}}$ or $\Pi_{\text{4-cycle}}$ is $O(|\mathcal{V}| + |\mathcal{E}| + n)$.*

In the following subsections, we show the proof of Theorem 1 by providing the pseudocode of retrieval and sampling algorithms along with the correctness proof and complexity analysis in Appendix A2.1 and A2.2 respectively.

### A2.1 PSEUDOCODE, CORRECTNESS PROOFS, AND COMPLEXITY ANALYSIS FOR 3-CYCLE AND 4-CYCLE PATTERN RETRIEVAL

In this part, we first provide two algorithms for searching and retrieving supporting subgraphs following the graph patterns in 3-cycle (i.e., $\Pi_{\text{3-cycle}}$) in Part 1; and one algorithm for searching and retrieving supporting subgraphs following the graph patterns in 4-cycle (i.e., $\Pi_{\text{4-cycle}}$) in Part 2.

**Part 1.** In Algorithm 2, we present the pseuduocode for searching and retrieving all the supporting subgraphs following the pattern $\Pi_{\text{3-cycle}}$. The input is a graph $\mathcal{G} = (\mathcal{V}, \mathcal{E})$ and the output is a set $\mathcal{B}$ containing all the supporting subgraphs satisfying $\Pi_{\text{3-cycle}}$. We denote the degree of node $u$ by $d_u$, and the set of neighbor nodes of node $u$ in graph $\mathcal{G}$ by $\mathcal{N}_{\mathcal{G}}(u)$.

---

**Algorithm 2:** Search and Retrieval Algorithm A for $\Pi_{\text{3-cycle}}$

---

$\mathcal{V}_1 \leftarrow \{u | d_u > |\mathcal{E}|^{\frac{1}{2}}\}, \mathcal{V}_2 \leftarrow \{u | d_u \leq |\mathcal{E}|^{\frac{1}{2}}\}$
$\mathcal{B} \leftarrow \{\}$
**foreach** $u \in \mathcal{V}_1$ **do**
    **foreach** $v \in \mathcal{V}_1$ **do**
        **foreach** $w \in \mathcal{V}_1$ **do**
            **if** $\langle u, v \rangle \in \mathcal{E}$ and $\langle u, w \rangle \in \mathcal{E}$ and $\langle v, w \rangle \in \mathcal{E}$ **then**
                $\mathcal{B} \leftarrow \mathcal{B} \cup \{(u, v, w)\}$
            **end**
        **end**
    **end**
**end**
**foreach** $u \in \mathcal{V}_2$ **do**
    **foreach** $v \in \mathcal{N}_{\mathcal{G}}(u)$ **do**
        **foreach** $w \in \mathcal{N}_{\mathcal{G}}(u)$ **do**
            **if** $\langle v, w \rangle \in \mathcal{E}$ **then**
                $\mathcal{B} \leftarrow \mathcal{B} \cup \{(u, v, w)\}$
            **end**
        **end**
    **end**
**end**

---

Now, we investigate the correctness and the time complexity of Algorithm 2.

*Proof.* For any 3-cycle (i.e., 3 nodes connected by 3 edges) $(u, v, w)$, if the degrees of all three nodes are greater than $|\mathcal{E}|^{\frac{1}{2}}$ (i.e., $d_u > |\mathcal{E}|^{\frac{1}{2}}$ and $d_v > |\mathcal{E}|^{\frac{1}{2}}$ and $d_w > |\mathcal{E}|^{\frac{1}{2}}$), it will be enumerated in the first loop. Otherwise, if the degree of any node is not greater than $|\mathcal{E}|^{\frac{1}{2}}$, the 3-cycle will be enumerated in the second loop.

In order to analyze the time complexity, we first investigate an upper bound of $\mathcal{V}_1 = \{u | d_u > |\mathcal{E}|^{\frac{1}{2}}\}$:

$$2|\mathcal{E}| = \sum_{u \in \mathcal{V}} d_u \geq \sum_{u \in \mathcal{V}_1} d_u > \sum_{u \in \mathcal{V}_1} |\mathcal{E}|^{\frac{1}{2}} \Rightarrow |\mathcal{V}_1| < 2|\mathcal{E}|^{\frac{1}{2}}. \tag{4}$$

For the first main loop of the algorithm, we have

$$\sum_{u,v,w\in\mathcal{V}_1} 1 = |\mathcal{V}_1|^3 < 8|\mathcal{E}|^{\frac{3}{2}} = O(|\mathcal{E}|^{\frac{3}{2}}). \tag{5}$$

For the second main loop, we have

$$\sum_{u\in\mathcal{V}_2}\sum_{\langle u,v\rangle\in\mathcal{E}}\sum_{\langle u,v\rangle\in\mathcal{E}} 1 = \sum_{u\in\mathcal{V}_2} d_u^2 \le \sum_{u\in\mathcal{V}_2}|\mathcal{E}|^{\frac{1}{2}} d_u \le 2|\mathcal{E}|^{\frac{3}{2}} = O(|\mathcal{E}|^{\frac{3}{2}}). \tag{6}$$

Summarizing the analysis above, we conclude that the overall time complexity is $O(|\mathcal{E}|^{\frac{3}{2}})$. $\qquad\square$

Alternatively, we provide another searching and retrieving algorithm for all the supporting subgraphs following $\Pi_{3-\texttt{cycle}}$ in Algorithm 3. The inputs and outputs of the algorithm are consistent with Algorithm 2.

---

**Algorithm 3:** Search and Retrieval Algorithm B for $\Pi_{3-\texttt{cycle}}$

---

$\mathcal{B} \leftarrow \{\}$
**foreach** $u \in \mathcal{V}$ **do**
    **foreach** $v \in \mathcal{N}_\mathcal{G}(u)$ where $d_v \le d_u$ **do**
        **foreach** $w \in \mathcal{N}_\mathcal{G}(v)$ where $d_w \le d_v$ **do**
            **if** $\langle v, w\rangle \in \mathcal{E}$ **then**
                $\mathcal{B} \leftarrow \mathcal{B} \cup \{(u,v,w)\}$
            **end**
        **end**
    **end**
**end**

---

Next, we analyze and prove the correctness and the time complexity of Algorithm 3.

*Proof.* It is not difficult to find that each 3-cycle $(u, v, w)$ satisfying $d_u \ge d_v \ge d_w$ is enumerated.

To analyze the time complexity, let us consider how many $w$s are enumerated:

$$\sum_{u\in\mathcal{V}}\sum_{\substack{\langle u,v\rangle\in\mathcal{E}\\ d_v\le d_u}}\sum_{\substack{\langle w,v\rangle\in\mathcal{E}\\ d_w\le d_v}} 1 \le \sum_{u\in\mathcal{V}}\sum_{\substack{\langle u,v\rangle\in\mathcal{E}\\ d_v\le d_u}}\sum_{\langle w,v\rangle\in\mathcal{E}} 1 = \sum_{u\in\mathcal{V}}\sum_{\substack{\langle u,v\rangle\in\mathcal{E}\\ d_v\le d_u}} d_v. \tag{7}$$

From Eq. (4), we know that for each $v$ satisfying $d_v > |\mathcal{E}|^{\frac{1}{2}}$, since $d_u \ge d_v > |\mathcal{E}|^{\frac{1}{2}}$, there are at most $2|\mathcal{E}|^{\frac{1}{2}}$ different $u$s in the outer loop. Hence, we have

$$
\begin{aligned}
\sum_{u\in\mathcal{V}}\sum_{\substack{\langle u,v\rangle\in\mathcal{E}\\ d_v\le d_u}} d_v &= \sum_{v\in\mathcal{V}} d_v \sum_{\substack{\langle u,v\rangle\in\mathcal{E}\\ d_v\le d_u}} 1 \\
&= \sum_{v\in\mathcal{V}}\left(\left[d_v\le|\mathcal{E}|^{\frac{1}{2}}\right]+\left[d_v>|\mathcal{E}|^{\frac{1}{2}}\right]\right)d_v\sum_{\substack{\langle u,v\rangle\in\mathcal{E}\\ d_v\le d_u}} 1 \\
&= \sum_{v\in\mathcal{V}}\left(\left[d_v\le|\mathcal{E}|^{\frac{1}{2}}\right]d_v\sum_{\substack{\langle u,v\rangle\in\mathcal{E}\\ d_v\le d_u}} 1+\left[d_v>|\mathcal{E}|^{\frac{1}{2}}\right]d_v\sum_{\substack{\langle u,v\rangle\in\mathcal{E}\\ d_v\le d_u}} 1\right) \\
&\le \sum_{v\in\mathcal{V}}\left(\left[d_v\le|\mathcal{E}|^{\frac{1}{2}}\right]|\mathcal{E}|^{\frac{1}{2}}d_v+\left[d_v>|\mathcal{E}|^{\frac{1}{2}}\right]|\mathcal{E}|^{\frac{1}{2}}d_v\right) \\
&= |\mathcal{E}|^{\frac{1}{2}}\sum_{v\in\mathcal{V}} d_v = 2|\mathcal{E}|^{\frac{3}{2}},
\end{aligned}
\tag{8}
$$

where $[\cdot]$ represents the boolean indicator function (i.e., $[x]=1$ if $x$ is true; otherwise false).

Therefore, the overall time complexity is also $O(|\mathcal{E}|^{\frac{3}{2}})$. $\qquad\square$

**Part 2.** In Algorithm 4, we demonstrate the algorithm for searching and retrieving all the supporting subgraphs following graph patterns in 4-cycles (i.e., $\Pi_{4-\text{cycle}}$). The inputs, outputs, and notations are similar to those in Algorithm 2.

---

**Algorithm 4:** Search and Retrieval Algorithm for $\Pi_{4-\text{cycle}}$

---

$\mathcal{B} \leftarrow \{\}$
**foreach** $u \in \mathcal{V}$ **do**
    $\mathcal{T}_x \leftarrow \{\}$ for each $x \in \mathcal{V}$
    **foreach** $v \in \mathcal{N}_{\mathcal{G}}(u)$ where $d_v \leq d_u$ **do**
        **foreach** $w \in \mathcal{N}_{\mathcal{G}}(v)$ where $d_w \leq d_u$ **do**
            **foreach** $x \in \mathcal{T}_w$ **do**
                $\mathcal{B} \leftarrow \mathcal{B} \cup \{(u, v, w, x)\}$
            **end**
            $\mathcal{T}_w \leftarrow \mathcal{T}_w \cup \{v\}$
        **end**
    **end**
**end**

---

We provide the correctness and the time complexity of Algorithm 4 as follows.

*Proof.* For correctness, we can see from the algorithm that every 4-cycle $(u, v, w, x)$ satisfying $d_u \geq \max(d_v, d_w, d_x)$ is enumerated.

For time complexity, let $N_{4-\text{cycle}}$ denote the number of quadratic cycles in $\mathcal{G}$. We first have the time complexity $\Theta(N_{4-\text{cycle}})$ of the innermost loop. For the outer three loops, let us consider how many $w$s are enumerated:

$$\sum_u \sum_{\substack{\langle u,v \rangle \in \mathcal{E} \\ d_u \leq d_v}} \sum_{\substack{\langle w,v \rangle \in \mathcal{E} \\ d_w \leq d_u}} 1 \leq \sum_u \sum_{\substack{\langle u,v \rangle \in \mathcal{E} \\ d_u \leq d_v}} \sum_{\langle w,v \rangle \in \mathcal{E}} 1 = \sum_u \sum_{\substack{\langle u,v \rangle \in \mathcal{E} \\ d_u \leq d_v}} d_v. \tag{9}$$

From Eq. (8), we have

$$\sum_u \sum_{\langle u,v \rangle \in \mathcal{E}, \ d_u \leq d_v} d_v \leq 2|\mathcal{E}|^{\frac{3}{2}}. \tag{10}$$

Thus, the overall time complexity is $O(\max(|\mathcal{E}|^{\frac{3}{2}}, N_{4-\text{cycle}}))$. $\qquad\qquad\square$

### A2.2    Pseudocode, Correctness Proofs, and Complexity Analysis for 3-cycle and 4-cycle Pattern Sampling

From the subsection above, we know that given a graph $\mathcal{G} = (\mathcal{V}, \mathcal{E})$, we have different algorithmic complexities depending on the graph pattern $\Pi$. For $\Pi_{3-\text{cycle}}$ (i.e., 3-cycles), the time complexity of retrieving all supporting subgraphs is $O(|\mathcal{E}|^{\frac{3}{2}})$. For $\Pi_{4-\text{cycle}}$ (i.e., 4-cycles), the time complexity is $O(\max(|\mathcal{E}|^{\frac{3}{2}}, N_{4-\text{cycle}}))$, where $N_{4-\text{cycle}}$ is the number of quadratic cycles in $\mathcal{G}$.

However, for large-scale graphs, it is impractical to retrieve all quadratic-cycle graph patterns since $N_{4-\text{cycle}}$ could be too large. In addition, the number of refuting cases for both $\Pi_{3-\text{cycle}}$ and $\Pi_{4-\text{cycle}}$ may also be large. These problems may lead to the high computational cost of these search algorithms. Realizing this, we further introduce the technique of uniform sampling to keep the time complexity within an acceptable level.

Therefore, in the following, we first provide one algorithm for uniformly sampling and retrieving graph patterns following $\Pi^+_{4-\text{cycle}}$ in Part 1; and two algorithms for uniform sampling and retrieving graph patterns following the refuting patterns in 3-cycle shape (i.e., $\Pi^-_{3-\text{cycle}}$) and 4-cycle shape (i.e., $\Pi^-_{4-\text{cycle}}$) in Part 2. As a reminder, while practical, another approach to obtain graph patterns following $\Pi^-_{3-\text{cycle}}$ and $\Pi^-_{4-\text{cycle}}$ is random walk, although random walk is often not efficient enough.

**Part 1.** We present the algorithm for sampling and retrieving all the supporting subgraphs following $\Pi_{4-\text{cycle}}$ in Algorithm 5. The inputs to the algorithm are a graph $\mathcal{G} = (\mathcal{V}, \mathcal{E})$ and $n_{4-\text{cycle}}$, rep-

resenting the number of subgraphs to sample. The output is a set $\mathcal{B}$ containing all sampled graph patterns.

---

**Algorithm 5:** Uniform Sampling and Retrieval Algorithm for $\Pi^+_{4-\text{cycle}}$

---

$\mathcal{B} \leftarrow \{\}$
**foreach** $u \in \mathcal{V}$ **do**
    $\mathcal{T}_{uw} \leftarrow \{\}$ for $w \in \mathcal{V}$
    **foreach** $v \in \mathcal{N}_{\mathcal{G}}(u)$ where $d_v \leq d_u$ **do**
        **foreach** $w \in \mathcal{N}_{\mathcal{G}}(v)$ where $d_w \leq d_u$ **do**
            $\mathcal{T}_{uw} \leftarrow \mathcal{T}_{uw} \cup \{v\}$
        **end**
    **end**
**end**
**foreach** $(u, w)$ where $\mathcal{T}_{uw} \neq \varnothing$ **do**
    $c_{uw} \leftarrow |\mathcal{T}_{uw}|(|\mathcal{T}_{uw}| - 1)$
**end**
$c \leftarrow \sum_{uw} c_{uw}$
**foreach** $(u, w)$ where $\mathcal{T}_{uw} \neq \varnothing$ **do**
    $p_{uw} \leftarrow c_{uw}/c$
**end**
**for** $i \leftarrow 1$ to $n_{4-\text{cycle}}$ **do**
    Sample $(u, w)$ where $\mathcal{T}_{uw} \neq \varnothing$ with probability $p_{uw}$
    Uniformly sample two different nodes $u$ and $x$ from $\mathcal{T}_{uw}$
    **if** $\langle u, w \rangle \in \mathcal{E}$ **then**
        $\mathcal{B} \leftarrow \mathcal{B} \cup \{(u, v, w, x)\}$
    **end**
**end**

---

We provide the correctness and the time complexity of Algorithm 5 as follows.

*Proof.* For correctness, we can observe from the algorithm that every 4-cycle $(u, v, w, x)$ satisfying $d_u \geq \max(d_v, d_w, d_x)$ is counted in $\mathcal{T}_{uw}$ exactly once, where $v \in \mathcal{T}_{uw}$ and $x \in \mathcal{T}_{uw}$. We sample $(u, w)$ with the probability of occurrence of the 4-cycle $(u, v, w, x)$, then sample different $v$ and $x$ uniformly from $\mathcal{T}_{uw}$.

For time complexity, it has been proved in Algorithm 4 that the time complexity of computing $\mathcal{T}$ is $O(|\mathcal{E}|^{\frac{3}{2}})$. Notably, there are $O(|\mathcal{E}|^{\frac{3}{2}})$ non-empty $\mathcal{T}$s.

Therefore, if we sample $(u, w)$ through Alias Method (Walker, 1977), then the overall time complexity is $O(|\mathcal{E}|^{\frac{3}{2}} + n_{4-\text{cycle}})$. $\qquad\square$

**Part 2.** We further provide two path sampling algorithms. In most practical cases, there are many more refuting examples than supporting ones, and if we assume so, then Algorithm 6 can be used to sample graph patterns following refuting patterns in 3-cycle shape (i.e., $\Pi^-_{3-\text{cycle}}$), and Algorithm 7 to sample graph patterns following refuting patterns in 4-cycle shape (i.e., $\Pi^-_{4-\text{cycle}}$).

Algorithm 6 displays the algorithm for uniformly sampling $n$ subgraphs following $\Pi^-_{3-\text{cycle}}$ in $\mathcal{G}$. The inputs, outputs, notations are identical to those in Algorithm 5.
We provide the correctness and the time complexity of Algorithm 5 as follows.

*Proof.* For correctness, as shown in the algorithm, we formulate the sampling of refuting cases of $\Pi_{3-\text{cycle}}$ into uniformly sampling three-node paths, namely $(u, v, w)$. We first sample the intermediate node of the path (i.e., $v$), where there are $d_v(d_v - 1)/2$ three-node paths with $v$ as its intermediate node. Next, we sample $v$ with probability $p_v$ to ensure uniformity, where $p_v = d_v(d_v - 1)/\sum_v d_v(d_v - 1)$, and then sample its two neighbors to form the path.

Using Alias Method, the overall time complexity is $O(|\mathcal{V}| + |\mathcal{E}| + n)$, where the computation for the degree requires $O(|\mathcal{E}|)$, the first two loops require $O(|\mathcal{V}|)$, and the last loop requires $O(n)$. $\qquad\square$

---

**Algorithm 6:** Uniform Sampling and Retrieval Algorithm for $\Pi_{3-\texttt{cycle}}^-$

---

$\mathcal{B} \leftarrow \{\}$
**foreach** $u \in \mathcal{V}$ **do**
  $\quad c_u \leftarrow d_u(d_u - 1)$
**end**
$c \leftarrow \sum_u c_u$
**foreach** $u \in \mathcal{V}$ **do**
  $\quad p_u \leftarrow c_u/c$
**end**
**for** $i \leftarrow 1$ to $n$ **do**
  $\quad$ Sample $v$ with probability $p_v$
  $\quad$ Uniformly sample two different neighbor nodes (i.e., $u$ and $w$) of $v$
  $\quad$ **if** $\langle u, w \rangle \notin \mathcal{E}$ **then**
  $\quad\quad \mathcal{B} \leftarrow \mathcal{B} \cup \{(u, v, w)\}$
  $\quad$ **end**
**end**

---

**Algorithm 7:** Uniform Sampling and Retrieval Algorithm for $\Pi_{4-\texttt{cycle}}^-$

---

$\mathcal{B} \leftarrow \{\}$
**foreach** $\langle u, v \rangle \in \mathcal{E}$ **do**
  $\quad$ **if** $d_v = 2$ and $d_u = 2$ and $|\mathcal{N}_\mathcal{G}(u) \cap \mathcal{N}_\mathcal{G}(v)| = 1$ **then**
  $\quad\quad c_{uv} \leftarrow 0$
  $\quad$ **end**
  $\quad$ **else**
  $\quad\quad c_{uv} \leftarrow (d_u - 1) \cdot (d_v - 1)$
  $\quad$ **end**
**end**
$c \leftarrow \sum_{uv} c_{uv}$
**foreach** $\langle u, v \rangle \in \mathcal{E}$ **do**
  $\quad p_{uv} \leftarrow c_{uv}/c$
**end**
**for** $i \leftarrow 1$ to $n$ **do**
  $\quad$ Sample $\langle v, w \rangle$ with probability $p_{vw}$
  $\quad$ Uniformly sample a neighbor node $u$ of $v$
  $\quad$ Uniformly sample a neighbor node $x$ of $w$
  $\quad$ **if** $u \neq x$ and $\langle u, x \rangle \notin \mathcal{E}$ **then**
  $\quad\quad \mathcal{B} \leftarrow \mathcal{B} \cup \{(u, v, w, x)\}$
  $\quad$ **end**
**end**

---

In Algorithm 7, we present the algorithm for uniformly sampling graph patterns following $\Pi_{4-\texttt{cycle}}^-$ in $\mathcal{G}$. We provide the correctness and the time complexity of Algorithm 7 as follows.

*Proof.* Analogous to Algorithm 6, we formulate the sampling of refuting cases of $\Pi_{4-\texttt{cycle}}$ into uniformly sampling four-node paths namely $(u, v, w, x)$. To first analyze the correctness of the algorithm, we first sample the intermediate edge of the path (i.e., $\langle v, w \rangle$). There are $(d_v - 1) \cdot (d_w - 1)$ three-edge paths (including 3-cycles) where the intermediate edge is $\langle v, w \rangle$. Hence, we sample $\langle v, w \rangle$ with the probability proportional to $(d_v - 1) \cdot (d_w - 1)$, and then sample a neighbor node $u$ from $v$ and another one $x$ from $w$, and reject the case when it forms a 3-cycle.

To guarantee the time complexity, we do not sample the edge $\langle v, w \rangle$ when $v$ and $w$ have the only common neighbor, thus $\frac{|\mathcal{N}_\mathcal{G}(v) \cap \mathcal{N}_\mathcal{G}(w)|}{(d_v - 1) \cdot (d_w - 1)} \leq \frac{1}{2}$ when $c \neq 0$, i.e., we reject with probability no more than $\frac{1}{2}$, the expected time complexity is $O(|\mathcal{V}| + |\mathcal{E}| + n)$. $\qquad\square$

---

**Algorithm 8:** Implementation of GraphANGEL

---

**Input:** Graph $\mathcal{G}$, Patterns $\Pi_1, \ldots, \Pi_P$, Source-target node dataset $\mathcal{D}$.
**Output:** Prediction $\hat{y}$ for each $\langle s, r, t \rangle$.
**for** *each pattern* $\Pi_p$ **do**
  Construct $\Pi_p^*, \Pi_p^+, \Pi_p^-$.
  Search/Sample and retrieve subgraphs following $\Pi_p$ through Algorithms 2,3,4,5,6,7.
  Store all the subgraphs in $\mathcal{B}$.
**end**
**for** *each* $\langle s, r, t \rangle \in \mathcal{D}$ **do**
  **for** *each logic* $\Pi_p$ **do**
    Sample and retrieve $K$ subgraphs $\mathcal{S}_{p,k}^*$ matching $\Pi_p^*$ in $\mathcal{B}$.
    Sample and retrieve $Q$ subgraphs $\mathcal{S}_{p,q}^+$ matching $\Pi_p^+$ in $\mathcal{B}$.
    Sample and retrieve $Q$ subgraphs $\mathcal{S}_{p,q}^-$ matching $\Pi_p^-$ in $\mathcal{B}$.
    Compute $\boldsymbol{e}_{p,k}^*, \boldsymbol{e}_{p,q}^+, \boldsymbol{e}_{p,q}^-$ according to Eq. (1).
  **end**
  Compute $s^+$, $s^-$ according to Eq. (2).
  Calculate $\hat{y}$ via $\hat{y} = \frac{s^+}{s^+ + s^-}$.
  Re-sample and retrieve subgraphs following $\Pi_p$ through Algorithms 6,7,5.
  Update $\mathcal{B}$ with the re-sampled subgraphs.
  Update parameters by minimizing loss $L$ shown in Eq. (3).
**end**

---

Notably, these algorithms are valuable because the number of the refuting cases are considerable in practice, and they make GraphANGEL more relevant for practical deployment.

## A3 IMPLEMENTATION ALGORITHM OF GRAPHANGEL

### A3.1 ALGORITHM FOR IMPLEMENTATION OF GRAPHANGEL ON STATIC GRAPHS

For the heterogeneous graph based recommendation task, we follow (Hu et al., 2020; Wang et al., 2019b; Zhang et al., 2019a) to predict the existence of edge between each given node pair $\langle s, t \rangle$, while in the knowledge graph completion task, we follow (Bordes et al., 2013; Sun et al., 2019a; Dettmers et al., 2018) to mask head and tail entities instead of relation prediction. As the input of our model is a triplet $\langle s, r, t \rangle$, one can score all the possible triplets no matter either relation $r$ or tail entity (i.e., node) $t$ is masked. We then rank all these tuples for evaluation.

**Overall Time Complexity of Original Algorithm.** Theorem 1 shows the time complexity of searching and retrieving the subgraphs following the pre-defined 3-cycle and 4-cycle shaped patterns: $O(\max(|\mathcal{E}|^{\frac{3}{2}}, n_{4-\texttt{cycle}}))$ for sampling $n_{4-\texttt{cycle}}$ subgraphs satisfying $\Pi_{4-\texttt{cycle}}^+$, $O(|\mathcal{V}| + |\mathcal{E}| + n)$ for sampling $n$ refuting cases of $\Pi_{3-\texttt{cycle}}^-$ or $\Pi_{4-\texttt{cycle}}^-$.

Besides the subgraph retrieval part, the main components of GraphANGEL are the representation computation part (i.e., $\Phi$) and the similarity computation part (i.e., $\Psi$). We further analyze the time complexity of these parts as follows.

For $\Phi$, as described in Section 3, we employ R-GCN (Schlichtkrull et al., 2018) (See Appendix A3.3 for details). One can use other GNN architectures (e.g., R-GAT (Busbridge et al., 2019)). Hence, without loss of generality, we analyze the time complexity of the general GNNs following the message passing framework (Gilmer et al., 2017), most of whose update functions can be formulated as $\boldsymbol{H}^{(l+1)} = \texttt{ReLU}((\boldsymbol{L} + \boldsymbol{I})\boldsymbol{H}^{(l)}\boldsymbol{W}^{(l)})$, where $\boldsymbol{H} \in \mathbb{R}^{|\mathcal{V}| \times d_{l+1}}$ is the embedding matrix obtained after $l + 1$ steps of propagation, and $\boldsymbol{L}$ represents the Laplacian matrix for the graph, defined by $\boldsymbol{L} = \boldsymbol{D}^{-\frac{1}{2}}\boldsymbol{A}\boldsymbol{D}^{-\frac{1}{2}}$, $\boldsymbol{A}$ is the adjacency matrix and $\boldsymbol{D}$ is the diagonal degree matrix, $\boldsymbol{W}^l \in \mathbb{R}^{d_l \times d_{l-1}}$ is the trainable transformation matrix. The $l$-th propagation layer, the matrix multiplication has computational complexity $O(|\boldsymbol{L}^+| \cdot d_l \times d_{l-1})$, where $|\boldsymbol{L}^+|$ is the number of nonzero entities in the Laplacian matrix, and the $d_l$ and $d_{l-1}$ are the current and previous transformation size. Then, the time complexity of $\Phi$ with $L$ layers is $O(\sum_{l=1}^{L} |\boldsymbol{L}^+| \cdot d_l \times d_{l-1})$.

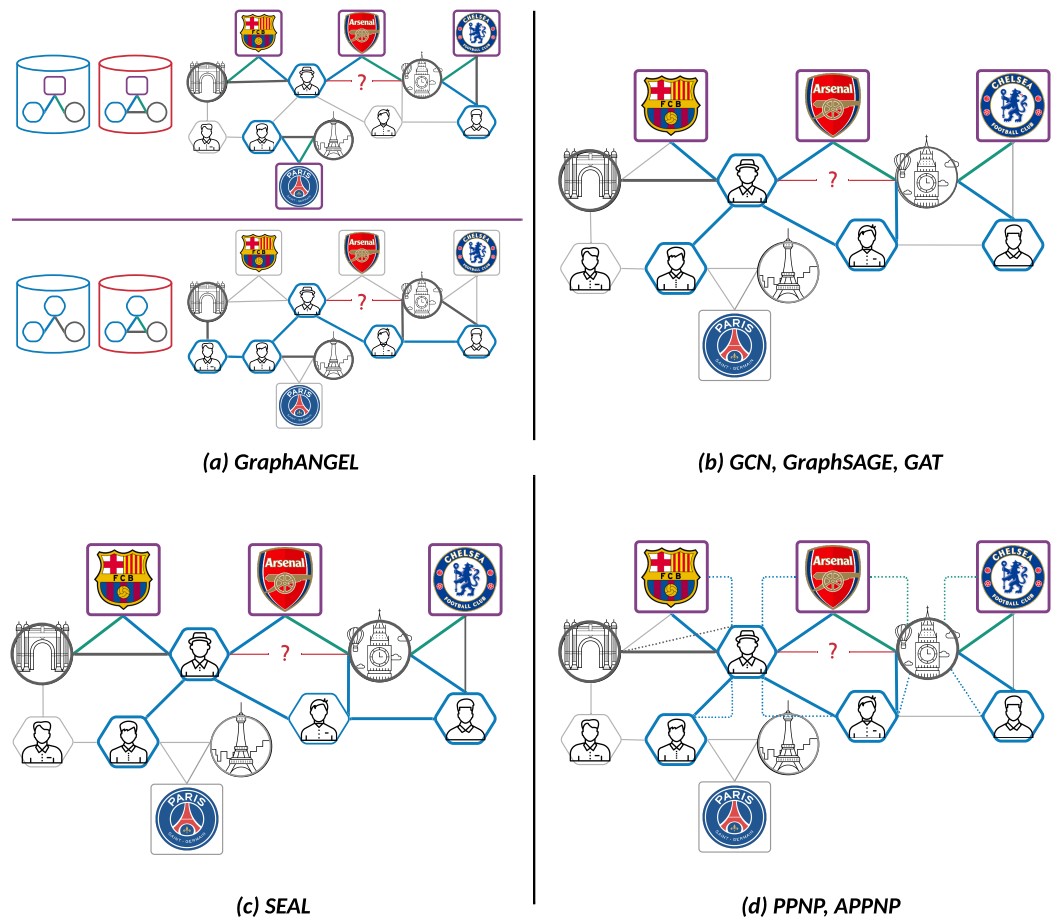

*(a) GraphANGEL*       *(b) GCN, GraphSAGE, GAT*

*(c) SEAL*       *(d) PPNP, APPNP*

Figure A2: An illustrated example of nodes and edges involved in GraphANGEL and other existing approaches, where nodes and edges in grey are not activated while the other colors of nodes and edges represent their types. As introduced in Table 2, we here show the results of two graph patterns in 3-cycle shape. For clarity and simplicity, in this case, the number of layers of GNN models are constrained as one.

For $\Psi$, as Section 3 shows, we use a co-attention mechanism (Lu et al., 2016) (See Appendix A3.4 for details). Without loss of generality, let $N_{\texttt{head}}$ denote the number of heads. Then, the time complexity of $\Psi$ is $O((P \times K) \cdot (P \times Q) \cdot d^2 \cdot N_{\texttt{head}})$.

Then, the overall time complexity of Algorithm 1 is to sum up all the above computation costs.

In Algorithm 1, we retrieve and sample subgraphs for every source-target node pair, which can be time-consuming. Algorithm 8 introduces additional steps that precompute all the graph patterns and their satisfying subgraphs as a *data preprocessing step*.

The intuition behind this is very straightforward that, as shown in Table 2, the graph patterns considered in this paper are quite simple, involving at most four nodes. For knowledge graphs, it's acceptable to pre-compute all the supporting subgraphs of 3-cycle shaped graph pattern using Algorithm 2 or 3. We also develop a novel search and retrieval algorithm (as shown in Algorithm 4) to retrieve all the supporting subgraphs with $O(\max(|\mathcal{E}|^{\frac{3}{2}}, N_{\texttt{4-cycle}}))$. However, as the number of supporting cases of 4-cycle shaped graph pattern (i.e., $N_{\texttt{4-cycle}}$) is too large (as shown in Table A3), while practical, we uniformly sample $n_{\texttt{4-cycle}}$ subgraphs following supporting graph patterns in 4-cycle shape using Algorithm 6. $n_{\texttt{4-cycle}}$ is a hyper-parameter, which we set as the number of 3-cycles in graph (i.e., $n_{\texttt{4-cycle}} = N_{\texttt{3-cycle}}$). All these subgraphs into a buffer $\mathcal{B}$. For heterogeneous graphs, since the graph schema of these graphs are usually not complex. We can cover all possible graph patterns by enumerating on the graph schema. Similarly, this enables us to pre-compute the graph patterns and store all the retrieved and sampled subgraphs following each graph pattern in the buffer $\mathcal{B}$. We then can build an index of the buffer by the graph patterns as well as the subgraphs following each graph pattern. During training, we sample and retrieve subgraphs in the buffer instead of the whole graph for each source-target node pair. Compared with the original one in Algorithm 1, we reduce the complexity and improve the efficiency. We can further set a update interval to refresh $\mathcal{B}$

by re-sampling the supporting cases of $\Pi_{4-\text{cycle}}$ and refuting cases of $\Pi_{3-\text{cycle}}$ and $\Pi_{4-\text{cycle}}$. While practical, we find that whether refreshing or not would not bring much influence, and establish an ablation study for the buffer size in Appendix A6.6.

Let $n_{3-\text{cycle}}^-, n_{4-\text{cycle}}^-$ denote the number of refuting samples of $\Pi_{3-\text{cycle}}^-$ and $\Pi_{4-\text{cycle}}^-$ respectively. In practice, we set $n = n_{3-\text{cycle}}^- = n_{4-\text{cycle}}^- = 2 \cdot N_{3-\text{cycle}}$. We sample $n$ refuting cases of $\Pi_{3-\text{cycle}}$ and $\Pi_{4-\text{cycle}}$ respectively and store them into $\mathcal{B}$.

**Space Complexity of Storing subgraphs.** From the above analysis, we know that if we choose to store all the supporting subgraphs, the space complexity of storing these subgraphs is $O(N_{3-\text{cycle}} + N_{4-\text{cycle}})$. While practical, it is $O(N_{3-\text{cycle}})$. And for the subgraphs of refuting cases, the space complexity is $O(n_{3-\text{cycle}}^- + n_{4-\text{cycle}}^-)$, and $O(n)$ for implementation.

Notably, this space complexity is different from the space complexity of the algorithms in Theorem 1, which is the same as its time complexity.

---

**Algorithm 9:** Search and Retrieval Algorithm for $\Pi_{3-\text{cycle}}^+$ in (a) Shape

---

$\mathcal{V}_1 \leftarrow \{u \in \mathcal{V}_{\text{old}} | d_u > \sigma\}, \mathcal{V}_2 \leftarrow \{u \in \mathcal{V}_{\text{old}} | d_u \leq \sigma\}$ where $\sigma \leftarrow |\mathcal{E}_{\text{old}}|^{\frac{1}{2}} / (|\Delta \mathcal{V}_{\text{new}}^+|^{\frac{1}{4}} |\mathcal{V}_{\text{old}}|^{\frac{1}{4}})$
$\mathcal{B} \leftarrow \{\}$
**foreach** $u \in \mathcal{V}_1$ **do**
    **foreach** $v \in \mathcal{V}_1$ **do**
        **foreach** $w \in \Delta \mathcal{V}_{\text{new}}^+$ **do**
            **if** $\langle u, v \rangle \in \mathcal{E}_{\text{old}}$ and $\langle u, w \rangle \in \Delta \mathcal{E}_{\text{old}}^+$ and $\langle v, w \rangle \in \Delta \mathcal{E}_{\text{old}}^+$ **then**
                $\mathcal{B} \leftarrow \mathcal{B} \cup \{(u, v, w)\}$
            **end**
        **end**
    **end**
**end**
**foreach** $u \in \mathcal{V}_2$ **do**
    **foreach** $v \in \mathcal{N}_{\mathcal{G}_{\text{old}}}(u)$ **do**
        **foreach** $w \in \mathcal{N}_{\Delta \mathcal{G}_{\text{old}}^+}(u)$ **do**
            **if** $\langle v, w \rangle \in \Delta \mathcal{E}_{\text{old}}^+$ **then**
                $\mathcal{B} \leftarrow \mathcal{B} \cup \{(u, v, w)\}$
            **end**
        **end**
    **end**
**end**

---

### A3.2 ALGORITHM FOR IMPLEMENTATION OF GRAPHANGEL ON DYNAMIC GRAPHS

Note that all the algorithms introduced in Appendix A2 mainly focus on searching and retrieving the subgraphs in the context of the static graphs. However, the graphs used in the real-world scenario almost are dynamic graphs, which always need to be modified and updated. In the followings, we show the (incremental) subgraph retrieval algorithms which updates the old buffer (denoted as $\mathcal{B}_{\text{old}}$) calculated by the old graph (denoted as $\mathcal{G}_{\text{old}}$) to the new buffer (denoted as $\mathcal{B}_{\text{new}}$) computed by new graph (denoted as $\mathcal{G}_{\text{new}}$). Formally, we use $\mathcal{G}_{\text{new}} = (\mathcal{V}_{\text{new}}, \mathcal{E}_{\text{new}})$ and $\mathcal{G}_{\text{old}} = (\mathcal{V}_{\text{old}}, \mathcal{E}_{\text{old}})$ to denote the new graph and the old graph respectively, where $\mathcal{V}_{\text{old}}$ and $\mathcal{V}_{\text{new}}$ denote node sets, $\mathcal{E}_{\text{old}}$ and $\mathcal{E}_{\text{new}}$ denote node sets. We do not introduce the relation sets, as our subgraph retrieval does not consider the relations. We use $\boldsymbol{A}_{\text{old}}$ and $\boldsymbol{A}_{\text{new}}$ to denote the adjacency matrices of $\mathcal{G}_{\text{new}}$ and $\mathcal{G}_{\text{old}}$ regardless of the edge types (i.e., relations).

Let $\Delta \mathcal{G}^- = (\Delta \mathcal{V}^-, \Delta \mathcal{E}^-)$ denote a graph including all the nodes and edges removed from $\mathcal{G}_{\text{old}}$ comparing to $\mathcal{G}_{\text{new}}$; and $\Delta \mathcal{G}^+ = (\Delta \mathcal{V}^+, \Delta \mathcal{E}^+)$ denote a graph consisting of all the nodes and edges added to $\mathcal{G}_{\text{old}}$ comparing to $\mathcal{G}_{\text{new}}$. As some edges in $\Delta \mathcal{G}^+$ may connect the a node in $\mathcal{G}_{\text{old}}$ and a new node, we can further divide $\Delta \mathcal{G}^+$ into two graphs. One is a graph including all the new nodes and edges connecting these nodes, denoted as $\Delta \mathcal{G}_{\text{new}}^+ = (\Delta \mathcal{V}_{\text{new}}^+, \Delta \mathcal{E}_{\text{new}}^+)$. And the rest edges together with corresponding nodes construct another graph denoted as $\Delta \mathcal{G}_{\text{old}}^+ = (\Delta \mathcal{V}_{\text{old}}^+, \Delta \mathcal{E}_{\text{old}}^+)$. Hence, putting $\Delta \mathcal{G}^-$ aside, the original new graph $\mathcal{G}_{\text{new}}$ can be divided into three parts to consider: $\mathcal{G}_{\text{old}}$,

---

**Algorithm 10:** Generation of Uniform Sampling Rate for Each Node and Edge

---

\# Following is Generation of Uniform Sampling Rate for Each Node
**foreach** $u \in \mathcal{V}$ **do**
  |   $c_u \leftarrow d_u(d_u - 1)$
**end**
$c \leftarrow \sum_u c_u$
**foreach** $u \in \mathcal{V}$ **do**
  |   $p_u \leftarrow c_u/c$
**end**
\# Following is Generation of Uniform Sampling Rate for Each Edge
**foreach** $\langle u, v \rangle \in \mathcal{E}$ **do**
  | **if** $d_v = 2$ and $d_u = 2$ and $|\mathcal{N}_\mathcal{G}(u) \cap \mathcal{N}_\mathcal{G}(v)| = 1$ **then**
  |   |   $c_{uv} \leftarrow 0$
  | **end**
  | **else**
  |   |   $c_{uv} \leftarrow (d_u - 1) \cdot (d_v - 1)$
  | **end**
**end**
$c \leftarrow \sum_{uv} c_{uv}$
**foreach** $\langle u, v \rangle \in \mathcal{E}$ **do**
  |   $p_{uv} \leftarrow c_{uv}/c$
**end**

---

$\Delta\mathcal{G}_{\text{new}}$ and $\Delta\mathcal{G}_{\text{old}}$. Notably, these three parts have overlapping nodes but do not have overlapping edges, as shown in Figure A3.

To efficiently obtain $\Delta\mathcal{G}^-$, $\Delta\mathcal{G}^+_{\text{old}}$ and $\Delta\mathcal{G}^-_{\text{new}}$ from $\mathcal{G}_{\text{old}}$ and $\mathcal{G}_{\text{new}}$, we can first compute $\Delta\mathcal{G}^-$ by comparing $\mathcal{G}_{\text{new}}$ to $\mathcal{G}_{\text{old}}$ (or comparing $\boldsymbol{A}_{\text{new}}$ to $\boldsymbol{A}_{\text{old}}$). We can then calculate $\Delta\mathcal{V}^+_{\text{new}}$ consisting of the nodes in $\mathcal{V}_{\text{new}}$ but not in $\mathcal{V}_{\text{old}}$ by $\Delta\mathcal{V}^+_{\text{new}} = \mathcal{V}_{\text{new}} + \Delta\mathcal{V}^- - \mathcal{V}_{\text{old}}$. Next, we can calculate $\Delta\mathcal{E}^+_{\text{new}}$ by referring to $\boldsymbol{A}_{\text{new}}$. And, $\Delta\mathcal{E}^+_{\text{old}}$ can be derived by $\Delta\mathcal{E}^+_{\text{old}} = \mathcal{E}_{\text{new}} + \Delta\mathcal{E}^- - \mathcal{E}_{\text{old}} - \Delta\mathcal{E}^+_{\text{new}}$.

From Algorithms 2,3,4,6,7,5, we can see that the buffer $\mathcal{B}$ only store the of nodes (i.e., node IDs) of each subgraph instance. These IDs for subgraphs following the supporting graph patterns in 3-cycle shape represent all the 3-cycles in the graph, and the IDs for subgraphs following the supporting graph patterns in 4-cycle shape represent a number of uniformly sampled 4-cycles in the graph. Therefore, we are seeking for an efficient way to update the buffer from $\mathcal{B}_{\text{old}}$, the buffer computed by $\mathcal{G}_{\text{old}}$, to $\mathcal{B}_{\text{new}}$, the buffer computed by $\mathcal{G}_{\text{new}}$. From the above analysis, we are required to search and retrieve all the new subgraphs following $\Pi^+_{3-\text{cycle}}$ and uniformly sample the new subgraphs following $\Pi^+_{4-\text{cycle}}$. As the time complexity of sampling $n$ refuting subgraphs following $\Pi^-_{3-\text{cycle}}$ or $\Pi^-_{4-\text{cycle}}$ in $\mathcal{G}_{\text{new}}$ is $O(|\mathcal{V}_{\text{new}}| + |\mathcal{E}_{\text{new}}| + n)$ (as shown in Theorem 1), and thus it's yet efficient to directly re-compute these refuting subgraphs.

We summarize the pipeline of updating the buffer $\mathcal{B}$ in Algorithm 11 where $\lfloor \cdot \rfloor$ denotes the flooring function, and provide detailed descriptions as follows.

Firstly, we deal with "delete modifications" from $\mathcal{G}_{\text{old}}$ to $\mathcal{G}_{\text{new}}$ (i.e., $\Delta\mathcal{G}^-$) by removing all the instances that include the nodes in $\Delta\mathcal{V}^-$. Secondly, for new

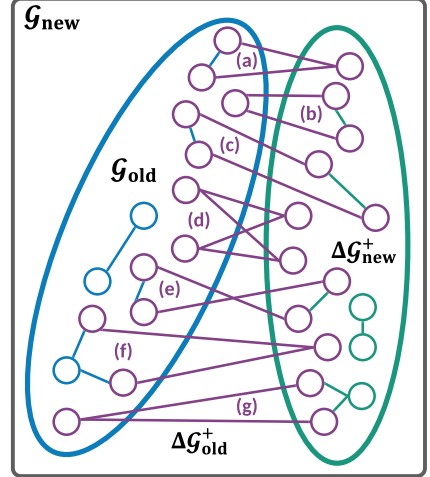

Figure A3: An illustrated example of dividing $\mathcal{G}_{\text{new}}$ (including the nodes and edges in the black box) into $\mathcal{G}_{\text{old}}$ (including the nodes and edges in the blue circle), $\Delta\mathcal{G}_{\text{new}}$ (including the nodes and edges in the green circle) and $\Delta\mathcal{G}_{\text{old}}$ (including the nodes and edges in purple color), when putting $\Delta\mathcal{G}^-$ aside. Note that there is overlapping nodes but no overlapping edges between $\mathcal{G}_{\text{old}}$ and $\Delta\mathcal{G}_{\text{old}}$, $\Delta\mathcal{G}_{\text{new}}$ and $\Delta\mathcal{G}_{\text{old}}$. (a)(b) denote the two types of 3-cycles and (c)-(g) denote five types of 4-cycles not within either $\mathcal{G}_{\text{old}}$ or $\Delta\mathcal{G}_{\text{new}}$.

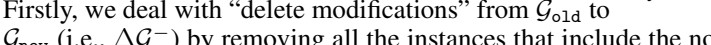

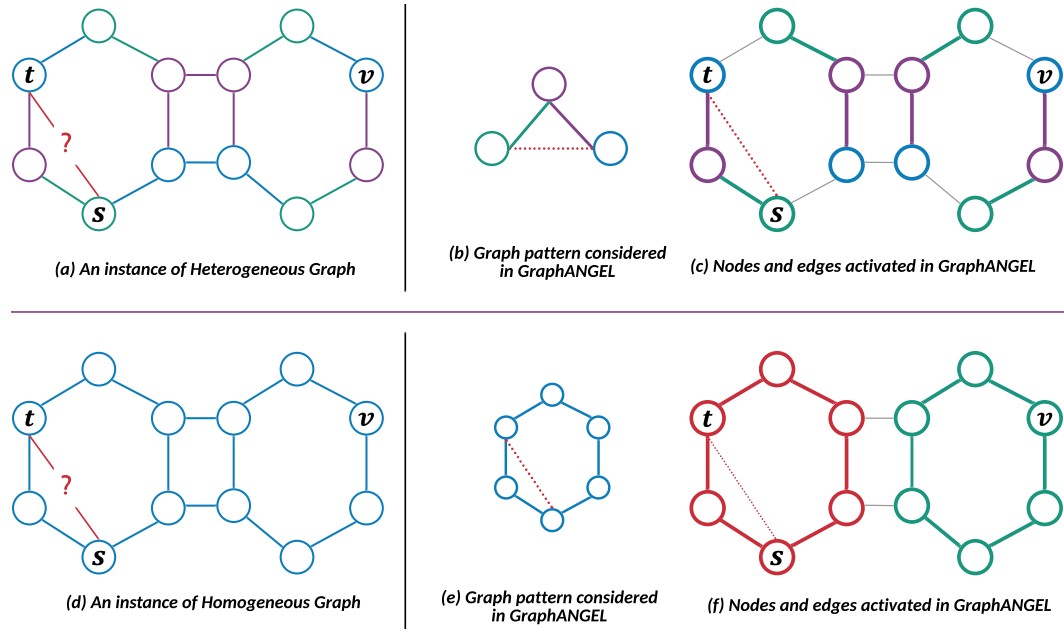

Figure A4: An illustrated example of how GraphANGEL works on both the heterogeneous graph shown in (a) and the homogeneous graph shown in (d), where the colors of the nodes and edges represent their types in the heterogeneous graph. For the heterogeneous graph, GraphANGEL can automatically find the graph pattern by scanning on the graph schema by given shapes (e.g., Pair, 3-cycle, 4-cycle). We show an example graph pattern in (b). Based on the graph and graph pattern, we show the nodes and edges activated in GraphANGEL in (c), where the nodes and edges in colors are activated and those in black are not activated. However, for the homogeneous graph, since there is no graph pattern based on the node and edge type, we design the graph pattern directly based on the shape (i.e., hexagon shape in (e)). We then show the nodes and edges activated in (f), where nodes and edges in red construct a subgraph following target pattern and the green ones compose a subgraph following analogy pattern.

subgraphs following $\Pi_{3-\text{cycle}}^{+}$, these 3-cycles might either in $\Delta\mathcal{G}_{\text{new}}^{+}$ or include the nodes and edges across $\mathcal{G}_{\text{old}}$ and $\Delta\mathcal{G}_{\text{new}}^{+}$. For the former one, we can directly apply Algorithm 2 or 3 in $\Delta\mathcal{G}_{\text{new}}^{+}$, while for the latter one, we need to consider two different cases, shown as (a) and (b) in Figure A3. For searching and retrieving subgraphs in (b) shape, it's practical to directly enumerate all the edges in $\Delta\mathcal{V}_{\text{new}}^{+}$, as $|\Delta\mathcal{V}_{\text{new}}^{+}|$ is expected to be small. However, $|\mathcal{V}_{\text{old}}|$ is large in the most time, and thus we develop Algorithm 9 to efficiently search and retrieve subgraphs in (a) shape, whose design is analogous to Algorithm 2.

Thirdly, for $\Pi_{4-\text{cycle}}^{+}$, similar to $\Pi_{3-\text{cycle}}^{+}$, there are also two classes of subgraphs. For the first one, we can uniformly sample a number of subgraphs following $\Pi_{4-\text{cycle}}^{+}$ by directly applying Algorithm 5 in $\Delta\mathcal{G}_{\text{new}}^{+}$. For the second one, we need to consider five different cases, shown as (c), (d), (e), (f), (g) in Figure A3. We can simultaneously deal with (c), (d), (e) by uniformly sampling edges in $\Delta\mathcal{V}_{\text{old}}^{+}$ where the sample rate of edges can be computed by using Algorithm 10. For (f) and (g), we first uniformly sample a 3-cycle in one of $\mathcal{G}_{\text{old}}$ and $\mathcal{G}_{\text{new}}^{+}$, which is similar to Algorithm 6, and then uniformly sample one node in the other one of $\mathcal{G}_{\text{old}}$ and $\mathcal{G}_{\text{new}}^{+}$. The sample rate in these cases can also obtain by using Algorithm 10.

For $\Pi_{3-\text{cycle}}^{-}$ and $\Pi_{4-\text{cycle}}^{-}$, considering Algorithm 7 and 5 only have the linear complexity, we directly re-sample the refuting cases of $\Pi_{3-\text{cycle}}^{-}$ and $\Pi_{4-\text{cycle}}^{-}$ by running these algorithms on $\mathcal{G}_{\text{new}}$.

**Time Complexity of Incremental Updating Algorithm.** As shown in Algorithm 11, there are mainly four parts: removing the instances including the nodes in $\Delta\mathcal{V}^{-}$; searching and retrieving the supporting cases of $\Pi_{3-\text{cycle}}^{+}$; sampling the the supporting cases of $\Pi_{4-\text{cycle}}^{+}$; sampling the refuting cases of $\Pi_{3-\text{cycle}}^{-}$ and $\Pi_{4-\text{cycle}}^{-}$. Let $|\mathcal{B}|$ denote the number of instances in the buffer $\mathcal{B}$.

The first part requires $O(|\mathcal{B}_{\text{old}}| \cdot |\Delta\mathcal{V}^{-}|)$ computations, the second part requires $O(|\Delta\mathcal{E}_{\text{new}}^{+}|^{\frac{3}{2}} + |\Delta\mathcal{E}_{\text{new}}^{+}| \cdot |\mathcal{V}_{\text{old}}| + |\mathcal{E}_{\text{old}}| \cdot \min(|\Delta\mathcal{V}_{\text{new}}^{+}|, \lceil\frac{|\mathcal{V}_{\text{old}}|^{\frac{1}{2}}}{|\Delta\mathcal{V}_{\text{new}}^{+}|^{\frac{1}{2}}}\rceil))$ computations, the third part requires $O(|\Delta\mathcal{E}_{\text{new}}^{+}|^{\frac{3}{2}} + \lceil\frac{|\Delta\mathcal{V}_{\text{new}}^{+}|^{4}}{|\mathcal{V}_{\text{old}}|^{4}}n_{4-\text{cycle}}\rceil + |\mathcal{V}_{\text{old}}| + |\mathcal{E}_{\text{old}}| + |\Delta\mathcal{V}_{\text{old}}| + |\Delta\mathcal{E}_{\text{old}}| + |\Delta\mathcal{V}_{\text{new}}| + |\Delta\mathcal{E}_{\text{new}}| + \lceil\frac{|\Delta\mathcal{E}_{\text{old}}^{+}|^{2}}{|\mathcal{E}_{\text{old}}|^{2}}n_{4-\text{cycle}}\rceil +$

Table A1: Main statistics of heterogeneous graphs.

| Datasets | Relations (A-B) | #A | #B | #A-B | Datasets | Relations (A-B) | #A | #B | #A-B |
|---|---|---|---|---|---|---|---|---|---|
| LastFM | User-Artist | 2,101 | 18,746 | 92,834 | Amazon | User-Item | 6,170 | 2,753 | 146,230 |
| | User-User | 2,101 | 2,101 | 25,434 | | Item-View | 2,753 | 3,857 | 5,694 |
| | User-Tag | 2,101 | 12,648 | 186,479 | | Item-Brand | 2,753 | 334 | 2,753 |
| | Artist-Tag | 18,746 | 12,648 | 186,479 | | User-User | 6,170 | 6,170 | 37,958 |
| Yelp | User-Business | 16,239 | 14,284 | 198,397 | Douban Book | User-Book | 13,024 | 22,347 | 598,420 |
| | User-User | 16,239 | 16,239 | 158,590 | | User-User | 13,024 | 13,024 | 169,150 |
| | User-City | 16,239 | 47 | 76,875 | | User-Group | 13,024 | 2,936 | 1,189,271 |
| | Business-City | 14,284 | 47 | 14,267 | | Book-Author | 22,347 | 10,805 | 21,907 |
| | Business-Genre | 14,284 | 511 | 40,009 | | Book-Publisher | 22,347 | 1,815 | 21,773 |

Table A2: Main statistics of knowledge graphs.

| Datasets | #Entities | #Relations | #Training | #Validation | #Test |
|---|---|---|---|---|---|
| FB15k-237 | 14,541 | 237 | 272,115 | 17,535 | 20,466 |
| WN18RR | 40,943 | 11 | 86,835 | 3,304 | 3,134 |

$[\frac{|\Delta\mathcal{E}_{\text{old}}^+|^2|\Delta\mathcal{E}_{\text{new}}^+|^2}{|\mathcal{E}_{\text{old}}|^4}n_{4-\text{cycle}}])$, the fourth part requires $O(|\mathcal{V}_{\text{new}}|+|\mathcal{E}_{\text{new}}|+[\frac{|\mathcal{E}_{\text{new}}|^3}{|\mathcal{E}_{\text{old}}|^3}n])$, where $[\cdot]$ denote the floor function. The overall complexity is to sum up the above computations. One can easily conclude that when the numbers of new nodes and edges are relatively much smaller than the numbers of old nodes and edges, Algorithm 11 is more efficient than re-running the subgraph retrieval algorithms on the new graph, which are the common cases in the real-world scenario.

## A3.3 DETAILED FORMULATION OF $\Phi$ IN GRAPHANGEL

As introduced in Section 3, we apply a neural network $\Phi(\cdot)$ over each subgraph. These subgraphs can be from either target subgraph set $\{\mathcal{S}_{p,k}^*\}_{p=1,k=1}^{P,K}$, supporting subgraph set $\{\mathcal{S}_{p,q}^+\}_{p=1,q=1}^{P,Q}$, or refuting subgraph set $\{\mathcal{S}_{p,q}^-\}_{p=1,q=1}^{P,Q}$, as shown in Eq. (1). Formally, $\Phi$ maps each subgraph $\mathcal{S}_{p,k}^*$, $\mathcal{S}_{p,q}^+$ or $\mathcal{S}_{p,q}^-$ to the corresponding graph-level representations $\boldsymbol{e}_{p,k}^*$, $\boldsymbol{e}_{p,q}^+$ or $\boldsymbol{e}_{p,q}^-$. In the implementation, we adopt single layer R-GCN (Schlichtkrull et al., 2018) followed by any readout function (e.g., Mean$(\cdot)$, Max$(\cdot)$) as $\Phi(\cdot)$.

Taking $\mathcal{S}_{p,k}^*$ as an instance, we use $\boldsymbol{h}_{v,p,k}$ to denote the embedding of node $v$ in the subgraph namely $v \in \mathcal{S}_{p,k}^*$, and $\mathcal{N}_{\mathcal{S}_{p,k}}^r(v)$ to denote the set of neighbor nodes of node $v$ in $\mathcal{S}_{p,k}$ regarding relation type $r$. We compute $\boldsymbol{h}_{v,p,k}$ by

$$\boldsymbol{h}_{v,p,k} = \sigma(\sum_{r\in\mathcal{R}}\sum_{u\in\mathcal{N}_{\mathcal{S}_{p,k}}^r(v)}\boldsymbol{W}_r\boldsymbol{x}_u + \boldsymbol{W}_0\boldsymbol{x}_v), \tag{11}$$

where $\boldsymbol{x}_.$ denote the input node features or learnable node embeddings, $\boldsymbol{W}_.$ denote trainable parameters, and $\sigma$ is non-linearity function such as ReLU. After updating the embedding vectors of all the nodes, we further compute $\boldsymbol{e}_{p,k}^*$ by $\boldsymbol{e}_{p,k}^* = \text{Mean}(\{\boldsymbol{h}_{v,p,k}|v\in\mathcal{S}_{p,k}^*\})$. Computations of $\boldsymbol{e}_{p,q}^+$ and $\boldsymbol{e}_{p,q}^-$ are similar.

Specifically, in the case of Pair shape, R-GCN reduces to an MLP, and thus $\Phi$ here generally would not be worse than what an MLP could achieve.

## A3.4 DETAILED FORMULATION OF $\Psi$ IN GRAPHANGEL

As stated in Section 3, we deploy a neural network $\Psi(\cdot)$ to measure the similarity $s^+$ between the set of subgraphs matching $\Pi_1^*, \Pi_2^*, \ldots, \Pi_P^*$ and the set of subgraphs matching $\Pi_1^+, \Pi_2^+, \ldots, \Pi_P^+$; and the similarity $s^-$ between the set of subgraphs matching $\Pi_1^*, \Pi_2^*, \ldots, \Pi_P^*$ and the set of subgraphs matching $\Pi_1^-, \Pi_2^-, \ldots, \Pi_P^-$.

Take the computation of $s^+$ as an example. We first reshape the subgraph embedding tensors $\boldsymbol{e}_{p,k}^*$ into matrices: $\boldsymbol{E}^* = \text{CONCAT}[\{\boldsymbol{e}_{p,k}^*, p = 1, \ldots, P; k = 1, \ldots, K\}] \in \mathbb{R}^{d\times(P\times K)}$, and similarly form $\boldsymbol{e}_{p,q}^+$ to $\boldsymbol{E}^+ \in \mathbb{R}^{d\times(P\times Q)}$, where $d$ is the hidden representation size. We then compute the affinity matrix as

$$\boldsymbol{A} = \tanh(\boldsymbol{E}^{*\top}\boldsymbol{W}_c\boldsymbol{E}^+) \in \mathbb{R}^{(P\times K)\times(P\times Q)}, \tag{12}$$

where $\boldsymbol{W}_c \in \mathbb{R}^{d\times d}$ are trainable weights. With the affinity matrix $\boldsymbol{A}$, one possible way of computing attention is to simply maximize out the affinity, namely $\boldsymbol{a}^+[j] = \text{Max}_i(\boldsymbol{A}_{i,j})$ and $\boldsymbol{a}^*[i] = \text{Max}_j(\boldsymbol{A}_{i,j})$.

Table A3: Main statistics of patterns in heterogeneous and knowledge graphs

| Patterns | LastFM | Amazon | Yelp | Douban Book | FB15k-237 | WN18RR |
|---|---|---|---|---|---|---|
| 3-cycle | 29K | 56K | 137K | 620K | 519K | 11,590K |
| 4-cycle | 28M | 52M | 81M | 62M | 51M | 2,623M |
| average coverage of 3-hop neighborhood | 78% | 72% | 78% | 79% | 78% | 68% |
| average coverage of 4-hop neighborhood | 98% | 92% | 97% | 99% | 99% | 90% |

Instead of using the max activation, we follow (Lu et al., 2016), considering this affinity matrix as a feature and learning the attention map via following:

$$\boldsymbol{H}^* = \boldsymbol{W}_* \boldsymbol{E}^*, \ \boldsymbol{H}^+ = \boldsymbol{W}_+ \boldsymbol{E}^+ \tag{13}$$

which are used to compute the attention map among $\boldsymbol{E}^*$ and $\boldsymbol{E}^+$ through

$$\boldsymbol{a}^* = \texttt{softmax}(\boldsymbol{w}_*^\top \tanh(\boldsymbol{H}^* + \boldsymbol{H}^+ \boldsymbol{A})), \ \boldsymbol{a}^+ = \texttt{softmax}(\boldsymbol{w}_+^\top \tanh(\boldsymbol{H}^+ + \boldsymbol{H}^* \boldsymbol{A})), \tag{14}$$

where $\boldsymbol{W}_* \in \mathbb{R}^{d \times d}$ and $\boldsymbol{W}_+ \in \mathbb{R}^{d \times d}$, $\boldsymbol{w}_* \in \mathbb{R}^d$ and $\boldsymbol{w}_+ \in \mathbb{R}^d$ are the weight parameters. $\boldsymbol{a}^* \in \mathbb{R}^{(P \times K)}$ and $\boldsymbol{a}^+ \in \mathbb{R}^{(P \times Q)}$. We finally obtain the set embedding via

$$\boldsymbol{h}^+ = \boldsymbol{E}^+ \boldsymbol{a}^{+\top}, \ \boldsymbol{h}^* = \boldsymbol{E}^* \boldsymbol{a}^{*\top}, \tag{15}$$

which is the weighted average of the subgraph embeddings in matrix $\boldsymbol{E}^+$ and $\boldsymbol{E}^*$. The similarity of supporting patterns is $s^+ = {\boldsymbol{h}^+}^\top \boldsymbol{h}^*$. Computation of $s^-$ is similar.

## A4 FURTHER ANALYSIS OF GRAPHANGEL

### A4.1 LIMITATIONS OF GRAPHANGEL

In order to further analyse the limitations of GraphANGEL, we first compare our framework with exsiting GNN methods. As shown in Figure A2, we provide an illustrated example of nodes and edges activated and not activated in various GNN models, including GCN (Kipf & Welling, 2016), GraphSAGE (Hamilton et al., 2017), GAT (Veličković et al., 2017), SEAL (Zhang & Chen, 2018), PPNP and APPNP (Approximate PPNP) (Klicpera et al., 2018), and our model GraphANGEL. With a limited number of GNN layers, one can easily observe that these existing approaches only involve all context within the "neighborhood". Instead, GraphANGEL leverages the graph structure by searching and retrieving the subgraphs in the whole graph, which not only removes the limitation of neighborhood constriction but also selects pattern (i.e., logic) related information. Figure A2 depicts an illustrated example of the edges activated in GraphANGEL comparing to other GNNs, where we can see that our method can be regarded as to restrict the message passing within the subgraphs satisfying certain pre-defined patterns, (e.g., $\Pi_{\texttt{3-cycle}}$, $\Pi_{\texttt{4-cycle}}$).

Although from the analysis above, we can see that GraphANGEL can succeed in those cases where existing GNN models fail; however, leveraging logically-induced graph patterns can also be a limitation of GraphANGEL, especially when it is hard to find or design such graph patterns. For example, if the two nodes for the link prediction are far away from each other, the graph patterns considered in our experiments may be not able to cover the cases, and it is also hard to define what a graph pattern should be, which is previously pointed out in Section 3. We argue that such cases are rare in real-world graphs as they are usually scale-free, meaning that the topological distance is usually small. The topological distance of graphs are well measured by Wiener index, and detailed theoretical description and empirical study can be found in (Dobrynin et al., 2001). Due to the high computation costs for shortest path algorithms (Floyd, 1962; Gallo & Pallottino, 1988), in this paper, instead, we calculate the fraction of 3-hop and 4-hop neighborhoods over all the nodes for every node in each dataset, average them, and report the results in Table A3, which shows that 3-hop and 4-hop neighborhoods can cover almost all the nodes in the graph. Since there is no connectivity requirement between the source and target nodes in the target pattern as shown in Table 1, it is unlikely to fail to find at least one subgraph satisfying the target patterns.

The other limitation pointed out in Section 3 is the time complexity of Algorithm 1 and the storage complexity of Algorithm 8. We provide the complexity analysis in Appendix A3 and report the overall time of training and inference in Appendix A6.7. We argue that there is a recent trend in building language models (Khandelwal et al., 2019; He et al., 2021; Khandelwal et al., 2020; Zheng et al., 2021) by searching and storing the representation vectors of neighbor nodes. Comparing to these methods, our subgraph retrieval only search, retrieve and further store the node IDs, which

Table A4: Comparable results with baselines, where there are 20% least frequent relations. See Appendix A6.1 for the full version. The numbers in brackets show the descent degree comparing to Table A4.

| Models | FB15k-237 | | | WN18RR | | |
|---|---|---|---|---|---|---|
| | Hit@1 | Hit@3 | Hit@10 | Hit@1 | Hit@3 | Hit@10 |
| pLogicNet | 0.209(11.8%↓) | 0.342(6.81%↓) | 0.500(4.58%↓) | 0.341(14.3%↓) | 0.406(8.97%↓) | 0.491(8.57%↓) |
| TransE | 0.197(14.0%↓) | 0.339(6.61%↓) | 0.494(5.18%↓) | 0.123(8.89%↓) | 0.367(8.48%↓) | 0.487(8.29%↓) |
| ConvE | 0.207(12.7%↓) | 0.324(8.99%↓) | 0.478(4.59%↓) | 0.364(9.00%↓) | 0.391(11.1%↓) | 0.479(7.88%↓) |
| ComplEx | 0.140(11.4%↓) | 0.261(5.09%↓) | 0.409(4.44%↓) | 0.375(8.54%↓) | 0.428(6.96%↓) | 0.475(6.86%↓) |
| MLN | 0.051(23.9%↓) | 0.077(25.2%↓) | 0.143(10.6%↓) | 0.166(13.1%↓) | 0.285(11.5%↓) | 0.333(7.76%↓) |
| RotatE | 0.211(12.3%↓) | 0.351(6.34%↓) | 0.505(5.32%↓) | 0.386(9.83%↓) | 0.445(9.57%↓) | 0.529(7.38%↓) |
| RNNLogic | 0.219(13.2%↓) | 0.333(12.5%↓) | 0.499(5.76%↓) | 0.407(8.61%↓) | 0.444(10.7%↓) | 0.511(8.42%↓) |
| ComplEx-N3 | 0.242(11.1%↓) | 0.361(9.85%↓) | 0.534(4.85%↓) | 0.407(**7.49%**↓) | 0.456(8.74%↓) | 0.539(7.15%↓) |
| GraIL | 0.197(11.5%↓) | 0.315(12.8%↓) | 0.484(6.84%↓) | 0.307(12.7%↓) | 0.387(11.6%↓) | 0.458(8.52%↓) |
| QuatE | 0.219(11.7%↓) | 0.355(7.14%↓) | 0.521(5.30%↓) | 0.400(8.75%↓) | 0.462(9.06%↓) | 0.535(8.06%↓)) |
| GraphANGEL$_{3-cycle}$ | 0.243(9.87%↓) | 0.384(3.53%↓) | 0.539(**3.80%**↓) | 0.418(9.78%↓) | 0.465(6.42%↓) | 0.549(6.93%↓) |
| GraphANGEL$_{4-cycle}$ | 0.215(9.96%↓) | 0.366(4.04%↓) | 0.526(3.94%↓) | 0.421(9.37%↓) | 0.467(6.92%↓) | 0.549(**6.47%**↓) |
| GraphANGEL | **0.248**(9.62%↓) | **0.394**(3.27%↓) | **0.541**(3.84%↓) | **0.429**(8.74%↓) | **0.481**(6.21%↓) | **0.557**(6.75%↓) |

Table A5: Comparable results with baselines, where there are 20% least frequent relations. Here we report the results in term of MR, MRR, Hit@K (K=1,3,10).

| Models | FB15k-237 | | | | | WN18RR | | | | |
|---|---|---|---|---|---|---|---|---|---|---|
| | MR | MRR | Hit@1 | Hit@3 | Hit@10 | MR | MRR | Hit@1 | Hit@3 | Hit@10 |
| pLogicNet | 192 | 0.284 | 0.209 | 0.342 | 0.500 | 3891 | 0.378 | 0.341 | 0.406 | 0.491 |
| TransE | 198 | 0.296 | 0.197 | 0.339 | 0.494 | 3713 | 0.203 | 0.123 | 0.367 | 0.487 |
| ConvE | 266 | 0.295 | 0.207 | 0.324 | 0.478 | 4561 | 0.391 | 0.364 | 0.391 | 0.479 |
| ComplEx | 367 | 0.226 | 0.140 | 0.261 | 0.409 | 5709 | 0.402 | 0.375 | 0.428 | 0.475 |
| MLN | 2235 | 0.087 | 0.051 | 0.077 | 0.143 | 13056 | 0.225 | 0.166 | 0.285 | 0.333 |
| RotatE | 200 | 0.302 | 0.211 | 0.351 | 0.505 | 3630 | 0.435 | 0.386 | 0.445 | 0.529 |
| RNNLogic | 263 | 0.302 | 0.219 | 0.333 | 0.499 | 4659 | 0.438 | 0.407 | 0.444 | 0.511 |
| ComplEx-N3 | 177 | 0.334 | 0.242 | 0.361 | 0.534 | 3481 | 0.448 | 0.407 | 0.456 | 0.539 |
| GraIL | 230 | 0.281 | 0.197 | 0.315 | 0.484 | 3949 | 0.362 | 0.307 | 0.387 | 0.458 |
| QuatE | **101** | 0.296 | 0.219 | 0.355 | 0.521 | **2705** | 0.421 | 0.400 | 0.462 | 0.535 |
| GraphANGEL$_{3-cycle}$ | 174 | 0.324 | 0.243 | 0.384 | 0.539 | 3139 | 0.445 | 0.418 | 0.465 | 0.549 |
| GraphANGEL$_{4-cycle}$ | 181 | 0.317 | 0.215 | 0.366 | 0.526 | 3169 | 0.450 | 0.421 | 0.467 | 0.549 |
| GraphANGEL | 167 | **0.336** | **0.248** | **0.394** | **0.541** | 3108 | **0.458** | **0.429** | **0.481** | **0.557** |

are relatively much more retrieval-efficient and storage-efficient than searching and storing the representation vectors of nodes.

Besides the above discussions about the limitations pointed out in Section 3, we also provide the following discussions regarding the assumptions proposed in Section 2.

One of the assumptions introduced in Section 2 is that existing relations is enough to cover unseen relations by logical inference, which motivates us to not explicitly construct embeddings for the relation to predict. However, this assumption also disables GraphANGEL to consider the correlations among unseen relations. In other words, it's not possible for current GraphANGEL to incorporate unseen relations as a part of the graph after actually observing them during inference. One possible solution is to incorporate our model with EmbedRule (Yang et al., 2014) to first generate explicit embeddings for each unseen relations by EmbedRule, and then apply GraphANGEL to generate the final predictions. Considering that both EmbedRule and GraphANGEL have strong generalizability to unseen relations, it is hard to say which one plays a more important role in this case. Hence, we leave it as future work.

The other one of the assumptions is to bridge the graph patterns with logical rules. Therefore, GraphANGEL is capable of distinguishing different semantics of the same relation as long as the local subgraphs, hence contexts, are different.

## A4.2 EXTENSIONS OF GRAPHANGEL TO HOMOGENEOUS GRAPH

One may consider that heterogeneous graph may be another potential limitation of GraphANGEL. However, we note that extending GraphANGEL to homogeneous graphs is possible with graph patterns that are not type-sensitive, i.e., not containing type-specific indicator operators $\text{Edge}_r(x, y)$. In other words, the graph patterns will be purely based on topology. For example, we can define 6-cycle for the graph in Figure A4(d).

Table A6: Result comparisions with baselines on generalization setting by randomly removing 5% relations. Here we report the results in term of MR, MRR, Hit@K (K=1,3,10).

| Models | FB15k-237 | | | | | WN18RR | | | | |
|---|---|---|---|---|---|---|---|---|---|---|
| | MR | MRR | Hit@1 | Hit@3 | Hit@10 | MR | MRR | Hit@1 | Hit@3 | Hit@10 |
| pLogicNet* | 185 | 0.309 | 0.220 | 0.341 | 0.487 | 3646 | 0.410 | 0.370 | 0.414 | 0.500 |
| TransE* | 191 | 0.306 | 0.215 | 0.338 | 0.489 | 3614 | 0.209 | 0.230 | 0.378 | 0.499 |
| ConvE* | 268 | 0.292 | 0.213 | 0.320 | 0.450 | 4605 | 0.387 | 0.361 | 0.396 | 0.469 |
| ComplEx* | 366 | 0.227 | 0.145 | 0.252 | 0.393 | 5681 | 0.404 | 0.371 | 0.423 | 0.472 |
| MLN* | 2277 | 0.083 | 0.056 | 0.087 | 0.135 | 13281 | 0.220 | 0.168 | 0.273 | 0.306 |
| RotatE* | 189 | 0.317 | 0.227 | 0.353 | 0.500 | 3624 | 0.439 | 0.398 | 0.458 | 0.531 |
| RNNLogic* | 248 | 0.319 | 0.235 | 0.353 | 0.493 | 4865 | 0.456 | 0.419 | 0.472 | 0.523 |
| ComplEx-N3* | 167 | 0.350 | 0.256 | 0.380 | 0.527 | 3625 | 0.466 | 0.418 | 0.475 | 0.551 |
| GraIL* | 221 | 0.296 | 0.205 | 0.332 | 0.478 | 3817 | 0.372 | 0.326 | 0.402 | 0.459 |
| QuatE* | **93** | 0.327 | 0.233 | 0.360 | 0.520 | **2453** | 0.461 | 0.413 | 0.480 | 0.547 |
| GraphANGEL$_{3-cycle}$ | 166 | 0.349 | 0.259 | 0.379 | 0.532 | 3061 | 0.468 | 0.441 | 0.472 | 0.566 |
| GraphANGEL$_{4-cycle}$ | 173 | 0.335 | 0.229 | 0.365 | 0.525 | 3049 | 0.472 | 0.446 | 0.480 | 0.558 |
| GraphANGEL | 158 | **0.357** | **0.263** | **0.388** | **0.537** | 2948 | **0.482** | **0.450** | **0.492** | **0.572** |

Table A7: Result comparisions with baselines on generalization setting by randomly removing 10% relations. Here we report the results in term of MR, MRR, Hit@K (K=1,3,10).

| Models | FB15k-237 | | | | | WN18RR | | | | |
|---|---|---|---|---|---|---|---|---|---|---|
| | MR | MRR | Hit@1 | Hit@3 | Hit@10 | MR | MRR | Hit@1 | Hit@3 | Hit@10 |
| pLogicNet* | 204 | 0.272 | 0.194 | 0.298 | 0.429 | 4021 | 0.361 | 0.326 | 0.365 | 0.440 |
| TransE* | 215 | 0.264 | 0.185 | 0.294 | 0.422 | 4057 | 0.180 | 0.109 | 0.324 | 0.430 |
| ConvE* | 290 | 0.263 | 0.191 | 0.285 | 0.405 | 4982 | 0.348 | 0.324 | 0.356 | 0.421 |
| ComplEx* | 393 | 0.207 | 0.132 | 0.229 | 0.359 | 6102 | 0.369 | 0.3444 | 0.386 | 0.428 |
| MLN* | 2475 | 0.007 | 0.05 | 0.077 | 0.120 | 14436 | 0.194 | 0.143 | 0.241 | 0.270 |
| RotatE* | 207 | 0.279 | 0.199 | 0.307 | 0.437 | 3941 | 0.391 | 0.353 | 0.403 | 0.470 |
| RNNLogic* | 275 | 0.280 | 0.206 | 0.310 | 0.431 | 5469 | 0.395 | 0.364 | 0.408 | 0.452 |
| ComplEx-N3* | 187 | 0.304 | 0.224 | 0.331 | 0.462 | 4084 | 0.404 | 0.362 | 0.412 | 0.479 |
| GraIL* | 243 | 0.261 | 0.181 | 0.293 | 0.423 | 4208 | 0.327 | 0.287 | 0.359 | 0.410 |
| QuatE* | **128** | 0.286 | 0.203 | 0.313 | 0.452 | **2731** | 0.402 | 0.360 | 0.417 | 0.477 |
| GraphANGEL$_{3-cycle}$ | 184 | 0.306 | 0.226 | 0.333 | 0.466 | 3409 | 0.412 | 0.386 | 0.414 | 0.490 |
| GraphANGEL$_{4-cycle}$ | 192 | 0.293 | 0.200 | 0.319 | 0.460 | 3392 | 0.414 | 0.388 | 0.419 | 0.489 |
| GraphANGEL | 176 | **0.313** | **0.232** | **0.343** | **0.475** | **3259** | **0.423** | **0.398** | **0.434** | **0.504** |

Following the analysis above, although nodes $t$ and $v$ in both Figure A4(a) and (d) have similar neighborhood structures, our GraphANGEL is able to project these two source-target node pairs (i.e., $\langle s, r, t \rangle$ and $\langle v, r, t \rangle$) into different embeddings on both heterogeneous and homogeneous graphs.

### A4.3 Extensions of GraphANGEL to Complex Logic

Another potential limitation of GraphANGEL is that the graph searching stage limits the model's capability of exploring more complex graph patterns. We admit that the graph pattern involved in our paper is simple and rudimentary, but we argue that efficient finding arbitrary graph patterns including those complex ones should be mainly investigated by graph database and pattern matching literature, which is definitely not the focus of our paper. Our key idea is general since one can pre-define any pattern to search and retrieve subgraphs.

Moreover, the following corollary shows that the representations of complex logical formulas can be derived from those of simpler ones. This, to some extent, provides the theoretical support of only involving simple graph patterns (e.g., Pair, 3-cycle, 4-cycle) in the implementation.

**Lemma 1.** *(Barceló et al., 2019)* (Expressive Power on Logic Expressions of Nodes) *Each First-Order Classifier with 2 variables (FOC$_2$) can be captured by Aggregation and Combination GNN formulated as*

$$\boldsymbol{x}_i^{(l)} = \texttt{COM}(\boldsymbol{x}_i^{(l-1)}, \texttt{AGG}(\{\boldsymbol{x}_j^{(l-1)} | j \in \mathcal{N}_{\mathcal{G}}(i)\})), \ \forall i \in \mathcal{V}, \quad (16)$$

*where $\texttt{AGG}(\mathcal{X}) = \sum_{\boldsymbol{x} \in \mathcal{X}} \boldsymbol{x}$ and $\texttt{COM}(\boldsymbol{x}, \boldsymbol{y}) = \sigma(\boldsymbol{x}\boldsymbol{C} + \boldsymbol{y}\boldsymbol{A} + \boldsymbol{b})$.*

**Corollary 1.** (Expressive Power on Logic Expressions of Subgraphs) *Let $\boldsymbol{e}_x = \{e_1, \ldots, e_P\} \in \mathbb{R}^P$ be the representation vector of subgraph $\mathcal{S}_x$, where $e_p = 1$ if subgraph $\mathcal{S}_x$ matches graph pattern $\Pi_p^x$ and $e_p = 0$ otherwise, and $\Pi_1^x, \cdots, \Pi_P^x$ are sub-formulas of $\Pi^x$, in the order where if $\Pi_k^x$ is a sub-formula of $\Pi_l^x$ then $k \leq l$. Suppose that the representation vectors $\boldsymbol{e}_x$ and $\boldsymbol{e}_y$ of subgraph $\mathcal{S}_x$ and $\mathcal{S}_y$ matching pattern $\Pi^x$ and $\Pi^y$ respectively, are given. If $\Pi^z$ takes either of the following forms:*

- $\Pi^z = \neg \Pi^x$

Table A8: Result comparisions with baselines on generalization setting by randomly removing 15% relations. Here we report the results in term of MR, MRR, Hit@K (K=1,3,10).

| Models | FB15k-237 | | | | | WN18RR | | | | |
|---|---|---|---|---|---|---|---|---|---|---|
| | MR | MRR | Hit@1 | Hit@3 | Hit@10 | MR | MRR | Hit@1 | Hit@3 | Hit@10 |
| pLogicNet* | 230 | 0.222 | 0.158 | 0.245 | 0.351 | 4532 | 0.295 | 0.266 | 0.298 | 0.359 |
| TransE* | 249 | 0.202 | 0.142 | 0.224 | 0.323 | 4705 | 0.138 | 0.083 | 0.248 | 0.329 |
| ConvE* | 324 | 0.217 | 0.158 | 0.236 | 0.335 | 5568 | 0.288 | 0.268 | 0.294 | 0.348 |
| ComplEx* | 444 | 0.170 | 0.109 | 0.189 | 0.295 | 6891 | 0.303 | 0.282 | 0.317 | 0.351 |
| MLN* | 2633 | 0.065 | 0.044 | 0.229 | 0.107 | 15360 | 0.173 | 0.127 | 0.215 | 0.241 |
| RotatE* | 237 | 0.221 | 0.160 | 0.245 | 0.352 | 4482 | 0.319 | 0.287 | 0.325 | 0.376 |
| RNNLogic* | 311 | 0.225 | 0.165 | 0.249 | 0.350 | 6221 | 0.313 | 0.292 | 0.327 | 0.368 |
| ComplEx-N3* | 210 | 0.252 | 0.185 | 0.268 | 0.377 | 4557 | 0.334 | 0.297 | 0.337 | 0.392 |
| GraIL* | 275 | 0.213 | 0.147 | 0.239 | 0.347 | 4728 | 0.267 | 0.234 | 0.291 | 0.334 |
| QuatE* | **120** | 0.226 | 0.169 | 0.252 | 0.358 | **3055** | 0.312 | 0.289 | 0.330 | 0.361 |
| GraphANGEL$_{3-cycle}$ | 210 | 0.250 | 0.185 | 0.273 | 0.385 | 3823 | 0.340 | 0.319 | 0.343 | 0.407 |
| GraphANGEL$_{4-cycle}$ | 216 | 0.243 | 0.165 | 0.264 | 0.381 | 3797 | 0.344 | 0.323 | 0.349 | 0.407 |
| GraphANGEL | 198 | **0.259** | **0.192** | **0.289** | **0.391** | 3684 | **0.351** | **0.324** | **0.354** | **0.415** |

Table A9: Result comparisions with baselines on generalization setting by randomly removing 20% relations. Here we report the results in term of MR, MRR, Hit@K (K=1,3,10).

| Models | FB15k-237 | | | | | WN18RR | | | | |
|---|---|---|---|---|---|---|---|---|---|---|
| | MR | MRR | Hit@1 | Hit@3 | Hit@10 | MR | MRR | Hit@1 | Hit@3 | Hit@10 |
| pLogicNet* | 261 | 0.162 | 0.112 | 0.179 | 0.257 | 5146 | 0.216 | 0.141 | 0.222 | 0.267 |
| TransE* | 280 | 0.146 | 0.101 | 0.163 | 0.246 | 5285 | 0.100 | 0.072 | 0.200 | 0.260 |
| ConvE* | 366 | 0.162 | 0.104 | 0.178 | 0.247 | 6280 | 0.215 | 0.201 | 0.223 | 0.268 |
| ComplEx* | 501 | 0.128 | 0.078 | 0.142 | 0.226 | 7786 | 0.228 | 0.214 | 0.236 | 0.267 |
| MLN* | 3029 | 0.046 | 0.031 | 0.049 | 0.070 | 17669 | 0.121 | 0.092 | 0.154 | 0.178 |
| RotatE* | 269 | 0.176 | 0.121 | 0.187 | 0.271 | 4849 | 0.218 | 0.238 | 0.260 | 0.296 |
| RNNLogic* | 349 | 0.151 | 0.124 | 0.172 | 0.240 | 6728 | 0.248 | 0.244 | 0.260 | 0.281 |
| ComplEx-N3* | 232 | 0.208 | 0.142 | 0.208 | 0.289 | 5064 | 0.259 | 0.250 | 0.269 | 0.311 |
| GraIL* | 304 | 0.173 | 0.125 | 0.185 | 0.263 | 5315 | 0.212 | 0.195 | 0.222 | 0.267 |
| QuatE* | **137** | 0.173 | 0.127 | 0.190 | 0.282 | **3702** | 0.251 | 0.248 | 0.255 | 0.308 |
| GraphANGEL$_{3-cycle}$ | 223 | 0.225 | 0.168 | 0.230 | 0.333 | 4098 | 0.297 | 0.277 | 0.291 | 0.329 |
| GraphANGEL$_{4-cycle}$ | 227 | 0.206 | 0.147 | 0.222 | 0.328 | 4146 | 0.297 | 0.278 | 0.291 | 0.326 |
| GraphANGEL | 208 | **0.227** | **0.173** | **0.238** | **0.337** | 3956 | **0.305** | **0.284** | **0.299** | **0.334** |

- $\Pi^z = \Pi^x \wedge \Pi^y$
- $\Pi^z = \Pi^x \vee \Pi^y$

*Then there exists $\boldsymbol{W}^x \in \mathbb{R}^{P \times P}$, $\boldsymbol{W}^y \in \mathbb{R}^{P \times P}$, and $\boldsymbol{b} \in \mathbb{R}^P$, such that*

$$\boldsymbol{e}_z = \min(\max(\boldsymbol{W}^x \boldsymbol{e}_x + \boldsymbol{W}^y \boldsymbol{e}_y + \boldsymbol{b}, 0), 1)$$

*is a representation vector of a subgraph $\mathcal{S}_z$ matching $\Pi^z$.*

*Proof.* The entries of the $p$-th columns of $\boldsymbol{W}^x$, $\boldsymbol{W}^y$, and $\boldsymbol{b}$ depend on the formulas of $\Pi^{x,y}$ as follows.

- if $\Pi_p^{x,x} = \Pi_i^x$, then $\boldsymbol{W}_{ip}^x = 1$.
- if $\Pi_p^{x,y} = \Pi_i^x \wedge \Pi_j^y$, then $\boldsymbol{W}_{ip}^x = 1$, $\boldsymbol{W}_{jp}^x = 1$, and $\boldsymbol{b}_p = -1$.
- if $\Pi_p^{x,y} = \Pi_i^x \vee \Pi_j^y$, then $\boldsymbol{W}_{ip}^x = 1$, $\boldsymbol{W}_{jp}^x = 1$, and $\boldsymbol{b}_p = 0$.
- if $\Pi_p^{x,x} = \neg\Pi_i^x$, then $\boldsymbol{W}_{ip}^x = -1$ and $\boldsymbol{b}_p = 1$.

All other values in the $p$-th columns of $\boldsymbol{W}^x$, $\boldsymbol{W}^y$, and $\boldsymbol{b}$ are 0. Our proof follows the proof of Lemma 1 in (Barceló et al., 2019). The only differences are: (1) as the input of the basic logic formula in (Barceló et al., 2019) is nodes while in our paper is subgraph, thus the basic logic in (Barceló et al., 2019) is the color of nodes (i.e., Col($\cdot$)), but in our setting is the graph pattern for subgraphs; (2) we only involve the basic logical operators $\wedge$, $\vee$ and $\neg$ in our paper. $\square$

Besides the prior literature (Barceló et al., 2019) studying the theoretical expressive power of GNNs, our work is also kind of related to a thread of works (Minervini et al., 2020a;b; Weber et al., 2019) on Neural Theorem Provers, where they first assume template rules and then find which relations can satisfy them. In contrast, our work bridges the connection between logical expression and graph patterns and use the subgraphs to represent the rules. This intuition enables our model empowered by the expressive power of GNNs, and also allows our work to identify new relation types.

Table A10: Result comparisons with baselines on generalization setting by randomly adding 5% relations. Here we report the results in terms of MR, MRR, Hit@K (K=1,3,10).

| Models | FB15k-237 | | | | | WN18RR | | | | |
|---|---|---|---|---|---|---|---|---|---|---|
| | MR | MRR | Hit@1 | Hit@3 | Hit@10 | MR | MRR | Hit@1 | Hit@3 | Hit@10 |
| pLogicNet* | 187 | 0.305 | 0.218 | 0.337 | 0.482 | 3680 | 0.405 | 0.366 | 0.410 | 0.494 |
| TransE* | 197 | 0.296 | 0.208 | 0.330 | 0.474 | 3716 | 0.202 | 0.122 | 0.364 | 0.483 |
| ConvE* | 275 | 0.282 | 0.206 | 0.309 | 0.435 | 4731 | 0.374 | 0.348 | 0.382 | 0.452 |
| ComplEx* | 366 | 0.227 | 0.145 | 0.253 | 0.393 | 5681 | 0.404 | 0.377 | 0.423 | 0.469 |
| MLN* | 2277 | 0.083 | 0.056 | 0.087 | 0.136 | 13281 | 0.220 | 0.162 | 0.273 | 0.306 |
| RotatE* | 191 | 0.311 | 0.222 | 0.345 | 0.492 | 3590 | 0.438 | 0.398 | 0.452 | 0.526 |
| RNNLogic* | 252 | 0.315 | 0.231 | 0.347 | 0.484 | 5019 | 0.442 | 0.411 | 0.457 | 0.513 |
| ComplEx-N3* | 171 | 0.343 | 0.253 | 0.370 | 0.516 | 3728 | 0.369 | 0.323 | 0.405 | 0.462 |
| GraIL* | 223 | 0.293 | 0.213 | 0.332 | 0.475 | 3842 | 0.368 | 0.323 | 0.399 | 0.457 |
| QuatE* | **95** | 0.323 | 0.231 | 0.355 | 0.515 | **2481** | 0.458 | 0.411 | 0.473 | 0.543 |
| GraphANGEL$_{3-cycle}$ | 170 | 0.340 | 0.251 | 0.373 | 0.525 | 3104 | 0.458 | 0.434 | 0.465 | 0.555 |
| GraphANGEL$_{4-cycle}$ | 176 | 0.329 | 0.225 | 0.354 | 0.514 | 3109 | 0.461 | 0.434 | 0.470 | 0.552 |
| GraphANGEL | 161 | **0.349** | **0.258** | **0.382** | **0.527** | 3004 | **0.471** | **0.440** | **0.480** | **0.558** |

Table A11: Result comparisons with different GNNs as subgraph embedding methods (i.e., $\Phi$) on heterogeneous recommendation task.

| Models | LastFM | | | Yelp | | | Amazon | | | Douban Book | | |
|---|---|---|---|---|---|---|---|---|---|---|---|---|
| | AUC | ACC | F1 | AUC | ACC | F1 | AUC | ACC | F1 | AUC | ACC | F1 |
| GraphANGEL$_{GCN}^{*1}$ | 0.860 | 0.832 | 0.825 | 0.918 | 0.846 | 0.835 | 0.843 | 0.758 | 0.745 | 0.917 | 0.843 | 0.831 |
| GraphANGEL$_{GAT}^{*1}$ | 0.873 | 0.843 | 0.830 | 0.927 | 0.855 | 0.843 | 0.854 | 0.767 | 0.752 | 0.923 | 0.849 | 0.838 |
| GraphANGEL$_{R-GCN}^{*1}$ | 0.900 | 0.861 | 0.859 | **0.934** | 0.870 | **0.858** | 0.870 | 0.781 | 0.781 | **0.941** | 0.864 | **0.859** |
| GraphANGEL$_{R-GAT}^{*1}$ | 0.906 | **0.864** | 0.861 | 0.933 | 0.870 | 0.856 | **0.874** | 0.782 | 0.783 | 0.940 | **0.865** | 0.858 |
| GraphANGEL$_{R-GCN}^{*2}$ | 0.901 | 0.860 | **0.862** | **0.934** | 0.871 | 0.857 | 0.873 | **0.784** | 0.783 | 0.937 | 0.860 | 0.857 |
| GraphANGEL$_{R-GAT}^{*2}$ | **0.907** | 0.861 | **0.862** | 0.933 | **0.871** | 0.854 | **0.874** | 0.784 | **0.785** | 0.940 | 0.863 | **0.859** |

## A5 DETAILED EXPERIMENTAL SETTINGS

### A5.1 DATASET STATISTICS

The statistics of the datasets of heterogeneous graph based recommendation and knowledge graph completion used in our paper are displayed in Table A1 and Table A2, respectively. We also report the statistic of the patterns used in the datasets in Table A3, which can be regarded as the supporting evidence of using these patterns, since it is unlikely to fail to find at least one subgraph satisfying these patterns.

### A5.2 BASELINES

For the baseline methods, we employ these baseline methods based on Deep Graph Library (DGL) (Wang et al., 2019a) following their original setting or directly use their original implementations: (1) HetGNN (Zhang et al., 2019a) is a unified framework that jointly considers heterogeneous structured information as well as heterogeneous content information. (2) HAN (Wang et al., 2019b) is a hierarchical attention mechanism designed on heterogeneous graph to capture node-level and semantic-level information. (3) TAHIN (Bi et al., 2020) is a cross domain model from both source and target domain, which is then incorporated with three-level attention aggregations to get node representation. (4) HGT (Hu et al., 2020) is a heterogeneous graph transformer architecture that characterizes the heterogeneous attention over each relation. (5) R-GCN (Schlichtkrull et al., 2018) combines graph neural networks with factorization models for link prediction. (6) pLogicNet (Qu & Tang, 2019) is an EM-based algorithm that simultaneously trains a rule generator as well as reasoning predictor with logic rules. (7) TransE (Bordes et al., 2013) is a translation-based relation prediction model. (8) ConvE (Dettmers et al., 2018) is convolution-based relation prediction model. (9) ComplEx (Trouillon et al., 2016) uses the composition of complex embeddings to handle a large variety of binary relations. (10) MLN (Singla & Domingos, 2005) leverages domain knowledge to predict relations. (11) RotatE (Sun et al., 2019a) is a rotation-based relation prediction model. (12) RNNLogic (Qu et al., 2020) defines the joint distribution of all possible triplets by using Markov logic network which can be efficiently optimized with the variational EM algorithm. (13) ComplEx-N3 (Lacroix et al., 2018) introduces a regularizer based on tensor nuclear $p$-norms for relation prediction. (14) GraIL (Teru et al., 2020) reasons over local subgraph structures and holds a strong inductive bias to learn entity-independent relational semantics. (15) QuatE (Zhang et al., 2019b) introduces

Table A12: Result comparisons with different GNNs as subgraph embedding methods (i.e., $\Phi$) on knowledge graph completion task.

| Models | FB15k-237 | | | | | WN18RR | | | | |
|---|---|---|---|---|---|---|---|---|---|---|
| | MR | MRR | Hit@1 | Hit@3 | Hit@10 | MR | MRR | Hit@1 | Hit@3 | Hit@10 |
| GraphANGEL$_{\text{GCN}}^{*1}$ | 236 | 0.324 | 0.222 | 0.356 | 0.524 | 3476 | 0.410 | 0.367 | 0.445 | 0.511 |
| GraphANGEL$_{\text{GAT}}^{*1}$ | 230 | 0.328 | 0.226 | 0.361 | 0.528 | 3484 | 0.402 | 0.365 | 0.446 | 0.510 |
| GraphANGEL$_{\text{R}-\text{GCN}}^{*1}$ | 151 | 0.374 | 0.275 | 0.408 | **0.564** | 2834 | 0.504 | **0.470** | 0.515 | 0.598 |
| GraphANGEL$_{\text{R}-\text{GAT}}^{*1}$ | **141** | 0.376 | **0.276** | **0.410** | **0.564** | 2836 | 0.505 | **0.470** | 0.514 | **0.601** |
| GraphANGEL$_{\text{R}-\text{GCN}}^{*2}$ | 141 | 0.373 | 0.275 | 0.408 | **0.564** | 2829 | 0.505 | **0.470** | 0.514 | 0.597 |
| GraphANGEL$_{\text{R}-\text{GAT}}^{*2}$ | 153 | **0.377** | 0.275 | 0.408 | 0.562 | **2821** | **0.506** | 0.469 | 0.513 | 0.598 |

Table A13: Comparable results with baselines on robustness setting by randomly adding 5% noises. See Appendix A6.5 for detailed configuration on heterogeneous graph recommendation task.

| Models | LastFM | | | Yelp | | | Amazon | | | Douban Book | | |
|---|---|---|---|---|---|---|---|---|---|---|---|---|
| | AUC | ACC | F1 | AUC | ACC | F1 | AUC | ACC | F1 | AUC | ACC | F1 |
| HetGNN | 0.7756 | 0.7924 | 0.6984 | 0.8897 | 0.8104 | 0.7984 | 0.7532 | 0.6932 | 0.6962 | 0.8511 | 0.7723 | 0.7701 |
| HAN | 0.8736 | 0.8186 | 0.8134 | 0.8964 | 0.8313 | 0.8257 | 0.8315 | 0.7543 | 0.7424 | 0.9052 | 0.8335 | 0.8286 |
| TAHIN | 0.8724 | 0.8323 | 0.8187 | 0.8903 | 0.8334 | 0.8226 | 0.8344 | 0.7553 | 0.7512 | 0.9082 | 0.8354 | 0.8223 |
| HGT | 0.8212 | 0.7749 | 0.7718 | 0.8784 | 0.8183 | 0.8142 | 0.6962 | 0.6333 | 0.6150 | 0.8922 | 0.8171 | 0.8032 |
| R-GCN | 0.8355 | 0.8225 | 0.8173 | 0.8916 | 0.8258 | 0.8156 | 0.7969 | 0.7258 | 0.7218 | 0.9019 | 0.8247 | 0.8107 |
| GraphANGEL$_{3-\text{cycle}}$ | 0.8808 | 0.8392 | 0.8343 | 0.9036 | 0.8379 | 0.8394 | 0.8490 | 0.7680 | 0.7644 | 0.9126 | 0.8392 | 0.8360 |
| GraphANGEL$_{4-\text{cycle}}$ | 0.8853 | 0.8411 | 0.8365 | 0.9090 | 0.8403 | 0.8418 | 0.8502 | 0.7650 | 0.7623 | 0.9131 | 0.8391 | 0.8277 |
| GraphANGEL | 0.8808 | 0.8323 | 0.8302 | 0.9022 | 0.8310 | 0.8376 | 0.8462 | 0.7620 | 0.7582 | 0.9084 | 0.8357 | 0.8235 |
| GraphANGEL$^*$ | **0.8901** | **0.8511** | **0.8492** | **0.9230** | **0.8602** | **0.8481** | **0.8600** | **0.7720** | **0.7710** | **0.9301** | **0.8544** | **0.8492** |

quaternion embeddings, whose framework can be regarded as a generalization of ComplEx on hypercomplex space while offering better geometrical interpretations.

### A5.3 HYPER-PARAMETERS

We keep all hyper-parameter settings in their original implementations, except the following ones for fair comparisons. All models involving GNN layers are set to have similar layers (2 or 3 layers, depending on the performance) and are trained for the same number of epochs. We fix model configurations across all experiments. For all baseline GNN models and GraphANGEL, the dimension size of the embeddings is set to 128, and the learning rate space is {0.001, 0.0001}. The dropout of input is set to 0.6, of edge is set to 0.3. Notably, the results reported in our paper are comparable with those in their original papers. In the implementation of subgraph retrieval, the graph patterns in the implementation are reported in Table 2, and we search and retrieve all the supporting cases of 3-cycle and sample the same number of the supporting cases of 4-cycle, whose numbers are reported in Table A3. The number of the refuting cases of 3-cycle and 4-cycle is twice of the number of the supporting cases. The number of target subgraphs $K$ is set as 32 and the number of analogy subgraphs $Q$ (for both supporting and refuting patterns) is set as 64. The experimental comparisons are conducted with 10 different random seeds.

We report the main statistics of patterns in each datasets in Table A3, which includes the numbers of 3-cycles and 4-cycles. We further report the buffer sizes of each dataset as follows, which is in proportion to the numbers of the patterns. The buffer sizes of the supporting subgraphs of 3-cycles are: 4.5MB for LastFM, 8.7MB for Amazon, 21.2MB for Yelp, 96.2MB for Douban Book, 80.5MB for FB15k-237, 161.4MB for WN18RR where we sample twice number of the supporting subgraphs of 3-cycles in FB15k-237 for WN18RR. And, the buffer size of the supporting subgraphs of 4-cycles, and the refuting subgraphs of 3-cycles and 4-cycles can be easily derived by the above buffer size, since we store the node IDs into the buffer.

### A5.4 HARDWARE SETTINGS

The models are trained under the same hardware settings with an Amazon EC2 p3.8×large instance[1], where the GPU processor is NVIDIA Tesla V100 processor and the CPU processor is Intel Xeon E5-2686 v4 (Broadwell) processor.

---

[1]Detailed setting of AWS E2 instance can be found at https://aws.amazon.com/ec2/instance-types/?nc1=h_ls.

Table A14: Comparable results with baselines on robustness setting by randomly adding 5% noises on knowledge graph completion task. See Appendix A6.5 for detailed configuration.

| Models | FB15k-237 | | | | | WN18RR | | | | |
|---|---|---|---|---|---|---|---|---|---|---|
| | MR | MRR | Hit@1 | Hit@3 | Hit@10 | MR | MRR | Hit@1 | Hit@3 | Hit@10 |
| pLogicNet | 180 | 0.318 | 0.229 | 0.350 | 0.498 | 3532 | 0.427 | 0.380 | 0.420 | 0.518 |
| TransE | 187 | 0.316 | 0.218 | 0.346 | 0.499 | 3534 | 0.217 | 0.231 | 0.380 | 0.512 |
| ConvE | 253 | 0.286 | 0.229 | 0.326 | 0.478 | 4351 | 0.401 | 0.378 | 0.420 | 0.484 |
| ComplEx | 345 | 0.238 | 0.147 | 0.267 | 0.404 | 5452 | 0.420 | 0.394 | 0.441 | 0.494 |
| MLN | 2143 | 0.088 | 0.060 | 0.094 | 0.125 | 12667 | 0.236 | 0.172 | 0.285 | 0.334 |
| RotatE | 184 | 0.325 | 0.233 | 0.361 | 0.518 | 3531 | 0.445 | 0.404 | 0.470 | 0.552 |
| RNNLogic | 240 | 0.324 | 0.243 | 0.366 | 0.502 | 4768 | 0.468 | 0.453 | 0.484 | 0.534 |
| ComplEx-N3 | 163 | 0.358 | 0.264 | 0.381 | 0.550 | 3535 | 0.482 | 0.454 | 0.489 | 0.575 |
| GraIL | 214 | 0.310 | 0.215 | 0.324 | 0.480 | 3754 | 0.386 | 0.339 | 0.419 | 0.486 |
| QuatE | **93** | 0.330 | 0.235 | 0.367 | 0.531 | **2410** | 0.471 | 0.422 | 0.489 | 0.557 |
| GraphANGEL$_{3-cycle}$ | 163 | 0.356 | 0.265 | 0.385 | 0.550 | 2968 | 0.482 | 0.457 | 0.486 | 0.580 |
| GraphANGEL$_{4-cycle}$ | 168 | 0.345 | 0.230 | 0.372 | 0.530 | 3002 | 0.482 | 0.453 | 0.494 | 0.574 |
| GraphANGEL | 153 | **0.363** | **0.267** | **0.390** | **0.552** | 2910 | **0.489** | **0.460** | **0.495** | **0.581** |

## A6   ADDITIONAL RESULTS

### A6.1   ADDITIONAL RESULTS OF STANDARD TASK

We show the results of comparisons between GraphANGEL against the baseline methods testing with 20% least frequent relations in terms of MR, MRR, Hit@1, Hit@3 and Hit@10 in Table A5.

### A6.2   ADDITIONAL RESULTS OF GENERALIZATION STUDY

Table A6 shows the results of performance in terms of MR, MRR, Hit@K (K=1,3,10) after randomly removing 5% relations. We also report the results after randomly dropping 10%, 15% and 20% in Tables A7, A8 and A9. In addition to randomly dropping the relations, we also evaluate the performance when we randomly adding 5% relations by combining the existing relations through Composition Rules, Inverse Rules, Symmetric Rules, Subrelation Rules discussed in (Qu & Tang, 2019). The results are reported in Table A10. These experimental results verify that GraphANGEL is significantly less affected by dropping or adding relations, which demonstrates the better generalizability of our model to unseen relation types against these state-of-the-art methods.

### A6.3   ADDITIONAL RESULTS OF HEAT MAP

We show the detailed attention heat map of Douban Book dataset in Figure A5 and Figure 5 in the main text is a simplification of this figure. One can see that based on the heat map, we can not only have a global view for each graph pattern, but also see the importance of each subgraph.

Specifically, Figure A5 visualizes the attention heat map between target subgraphs and supporting subgraphs when predicting relations between users and books. The rows represent the supporting subgraphs while the columns represent the target subgraphs. We denote each subgraph as a tuple of node IDs, e.g., $\mathcal{S}_7^+ = (u_{107}, b_{11}, a_{33}, b_{12})$. Each cell represents the similarity between a target subgraph (at the top) and a supporting subgraph (at bottom). The color of each cell shows the attention weight for corresponding pair of supporting and target subgraphs. We can observe that the deep color of the cell located at the interaction of $\mathcal{S}_7^*$ and $\mathcal{S}_{32}^+$ indicates more similarity. We can see from the figure that the cells with darkest color have both target subgraph and supporting subgraph as $(u., b., a., b.)$, which indicates a subgraph with $\text{User} - \text{Book} - \text{Author} - \text{Book}$ pattern, meaning that this particular pattern is more influential in predicting $\text{User} - \text{Book}$ relation. Notably, this result reveals a simple logic $\text{User} \wedge \text{Book} \wedge \text{Author} \wedge \text{Book} \Rightarrow \text{User} \wedge \text{Book}$ indicating a simple factor that the users are likely to be interested in the book sharing the same authors with his reading ones. Hence, our method can intuitively be regarded as the a logical collaborative filtering technique working on graph. Similarly, we can find that 3-cycle pattern $(\text{Person}) - \text{Impersonate}^{-1} - (\text{Person}) \wedge (\text{Person}) - \text{Nationality} - (\text{Nation}) \Rightarrow (\text{Person}) - \text{Nationality} - (\text{Nation})$ plays an important role in FB15k-237 dataset, where inverse relations are denoted with a superscript $^{-1}$.

### A6.4   ADDITIONAL RESULTS OF ABLATION STUDY OF SUBGRAPH EMBEDDING METHODS

In Section 3, we use R-GCN (Schlichtkrull et al., 2018) with single layer (denoted as GraphANGEL$_{R-GCN}^{*1}$) as an example of $\Phi$. In this subsection, we further investigate the effect of $\Phi$

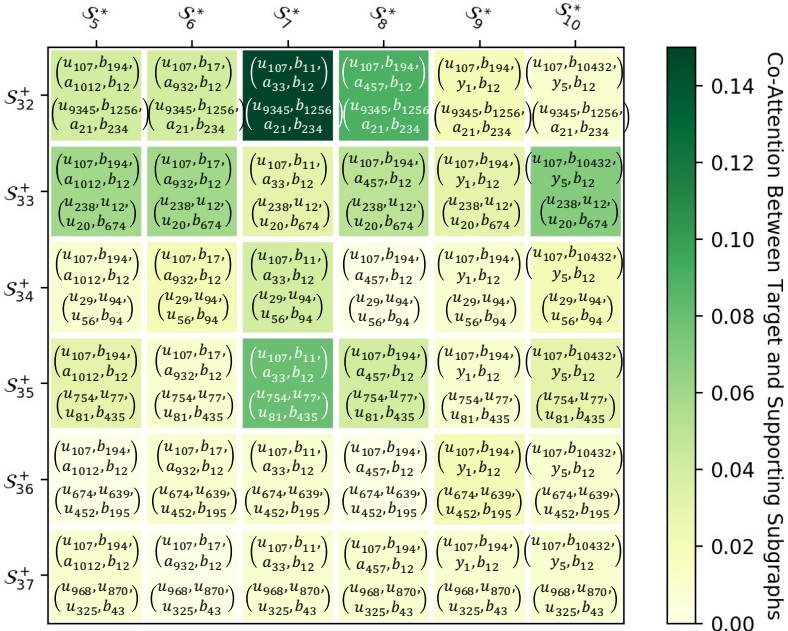

Figure A5: Illustrations of generated heat map of attention scores. While Figure 5 can provide a global view for each graph pattern, this figure can see the importance of detailed subgraphs.

by applying different GNNs including GCN (Kipf & Welling, 2016) (denoted as GraphANGEL$_{\text{GCN}}^{*1}$), GAT (Hamilton et al., 2017) (denoted as GraphANGEL$_{\text{GAT}}^{*1}$), R-GAT (Busbridge et al., 2019) (denoted as GraphANGEL$_{\text{R-GAT}}^{*1}$). We do not choose knowledge embedding based methods including TransE (Bordes et al., 2013), RotatE (Sun et al., 2019a), ComplEx (Trouillon et al., 2016), as these approaches need to introduce additional supervisions/gredients on the subgraph embeddings, which require significant modifications on the framework of GraphANGEL. We do not change any other settings of GraphANGEL for fair comparison, and report the results on heterogeneous recommendation task and knowledge graph completion task in Tables A11 and A12 respectively. Results shows that there are slight differences between GraphANGEL$_{\text{R-GCN}}^{*1}$ and GraphANGEL$_{\text{R-GAT}}^{*1}$. One possible explanation is that as our retrieved subgraphs only includes few nodes (e.g. 2,3,or 4 nodes), there is no requirement of GraphANGEL to mine some complex structure (e.g., high-order neighborhood).

We further study the impact of using the multiple layers of GNNs. We establish a series of variants using two layers of of R-GCN and R-GAT, denoted as GraphANGEL$_{\text{R-GCN}}^{*2}$ and GraphANGEL$_{\text{R-GAT}}^{*2}$. Results reported in Tables A11 and A12 show that there are only slight differences when comparing to GraphANGEL$_{\text{R-GCN}}^{*1}$ and GraphANGEL$_{\text{R-GAT}}^{*1}$, which is consistent with the above explanation.

## A6.5 ADDITIONAL RESULTS OF EFFECT OF NOISY GRAPHS

In order to further investigate the robustness of GraphANGEL, we randomly generate some noises and add them into the training graphs. Concretely, for heterogeneous graphs and knowledge graphs shown in Tables A1 and A2, we randomly change the source or target node of each triplet $\langle s, r, t \rangle$ with probability 5%. Since the heterogeneous graphs have the node types, our changes are made by replacing the current source or target node with a randomly sampled node in the same node type. We compare our model with all the baselines and report the results for heterogeneous graph recommendation and knowledge graph completion tasks in Tables A13 and A14. These results show that our method can consistently outperform these baseline methods, sometimes, even are less influenced by the noises. One explanation is that if we introduce noises in the labels $y$, then the learnable subgraph embeddings $s^+$ and $s^-$ will simultaneously be influenced by the noises. These influences might be counteracted according to Eq. (3).

## A6.6 ADDITIONAL RESULTS OF ABLATION STUDY OF BUFFER SIZE

We establish an ablation study to investigate the impact of the buffer size $\mathcal{B}$ in Algorithm 8. We here focus on the number of the subgraphs following $\Pi_{3-\text{cycle}}^+$ and $\Pi_{4-\text{cycle}}^+$. From Appendix A3, we know that we search and sample $N_{3-\text{cycle}}$ subgraphs following $\Pi_{3-\text{cycle}}^+$ and $\Pi_{4-\text{cycle}}^+$, where

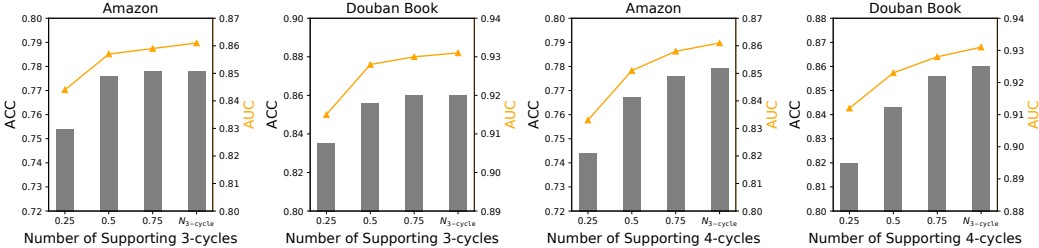

Figure A6: Performance change of GraphANGEL with different size of the buffer in term of ACC and AUC, where we investigate the performance of GraphANGEL using 0.25, 0.5, 0.75 number of supporting instance in 3-cycles (i.e., $\Pi^+_{3-\text{cycle}}$), or 0.25, 0.5, 0.75 number of supporting instance in 4-cycles (i.e., $\Pi^+_{4-\text{cycle}}$) on Amazon and Douban Book datasets.

$N_{3-\text{cycle}}$ is the number of subgraphs following $\Pi^+_{3-\text{cycle}}$. We reduce the number of either subgraphs following $\Pi^+_{3-\text{cycle}}$ or $\Pi^+_{4-\text{cycle}}$. Results reported in Figure A6 shows that once the numbers of supporting 3-cycles and 4-cycles surpass $0.5 \cdot N_{3-\text{cycle}}$ and $0.75 \cdot N_{3-\text{cycle}}$ respectively, there will be no significant difference to further increase the buffer size.

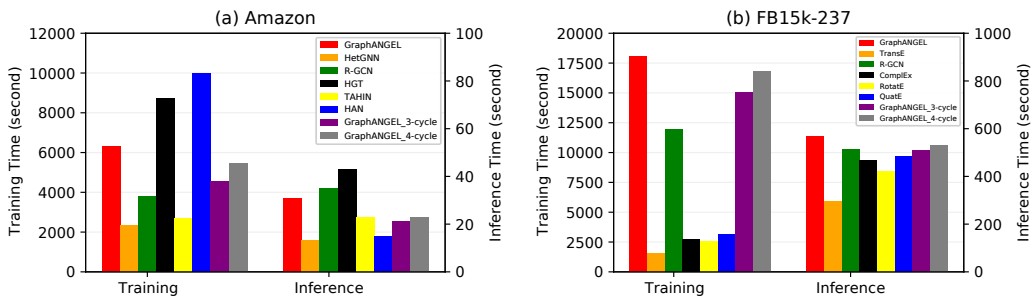

Figure A7: Time comparisons of GraphANGEL against baseline models on Amazon dataset in (a) and on FB15k-237 dataset in (b).

## A6.7 ADDITIONAL RESULTS OF TIME COMPLEXITY

We first compare the time complexity of GraphANGEL in term of both training and inference times on Amazon dataset against aforementioned baseline methods. Result in Figure A7(a) reveals that GraphANGEL is more efficient than HGT and HAN in term of training time and is more efficient than R-GCN and HGT in term of inference time. One possible explanation is that HAN, HGT, and R-GCN contain the whole graph, which is much more complex than sampled subgraphs. We further evaluate the training and inference time of GraphANGEL against the baselines on FB15k-237 dataset and report the results in Figure A7(b). As the prevailing methods on knowledge completion task are mainly based on either on translation (Bordes et al., 2013) or rotation (Sun et al., 2019a) assumptions regardless of the graph structure, they are naturally more efficient than GNN based relation prediction models. We do note that our efficiency is comparable against other graph neural networks (i.e., R-GCN) which have also been tried on knowledge graphs.

---

**Algorithm 11:** Incremental Updating Algorithm for $\mathcal{B}$

---

$\mathcal{B} \leftarrow \mathcal{B}_{\text{old}}$

**for** *each* $v \in \Delta\mathcal{V}^-$ **do**
   | Remove all the instances including $v$ in $\mathcal{B}$.
**end**

\# Following is Search and Retrieval for $\Pi^+_{3-\text{cycle}}$ in $\Delta\mathcal{G}^+_{\text{new}}$

Search, retrieve and store the instances following $\Pi^+_{3-\text{cycle}}$ in $\Delta\mathcal{G}^+_{\text{new}}$ using Algorithm 2 or 3.

\# Followings is Search and Retrieval for $\Pi^+_{3-\text{cycle}}$ in (a) Shape

Search, retrieve and store the instances following $\Pi^+_{3-\text{cycle}}$ in (a) shape using Algorithm 9.

\# Followings is Search and Retrieval for $\Pi^+_{3-\text{cycle}}$ in (b) Shape

**for** *each* $u \in \mathcal{V}_{\text{old}}$ **do**
   **for** *each* $\langle v,w \rangle \in \Delta\mathcal{V}^+_{\text{new}}$ **do**
      **if** $\langle u,v \rangle \in \Delta\mathcal{E}^+_{\text{old}}$ *and* $\langle u,w \rangle \in \Delta\mathcal{E}^+_{\text{old}}$ **then**
         | Add $(u,v,w)$ into $\mathcal{B}$.
      **end**
   **end**
**end**

\# Following is Uniform Sampling for $\Pi^+_{4-\text{cycle}}$ in $\Delta\mathcal{G}^+_{\text{new}}$

Sample and store $\lceil \frac{|\Delta\mathcal{V}^+_{\text{new}}|^4}{|\mathcal{V}_{\text{old}}|^4} n_{4-\text{cycle}} \rceil$ instances following $\Pi^+_{4-\text{cycle}}$ in $\Delta\mathcal{G}^+_{\text{new}}$ using Algorithm 5.

\# Following is Uniform Sampling for $\Pi^+_{4-\text{cycle}}$ in (c), (d), (e) Shapes

Compute sample rate $p^{\text{edge}}_{\Delta\mathcal{G}^+_{\text{old}}}$ by running purple part in Algorithm 10 on $\Delta\mathcal{G}^+_{\text{old}}$.

**for** $i \leftarrow 1$ to $\lceil \frac{|\Delta\mathcal{E}^+_{\text{old}}|^2}{|\mathcal{E}_{\text{old}}|^2} n_{4-\text{cycle}} \rceil$ **do**
   Sample $\langle u,v \rangle \in \Delta\mathcal{E}^+_{\text{old}}$ with probability $p^{\text{edge}}_{\Delta\mathcal{G}^+_{\text{old}}}$.
   Sample $\langle w,x \rangle \in \Delta\mathcal{E}^+_{\text{old}}$ with probability $p^{\text{edge}}_{\Delta\mathcal{G}^+_{\text{old}}}$.
   **if** *{$\langle u,w \rangle \in \mathcal{E}_{\text{old}} \cup \Delta\mathcal{E}^+_{\text{new}}$ and $\langle v,x \rangle \in \mathcal{E}_{\text{old}} \cup \Delta\mathcal{E}^+_{\text{new}}$} or {$\langle u,w \rangle \in \mathcal{E}_{\text{old}} \cup \Delta\mathcal{E}^+_{\text{new}}$ and $\langle v,x \rangle \in \mathcal{E}_{\text{old}} \cup \Delta\mathcal{E}^+_{\text{new}}$}* **then**
      | Add $(u.v,w,x)$ into $\mathcal{B}$.
   **end**
**end**

\# Following is Uniform Sampling for $\Pi^+_{4-\text{cycle}}$ in (f), (g) Shapes

Compute sample rate $p^{\text{edge}}_{\mathcal{G}_{\text{old}}}$ and $p^{\text{node}}_{\Delta\mathcal{G}^+_{\text{new}}}$ by running blue part in Algorithm 10 on $\mathcal{G}_{\text{old}}$ and $\Delta\mathcal{G}^+_{\text{new}}$.

**for** $i \leftarrow 1$ to $\lceil \frac{|\Delta\mathcal{E}^+_{\text{old}}|^2}{|\mathcal{E}_{\text{old}}|^2} n_{4-\text{cycle}} \rceil$ **do**
   Sample $x \in \Delta\mathcal{V}^+_{\text{new}}$ with probability $p^{\text{node}}_{\Delta\mathcal{G}^+_{\text{new}}}$.
   Sample $v \in \mathcal{V}_{\text{old}}$ with probability $p^{\text{node}}_{\mathcal{G}_{\text{old}}}$.
   Uniformly sample two different neighbor nodes $u,w \in \mathcal{V}_{\text{old}}$ of $v$.
   **if** $\langle u,x \rangle \in \Delta\mathcal{E}^+_{\text{old}}$ *and* $\langle w,x \rangle \in \Delta\mathcal{E}^+_{\text{old}}$ **then**
      | Add $(u,v,w,x)$ into $\mathcal{B}$.
   **end**
**end**

**for** $i \leftarrow 1$ to $\lceil \frac{|\Delta\mathcal{E}^+_{\text{old}}|^2 |\Delta\mathcal{E}^+_{\text{new}}|^2}{|\mathcal{E}_{\text{old}}|^4} n_{4-\text{cycle}} \rceil$ **do**
   Sample $x \in \mathcal{V}_{\text{old}}$ with probability $p^{\text{node}}_{\mathcal{G}_{\text{old}}}$.
   Sample $v \in \Delta\mathcal{V}^+_{\text{new}}$ with probability $p^{\text{node}}_{\Delta\mathcal{G}^+_{\text{new}}}$.
   Uniformly sample two different neighbor nodes $u,w \in \Delta\mathcal{V}^+_{\text{new}}$ of $v$.
   **if** $\langle u,x \rangle \in \mathcal{E}^+_{\text{old}}$ *and* $\langle w,x \rangle \in \mathcal{E}^+_{\text{old}}$ **then**
      | Add $(u,v,w,x)$ into $\mathcal{B}$.
   **end**
**end**

\# Following is Uniform Sampling for $\Pi^-_{3-\text{cycle}}$ and $\Pi^-_{4-\text{cycle}}$

Re-sample and store $\lceil \frac{|\mathcal{E}_{\text{new}}|^3}{|\mathcal{E}_{\text{old}}|^3} n^-_{3-\text{cycle}} \rceil$ and $\lceil \frac{|\mathcal{E}_{\text{new}}|^3}{|\mathcal{E}_{\text{old}}|^3} n^-_{4-\text{cycle}} \rceil$ refuting cases of $\Pi^-_{3-\text{cycle}}$ and $\Pi^-_{4-\text{cycle}}$ from $\mathcal{G}_{\text{new}}$ using Algorithms 7 and 5 with linear computation costs.

---

