# OpenReview forum: "Inductive Relation Prediction Using Analogy Subgraph Embeddings"
_ICLR.cc/2022/Conference — ICLR 2022 Poster_

### Official Review · Reviewer_y1cR · 2021-10-30

**Correctness:** 4
**Technical Novelty And Significance:** 2
**Empirical Novelty And Significance:** 2
**Recommendation:** 8
**Confidence:** 4

**Main Review:**

Strengths:

1) The proposed method doesn’t require learning explicitly embeddings for the relation types, which makes it quite effective. The ability to generalize to unseen relation types has been shown with an exhaustive set of experiments.

2) The gains provided in all the tables are very significant and the results in Tables 5 and 6 are particularly interesting and very substantial.

3) The monotonic increase in the performance reported in section 5.3 clearly demonstrates the importance of the number of the subgraphs that are used.

4) The attention heatmaps in Figure 5 are indeed very informative and interesting. Figures 5 and A4 clearly support the authors’ claim about their method being explainable, further aiding its utility.

5) The authors have also put some effort on providing the extension of their method to complex logic in section A 4.3. It will be interesting to see some empirical work for that in the future. Although trivial, they have also mentioned the extension to homogeneous graphs.

6) The authors also provide various search and sampling algorithms while also noting the proofs of correctness and performing the complexity analysis for all of them. The proposed techniques are certainly useful in practice, at least for sparse graphs such as Wordnet.


Weaknesses (and questions for authors):

1) Is it possible for the authors to compare against strong baselines methods for knowledge graph completion such [1] and [2]? I understand that the publication time for [1] is within 3 months of this submission, but it might be good to provide the results on at least one of the knowledge graph datasets to ensure completeness.

2) The authors’ claim about bridging logical expressions and graph patterns based on the two intuitions provided at the beginning of section 2 are indeed very sound and they have provided some examples in the subsequent paragraphs. Can they also provide reference to some relevant works that have explicitly identified such patterns?

3) Although the computation of 3-cycles and 4-cycles as a preprocessing step and storing explicitly is reasonable, this can limit the practical usage of this method, for which the authors suggest the use of Buffer B in section A 3.1. Can the authors provide more experimental details regarding the use of buffer and its size on various datasets? Can they also provide some statistics on the number of 3-cycles and 4-cycles generated for the benchmark graphs? An ablation study for the total number of cycles used in the training data could also be very useful.

4) Is there any specific reason that the authors used R-GCN? From Equation 11, it seems that some learnable parameters are required for the relation types present in training data, thus I expect that one can use any recent Graph based method proposed to operate on KGs. Can they also elaborate on the reason as to why a single layer has been used?

5) Can the authors also elaborate whether they have considered the graphs as directed or undirected? All the algorithms seem to be operating on undirected graphs. If that is the case, can they explain the reason to do so? The direction of the edges is extremely important in the benchmark graphs since it has explicit meaning. (If they have considered directed graphs, please ignore this point)

6) Can the authors also provide some more empirical runtime details, particularly on the Freebase dataset?

7) The authors mention this statement - “We note that such cases are rare in real-world graphs as they are usually scale-free, meaning that the topological distance is usually small.” at the end of section A 4.1 . Can they provide references to some relevant works that have explicitly studied this? At least for real-world knowledge graph completion tasks, the nodes can definitely be far away, with the hop distance (shortest path length) of the order of the diameter of the training graph.

[1] Zhu, Zhaocheng, Zuobai Zhang, Louis-Pascal Xhonneux, and Jian Tang. "Neural Bellman-Ford Networks: A General Graph Neural Network Framework for Link Prediction." arXiv preprint arXiv:2106.06935 (2021).

[2] Zhang, Shuai, Yi Tay, Lina Yao, and Qi Liu. "Quaternion knowledge graph embeddings." arXiv preprint arXiv:1904.10281 (2019)


**Summary Of The Paper:**

The authors propose a framework, which they refer to as GraphANGEL, for inductive relation prediction in heterogeneous graphs. One of the main and inarguably the most important aspects of this work is modeling the task of relation prediction as a subgraph matching problem. Since the method doesn’t require learning embeddings for relation types explicitly, it can easily generalize to new relations. Furthermore, since the proposed method requires generation of explicit graph patterns that represent logical rules, the method becomes inherently explainable. The authors further provide various algorithms for searching and retrieving subgraphs from the training data with appropriate proofs and runtime complexity wherever required. Exhaustive experiments have been conducted to demonstrate the efficacy of the proposed approach and the results provided in the main paper as well as the appendix thoroughly support the authors’ claims. The heatmaps provided in section 5.5 and A 6.3 further aid the claims about explainability and are indeed very interesting.

Formulating the problems of link and relation prediction as subgraph matching have been extensively studied in the literature and there is conclusive evidence that such approaches work very well.

It is also very informative that the authors talk about the limitations of their work in section A 4.1 .


**Summary Of The Review:**

Overall the paper is very well written, contains extensive experiments supporting their claims and proofs of correctness as well as runtime complexity analysis for all proposed algorithms, ensuring completeness. It will be very helpful if the authors can further address some of the  questions and weaknesses as pointed out in the previous subsection of this review, particularly the inclusion of some recent and strong knowledge graph completion baselines. I look forward to the rebuttal and am willing to improve the score if the authors can provide a satisfactory response.

After addressing these minor pointers, I believe that the contributions will be worth presenting at the conference. I additionally support the simplicity of the proposed approach and given its versatility, it can be plugged with various types of  deep learning based components used for subgraph embedding computation and similarity computation.

---

> ### Author Response · Authors · 2021-11-23
> **Response to Reviewer y1cR Part 1**
>
> Thanks for your feedback and suggested experiments. Please also see the main comment above.
>
> > Is it possible for the authors to compare against strong baselines methods for knowledge graph completion such [1] and [2]?
>
> Thanks for your suggestion.
> 1. We include [1] (named QuatE in standard setting and QuatE$^*$ in generalization settings) as a baseline method and update its results in Tables 6, A5, A6, A7, A8, A9. These results further verify the superiority of the proposed method in terms of MRR, Hit@1, Hit@3, Hit@10.
> 2. For [2], we are unable to find the source code and detailed hyper-parameter settings such as the dimensions of embedding in the paper. So, we do not presently include this work. However, we cite [2] and leave the comparison as future work.
>
> > Can they also provide reference to some relevant works that have explicitly identified such patterns?
>
> We added some references about using chain [3], tree [4], and graph [5] structures to represent the logical rules. We have updated Section 2 in the revised PDF.
>
> > Can the authors provide more experimental details regarding the use of buffer and its size on various datasets? Can they also provide some statistics on the number of 3-cycles and 4-cycles generated for the benchmark graphs? An ablation study for the total number of cycles used in the training data could also be very useful.
>
> We provide the experimental details of the buffers in Appendix 5.3, and report the statistics of the patterns in Table A3. We study the impact of using different sizes of the buffer in Appendix 6.6. Results reported in Figure A6 show that the performance of GraphANGEL will be significantly influenced if there are only a few subgraphs in the buffer, and there are slight differences once the buffer size surpasses a threshold.
>
> > Is there any specific reason that the authors used R-GCN? Can they also elaborate on the reason as to why a single layer has been used?
>
> To begin, we want to emphasize that the proposed GraphANGEL is a general framework and R-GCN is only one of the many possible components for graph representation computation (i.e., $\Psi$ in Eq. (1)). Additionally, we have provided an ablation study using GCN [6], GAT [7], R-GCN [8] R-GAT [9] with different numbers of layers as $\Phi$ in Appendix 6.4. Results reported in Tables A10 and A11 show that there is a slight difference between R-GCN and R-GAT with the different numbers of layers, as the retrieved subgraphs only include a few nodes (as shown in Table 2). However, there is a huge gap between R-GCN, R-GAT and GCN, GAT, which indicates the importance of considering the relation information in the graph encoding.
>
> > Can the authors also elaborate whether they have considered the graphs as directed or undirected?
>
> The heterogeneous graphs reported in Table A1 are naturally undirected graphs; while the knowledge graphs reported in Table A2 are directed graphs. Following previous works [10][11], we add the inverse relations in the pre-processing procedure and consider them as undirected graphs.
>
> > Can the authors also provide some more empirical runtime details, particularly on the Freebase dataset?
>
> We provide the training and inference time of GraphANGEL along with baselines in Appendix 6.7. Results depicted in Figure A7(b) show GraphANGEL is less efficient than knowledge graph based methods  (e.g., TransE [10]) on the FB15k-237 dataset which is constructed based on the Freebase dataset.  We do note that our efficiency is comparable to other graph neural networks (i.e., R-GCN [5]) which have also been tried on knowledge graphs.

---

> ### Author Response · Authors · 2021-11-23
> **Response to Reviewer y1cR Part 2**
>
> > The authors mention this statement - “We note that such cases are rare in real-world graphs as they are usually scale-free, meaning that the topological distance is usually small.” at the end of section A 4.1. Can they provide references to some relevant works that have explicitly studied this?
>
> Thanks for your question. Theoretically, it has been well formulated as Wiener index [12]; however, due to the high computation costs for shortest path algorithms, there is no empirical reference on the actual diameter of the graphs (used in this paper), instead, we calculated the fraction of 3-hop and 4-hop neighborhoods over all the nodes for every node in each dataset, and averaged them. Results reported in Table A3 show that generally speaking, it is very unlikely for two nodes to be more than 3-hops and 4-hops away from each other. Please refer to Appendix 4.1 for a detailed discussion.
>
> [1] Quaternion Knowledge Graph Embeddings. Zhang et al., NeurIPS 2019.
>
> [2] Neural Bellman-Ford Networks: A General Graph Neural Network Framework for Link Prediction. Zhu et al., Neurips 2021.
>
> [3] Differentiable Learning of Logical Rules for Knowledge Base Reasoning. Yang et al., NIPS 2017.
>
> [4] Learn to Explain Efficiently via Neural Logic Inductive Learning. Yang et al., ICLR 2020.
>
> [5] Differentiable Learning of Graph-like Logical Rules from Knowledge Graphs. Shi et al., arXiv 2021.
>
> [6] Semi-Supervised Classification with Graph Convolutional Networks. Kipf et al., ICLR 2017.
>
> [7] Graph Attention Networks. Velickovic et al., ICLR 2018.
>
> [8] Modeling Relational Data with Graph Convolutional Networks. Schlichtkrull, et al., ESWC 2018.
>
> [9] Relational Graph Attention Networks. Busbridge et al., arXiv 2019.
>
> [10] Translating Embeddings for Modeling Multi-relational Data. Bordes, et al., NIPS 2013.
>
> [11] RotatE: Knowledge Graph Embedding by Relational Rotation in Complex Space. Sun, et al., ICLR 2019.
>
> [12] Wiener index of trees: theory and applications. Dobrynin, et al., Acta Applicandae Mathematica 2001.

---

### Official Review · Reviewer_fant · 2021-10-31

**Correctness:** 4
**Technical Novelty And Significance:** 3
**Empirical Novelty And Significance:** Not applicable
**Recommendation:** 8
**Confidence:** 5

**Main Review:**

Strengths:
1. The idea of considering different analogy graph patterns as negative and positive ones with respect to target patterns is interesting that allows the relations to be inferred from similar relations.
2. Paper is mostly explained well and easy to follow.
3. Experiments show improved results for two tasks on 6 six datasets with significant margins.
4. Time complexity is provided to get different graph patterns.
5. The method can be plugged into any kind of GNN to see the significance of subgraph patterns being compared.

Weaknesses:
I find the idea interesting. However, I have some strong concerns:
1. What about the graphs with noise since the majority of the real-world graphs have errors or noise? The method seems to focus on learning from similar subgraph patterns that would lead to incorrect results in the case of noisy graphs.
2. I do not get the idea that says subgraphs have the same topology. All the subgraphs look similar (triangle and quadrangle) and would not make much difference in terms of structure.
3. I believe the majority of the learning would be from the feature embeddings. I don’t see any exploitation/ usage of graph structure in the proposed method.
4. The points mentioned under limitations in the paper are quite important. The results would be severely affected in case of missing links/ information in the graph. Also, how to overcome the second limitation. The problem would increase with larger/dense graphs.
5. Experiments should have been conducted with different GNN architectures to see the significance of the proposed graph pattern comparisons.

Questions to authors:
1. What happens if a relation is being used in a different context i.e., the presence of ambiguity. Does that mean the method produces incorrect results?
2. Did you consider cases where there are rare relations? For example, if the graph has a relation “live” only once in the graph.
3. How large graphs can the proposed method handle?

**Summary Of The Paper:**

The paper proposes GraphANGEL, a relation prediction framework that checks if subgraphs containing the node pair for predicting relation are similar to other subgraphs containing the particular relation. The overall idea is based on graph patterns in which analogy subgraphs are extracted for a node pair and compared against similar shape subgraphs. These subgraphs are based on three kinds of patterns I.e., target patterns, supporting patterns, and refuting patterns. The framework considers heterogeneous graph based recommendation as well as knowledge graph completion. Experiments are conducted for the above two tasks on six datasets against several baselines from the area of relation prediction.

**Summary Of The Review:**

The idea of the paper is different. Experiments show improved results. However, I have concerns about the methodology that majorly considers the feature embeddings without exploiting the graph structure. More experiments (mentioned in the main review) could be conducted to back the idea. Also, the points under limitations can severely affect the method in different scenarios.

---

> ### Author Response · Authors · 2021-11-23
> **Response to Reviewer fant Part 1**
>
> Thanks for your feedback and suggested experiments. Please also see the main comment above.
>
> > What about the graphs with noise since the majority of the real-world graphs have errors or noise?
>
> We devised the noisy graphs by modifying the source or target nodes with probability 5\%, and compared GraphANGEL to baselines on the graphs. Results reported in Tables A12 and A13 show the superiority of GraphANGEL against all the baselines in the context of noisy graphs in both heterogeneous graph recommendation and knowledge graph completion tasks. Please see Appendix 6.5 for details.
>
> > All the subgraphs look similar (triangle and quadrangle) and would not make much difference in terms of structure.
>
> We explain the difference among graph patterns (e.g., 3-cycle and 4-cycle) and why we use these patterns as follows:
> 1. As mentioned in Section 2, each pattern represents a specific logical rule, as illustrated in Figure 1. Therefore, the subgraphs following 3-cycle and 4-cycle shaped patterns might look similar in the structure, but there is a significant difference between the logical rules that these graph patterns represent.
> 2. We use simple graph patterns (e.g., 3-cycles and 4-cycles) instead of complex graph patterns, because (i) they are simple and efficient for subgraph search and retrieval as shown in Appendix 2, (ii) they already can achieve good results.
>
> > I believe the majority of the learning would be from the feature embeddings. I don’t see any exploitation/ usage of graph structure in the proposed method.
>
> We apologize that we may not fully understand the reviewer's meaning here in terms of exploitation/usage of graph structure (if the reviewer could elaborate further, we would be happy to respond more precisely).  But as brief background context, we do believe that GraphANGEL uses information from graph structure for the following reasons:
> 1) As introduced in Section 3, GraphANGEL uses information from graph structure by extracting subgraphs and running message passing on them. The subgraphs are extracted both around the center node by finding target patterns, and outside the center node's local neighborhood by finding supporting and refuting patterns. As illustrated in Figure A2, our method can be regarded as to firstly sample the subgraphs satisfying certain pre-defined patterns (e.g., 3-cycles, 4-cycles), and then restrict the message passing within these subgraphs. Please refer to Appendix 4.1 for a detailed discussion.
> 2) In the experiments, there are no primitive features for all the datasets reported in Table A1 and A2. Namely, there are no raw features for users or items in heterogeneous graphs and no knowledge embeddings pre-trained by other natural language processing models. Therefore, all we have is the graph structure including the adjacency matrix, node types for heterogeneous graphs, and edge types (i.e., relations) for knowledge graphs.  Hence we would argue that GraphANGEL predictions are based more-or-less entirely on graph structure.
>
> > How to overcome the limitation of time computations. The problem would increase with larger/dense graphs.
>
> We summarize our approaches to overcome the limitation of time computations as follows:
> 1. We provide a series of subgraph retrieval and uniform sampling algorithms in Appendix 2 (i.e., Algorithms 2,3,4 for searching and retrieving the supporting cases of 3-cycles and 4-cycles and Algorithms 5,6,7 for uniformly sampling the supporting cases of 4-cycles and refuting cases of 3-cycles and 4-cycles).
> 2. We propose to pre-compute and store the retrieved or sampled subgraphs in Algorithm 8, which is a one-time round computation. We also provide the corresponding analysis in Appendix 3.1.
> 3. We further provide Algorithm 11, the incremental updating algorithm (along with its time complexity analysis) in Appendix 3.2 to handle dynamic graphs in the context of Algorithm 8.
>
> > Experiments should have been conducted with different GNN architectures to see the significance of the proposed graph pattern comparisons.
>
> Although we note that R-GCN is only one of the possible components for GraphANGEL, we have provided an ablation study using GCN [1], GAT [2], R-GCN [3] R-GAT [4] with different numbers of layers as $\Phi$ in Appendix 6.4. Results reported in Tables A10 and A11 show that there is a slight difference between R-GCN and R-GAT with the different numbers of layers, as the retrieved subgraphs only include a few nodes (as shown in Table 2). However, there is a huge gap between R-GCN, R-GAT and GCN, GAT, which indicates the importance of considering the relation information in the graph encoding.

---

> > ### Comment · Reviewer_fant · 2021-11-25
> > **Thanks for such a detailed response and experiments.**
> >
> > Thanks to the authors for the response and for making the required changes to the paper.
> >
> > I'm quite satisfied with the response as it answers the majority of my concerns. The authors have done a good job in adding all the details asked in the reviews with the detailed discussions and experiments in the appendix. Although there are some limitations of the work I believe that the paper would be a good contribution to the knowledge graph/ relation prediction area in general with a new concept of different subgraph patterns. Thus, I'm increasing my score.

---

> ### Author Response · Authors · 2021-11-23
> **Response to Reviewer fant Part 2**
>
> > What happens if a relation is being used in a different context i.e., the presence of ambiguity. Does that mean the method produces incorrect results?
>
> As mentioned in Section 2, our method extracts subgraphs that match target patterns, supporting patterns and refuting patterns, and runs message passing to compute their representations. Therefore, GraphANGEL is capable of distinguishing different semantics of the same relation as long as the local subgraphs, hence contexts, are different.
>
> > Did you consider cases where there are rare relations?
>
> We show the experimental results for GraphANGEL and baselines on the 20\% least frequent relations in Table 5, where each model is trained with all the relations but tested only with the 20\% least frequent relations. Results show that our method can significantly outperform the existing methods in terms of MRR, Hit@1, Hit@3, Hit@10.
>
> > How large graphs can the proposed method handle?
>
> 1. We report the training and inference times of GraphANGEL along with baseline methods, which indicates that our method can handle the datasets reported in Tables A1 and A2.  We did not try scaling to larger graphs, which can be left as future work.
> 2. As shown in Algorithm 8, we re-compute and store the subgraphs by searching all the supporting cases of 3-cycles (using Algorithm 2 or 3), uniformly sampling a number of supporting cases of 4-cycles (using Algorithm 5) and uniformly sampling a number of refuting cases of 3-cycles and 4-cycles (using Algorithms 6 and 7). Then, we only need to compute once. If there are further modifications on graphs, we can incrementally update the buffer using Algorithm 11. To further increase the efficiency of GraphANGEL to accommodate even larger graphs, one can also consider using parallel computations; however, we leave this as a direction for future work.
>
> [1] Semi-Supervised Classification with Graph Convolutional Networks. Kipf et al., ICLR 2017.
>
> [2] Graph Attention Networks. Velickovic et al., ICLR 2018.
>
> [3] Modeling Relational Data with Graph Convolutional Networks. Schlichtkrull, et al., ESWC 2018.
>
> [4] Relational Graph Attention Networks. Busbridge et al., arXiv 2019.

---

### Official Review · Reviewer_jqL2 · 2021-11-02

**Correctness:** 4
**Technical Novelty And Significance:** 3
**Empirical Novelty And Significance:** 2
**Recommendation:** 8
**Confidence:** 4

**Main Review:**

Pros:
- The approach presented is simple and based on intuition from the real world.
- The approach is inductive and thus can be applied to unseen relations at inference time.
- Comparing multiple target subgraphs with multiple supporting/refuting subgraphs allows the use of non-local explicit structure at the prediction time.
- Proposes efficient methods for pre-computing sub-graphs with 3-cycle and 4-cycle patterns.
- The comparison of the target with the supporting/refuting subgraphs is done using a co-attention module that allows for interpretability which can be important in certain domains like the medical domain.
- The ablation analysis is quite thorough supporting the design choices of their model.

Cons:
- While the approach is general, but the experiments are conducted by using patterns of up to 4-cycle which can be a problem if the target nodes s and t are quite far in the graph and/or the graph is disconnected. This is also pointed out by the authors.
- Because there is no explicit embedding for relation, it will not be possible to incorporate unseen relations as a part of the graph after actually observing them during inference. For instance, r1_unseen (e.g. basedin), r2_unseen(e.g. live) are 2 unseen relations observed during inference. Knowing r1_unseen can help with r2_unseen as pointed out by authors as well in the paper. As far as I know, there is no way for RGCN to incorporate the new relationship without explicitly re-training, it will miss out on using r1_unseen for predicting about r2_unseen. While having an explicit embedding/matrix for relation would actually be useful in that setting.
- The results are marginally better and the error bars (from multiple runs) are not shown which makes it harder to say if the difference is statistically significant. It would be great to have that just for the models in the main table.
- The approach precomputes the subgraphs which would be a single-time operation in a static graph. The authors also allude to this fact in Limitations on page 5.  But this may not be the case for heterogeneous recommendation settings, where I would expect that the graph to evolve very fast. In that scenario, I am concerned that there is a need to precompute the subgraphs again?


**Summary Of The Paper:**

The paper proposes a novel inductive relation prediction framework called ANalogy SubGraph Embedding Learning (GraphANGEL). The core idea of the approach is to compare a target subgraph (containing the nodes s and t between which the relation r is to be predicted) with other subgraphs in the graph that supports/refute the presence of the relationship. If the target subgraph is closer to the supporting pattern/subgraph then the target relationship r should also exist or vice versa. The subgraphs are constructed using 3 graph patterns namely pairs, 3-cycles, and 4-cycles.  The proposed approach is inductive as the approach doesn’t build an explicit embedding for relationships (unlike existing approaches) for prediction but masks the relationship edge while computing the subgraph representations for the supporting subgraphs. The paper proposes to precompute patterns like pair, 3-cycle, and 4-cycle and their corresponding subgraphs which are then retrieved during training and inference to get the target, supporting and refuting subgraphs. Considering specifically pairs, 3-cycles, and 4-cycles helps to keep the approach tractable. The approach is shown to consistently outperform existing approaches on heterogeneous recommendation and Knowledge Graph Completion (in both inductive and transductive settings for relation prediction) tasks.

**Summary Of The Review:**

8: accept, Good Paper

Even though the proposed approach is simple, inductive, and is shown to be better than considered baselines, the gains seem to be marginal especially without error bars. Even in the inductive setting (i.e. section 5.4), the proposed approach doesn’t provide a significant boost compared to the baselines especially compared to CompEx-N3* baseline. For other reasons for my score please refer to the Cons section above.

But, I think the proposed approach is interesting because of its simplicity and the problem it is trying to solve and I am on the fence leaning towards a weak acceptance. But, I am willing to raise my score if the authors can provide error bars that will validate that the results are statistically significant. I also think that the authors should add a discussion about how to scale the method when the graph changes especially in the recommendation setting. Is there a need to resample the subgraphs which means that this computation cost is no longer one-time and should be reported as a part of at least the training time imho (i.e in fig. A5)? If not, then this should strengthen the contributions of the paper and discussed (in Appendix perhaps if there is no space).

Another interesting analysis to have is to see the performance of GraphANGEL based on the count of relations that are added for generalization study in section 5.4. Maybe, there is an interesting pattern there.  Even going a step further, instead of splitting the relations randomly split them based on the frequency of occurrence of the relation. I think that these additions will make this paper more impactful.

Edit: Updated after reading response from the authors.

---

> ### Author Response · Authors · 2021-11-23
> **Response to Reviewer jqL2**
>
> Thanks for your feedback and suggested experiments. Please also see the main comment above.
>
> > While the approach is general, but the experiments are conducted by using patterns of up to 4-cycle which can be a problem if the target nodes s and t are quite far in the graph and/or the graph is disconnected.
>
> We computed the fraction of 3-hop and 4-hop neighborhoods over all the nodes for every node in each dataset, and averaged them. Results reported in Table A3 show that generally speaking, it is very unlikely for two nodes to be more than 3-hops and 4-hops away from each other. Please refer to Appendix 4.1 for a detailed discussion.
>
> > Because there is no explicit embedding for relation, it will not be possible to incorporate unseen relations as a part of the graph after actually observing them during inference. While having an explicit embedding/matrix for relation would actually be useful in that setting.
>
> Thanks for your insightful suggestion. As mentioned in Section 2, one assumption of GraphANGEL is that existing relations are enough to cover unseen relations by logical inference. In other words, new relations will not influence much as long as this intuition holds. Even so, we agree that considering the impact between unseen relations themselves is interesting, and it could be a future direction.  Please refer to Appendix 4.1 for detailed discussions.
>
> > The results are marginally better and the error bars (from multiple runs) are not shown.
>
> We have modified Table 4 with the error bars from 10 runs. Results with the error bars further verify the superiority of GraphANGEL in the standard setting.
>
> > The approach precomputes the subgraphs which would be a single-time operation in a static graph. But this may not be the case for heterogeneous recommendation settings, where I would expect that the graph to evolve very fast.
>
> For dynamic graphs, we provide an incremental updating algorithm (named Algorithm 11) for the buffer $\mathcal{B}$ in the context of Algorithm 8. We also provide the corresponding time complexity analysis in Appendix 3.2. The latter shows that the proposed algorithm is much more efficient when the number of new nodes and edges are sufficiently smaller than the number of old nodes and edges, which is a reasonable assumption in the context of real-world recommender systems.
>
> > Even going a step further, instead of splitting the relations randomly split them based on the frequency of occurrence of the relation.
>
> We show the experimental results for GraphANGEL and baselines on the 20\% least frequent relations in Table 5, where each model is trained with all the relations but tested only with  20\% least frequent relations. Results show that our method can significantly outperform the existing methods in terms of MRR, Hit@1, Hit@3, Hit@10.

---

> > ### Comment · Reviewer_jqL2 · 2021-11-25
> > **Score updated.**
> >
> > I am satisfied with the response from the authors and hence I am raising my score to *8: accept, Good Paper* . I really appreciate the authors for providing a version suitable for evolving graphs.

---

### Official Review · Reviewer_1Ki2 · 2021-11-04

**Correctness:** 4
**Technical Novelty And Significance:** 4
**Empirical Novelty And Significance:** 4
**Recommendation:** 8
**Confidence:** 5

**Main Review:**

**Strengths**
1. The core idea is deep rooted in real world applications and combines the expressive power of GNNs and raw graph pattern matching.
2. The method is independent of the relation type and thus can be applied for *unseen* relations which makes the method more generalizable.
3. Authors do extensice experimentation to prove the effectiveness of their method on heterogeneous graph recommendation datasets and knowledge graph relation prediction. Moreover to support the claims of generalization to unseen relation types, author conduct a separate experimentation which shows decent improvements over baseline methods.
4. Paper is well written and authors do a good job of explaining their method to the reader.

**Weaknesses/Questions**
1. Since the method relies on retrieval of subgraphs, the time complexity can take a hit. However, to resolve this issue authors limit the subgraph computation to pairs, 3-cycles and 4 cycles and precompute all such subgraphs and store on disk. This can result in high storage requirements.
2. While the gains in inductive setting are decent, the gains for knowledge graph completion and heterogeneous graph recommendation tasks are marginal.

Q. How are the ties resolved while ranking for knowledge graph completion task? Please consider mentioning this in the main text.
For more reference please take a look at Section 4 of [this](https://drive.google.com/file/d/1WhTapGnWM8u50E8vCcx_acGBEngUBu-i/view) paper.

**Summary Of The Paper:**

This paper introduces a new framework for relation prediction by using "analogy" subgraphs. Authors extract some analogy subgraphs ( similar in shape ) containing the pair of source (*s*) and target (*t*) nodes, and compare them against other subgraphs sharing similar shapes. The method works by first storing base patterns (limited to pairs, 3-cycles and 4-cycles to manage complexity) and building three sets of subgraphs on top of each of these base patterns, *target*, *supporting* and *refuting* patterns for a given triplet *<s, r, t>*. Authors then sample *K* target subgraphs, and *Q* subgraphs each from *supporting* and *refuting*. This sampling is done for all *P* base patterns. These sets of subgraphs are then converted into a single vector representation using GNNs. The final prediction is derived by calculating the distance between embeddings for *target* and *supporting* (denoted as *s+*) and, *target* and *refuting* (denoted as *s-*) and feeding these numbers to a neural network, treating the final relation prediction problem as a binary classification problem.

**Summary Of The Review:**

Authors propose a new relation prediction framework *GraphANGEL* which is generalizable and combines GNNs and graph pattern matching. The experimentation done in the paper is extensive and shows the effectiveness of the method.
Overall I feel the authors make a significant contribution to the relation prediction literature and I would like to recommend **accepting** this work.

---

> ### Author Response · Authors · 2021-11-23
> **Response to Reviewer 1Ki2**
>
> Thanks for your feedback. Please also see the main comment above.
>
> > To resolve the time complexity issue authors limit the subgraph computation to pairs, 3-cycles and 4 cycles and precompute all such subgraphs and store on disk. This can result in high storage requirements.
>
> We summarize our approaches to deal with high time complexity of subgraph computation and high storage requirement as follows:
>
> 1. We only pre-compute all the supporting cases for 3-cycles and uniformly sample $N_\mathtt{3-cycle}$ supporting cases for 4-cycles, $2\cdot N_\mathtt{3-cycle}$ refuting cases for 3-cycles and 4-cycles and store them on disk, where $N_\mathtt{3-cycle}$ denote the number of 3-cycles. Please see Section 3 and Appendix 3.1 for detailed descriptions. The statistics of 3-cycles and 4-cycles are reported in Table A3, and the size of the buffer $\mathcal{B}$ in Appendix 5.3.
> 2. We implement GraphANGEL following Algorithm 8, where we only need to compute subgraphs once (please see Appendix 3.1 for detailed analysis). We further design Algorithm 11, the incremental updating algorithm in context of Algorithm 8 to handle the dynamic graphs. Please refer to Appendix 3.2 for its time complexity analysis.
>
> > While the gains in inductive setting are decent, the gains for knowledge graph completion and heterogeneous graph recommendation tasks are marginal.
>
> As emphasized in the main comment, the main advantage of our model is that it can generalize to relation types unseen in during training *without fine-tuning*.
>
> > How are the ties resolved while ranking for knowledge graph completion task? Please consider mentioning this in the main text. For more reference please take a look at Section 4 of this paper [1].
>
> We randomly break ties for triplets with the same score (denoted as RANDOM in [1]). We have clarified this in Section 5.1.
>
> [1] A Re-evaluation of Knowledge Graph Completion Methods. Sun et al., ACL 2020.

---

### Official Review · Reviewer_3Zrj · 2021-11-05

**Correctness:** 3
**Technical Novelty And Significance:** 3
**Empirical Novelty And Significance:** 3
**Recommendation:** 8
**Confidence:** 3

**Main Review:**

Pros:

- A solid paper, well-motivated, and well-presented. The authors provide detailed examples to demonstrate the main idea of the proposed method, making the paper easy to follow.

- The method combines the logic/subgraph patterns and graph representation learning together to build an explainable reasoning method for relation prediction, such that it can be generalized to unseen relations.

- Experiments on graph recommendation and link prediction demonstrated the effectiveness of the method. It can further produce explainable heat maps of attention scores to help users understand the reasoning process of relation prediction.

Cons:

- The main weakness of the method, I think, is its poor robustness, as it requires each triple to have supporting patterns. The authors also pointed out this in the paper. I think the constraints of logical patterns are too strong. Maybe, loosening some constraints, such as not requiring the subgraphs to have the same shapes, would be helpful? I do not know—just a free discussion.

- In my view, the proposed method would cost more time than others, as it has an additional process of subgraph retrieval.

Detailed comments:

- It seems that the authors only provide the complexity analysis of retrieval and sampling. What about the overall time complexity of the proposed method? In my view, the method could cost more time than other baselines, such as those purely embedding-based methods for link prediction.

- What is the standard for choosing baselines of link prediction? It seems that some recent strong methods are not included.

**Summary Of The Paper:**

The paper proposes a novel inductive method for relation prediction over heterogeneous graphs. The method extracts supporting and refuting subgraphs to judge whether a relation exists between two nodes. In this way, the relation embedding can be implicitly expressed by the other relations in the subgraphs. The method is generalized to unseen relations. The method uses GNNs to encode subgraphs for similarity computation. Experiments on graph recommendation and link prediction demonstrated the effectiveness of the method.

**Summary Of The Review:**

A solid, well-motivated, and well-presented work. Although the method would suffer from poor robustness, it is a meaningful attempt to connect logic/subgraph patterns and graph representation learning for explainable reasoning.

--- after rebuttal ---

The authors' response addressed my concerns. I would raise the score to 8.

---

> ### Author Response · Authors · 2021-11-23
> **Response to Reviewer 3Zrj**
>
> Thanks for your feedback. Please also see the main comment above.
>
> > I think, is its poor robustness, as it requires each triple to have supporting patterns.
>
> We devised new experiments to study its robustness on noisy graphs and provide the statistics of supporting patterns:
> 1. We empirically investigate the robustness of GraphANGEL in the context of noisy graphs in revised Appendix 6.5, where the source or target node of each triplet is modified with probability 5\%. Results reported in Tables A12 and A13 show that our method consistently outperforms the strong baselines in both heterogeneous graph recommendation and knowledge graph completion tasks.
> 2. We provide the statistics of patterns in the datasets (e.g., the number of 3-cycles and 4-cycles in each dataset) in Table A3. These results indicate that GraphANGEL is very unlikely to fail to find the supporting cases of the pre-defined patterns, as (i) there is no relation requirement for supporting patterns in knowledge graphs (as shown in Table 2) so any 3-cycle will work, and (ii) Table A1 shows that there are few types of nodes and edges in four heterogeneous graphs.
>
> > Maybe, loosening some constraints, such as not requiring the subgraphs to have the same shapes, would be helpful?
>
> Actually, GraphANGEL is more flexible than it might appear at first glance.  In this regard, we emphasize that GraphANGEL is a general framework that can work on any patterns; the patterns defined in Table 2 are just implementation examples. Moreover, we choose these patterns mainly for the following two reasons:
> 1. As mentioned in Section 2, each pattern represents a specific logical rule, as illustrated in Figure 1. In this regard, 3-cycle and 4-cycle shaped patterns might be similar in terms of shape, but different in terms of logics/semantics.
> 2.  These patterns (i.e., Pairs, 3-cycles, 4-cycles) are simple and efficient for subgraph search and retrieval as shown in Appendix 2, and already can achieve good results.
>
> > The proposed method would cost more time than others, as it has an additional process of subgraph retrieval.
>
> We agree that directly searching and retrieving subgraphs is time-inefficient. Indeed, this is the motivation for the following proposed approaches:
> 1. We provide a series of subgraph retrieval and uniform sampling algorithms in Appendix 2 (i.e., Algorithms 2,3,4 for searching and retrieving the supporting cases of 3-cycles and 4-cycles and Algorithms 5,6,7 for uniformly sampling the supporting cases of 4-cycles and refuting cases of 3-cycles and 4-cycles).
> 2. We propose to pre-compute and store the retrieved or sampled subgraphs in Algorithm 8, which only needs to be computed once. We also provide the corresponding analysis in Appendix 3.1.
> 3. We further provide Algorithm 11, the incremental updating algorithm (along with its time complexity analysis) in Appendix 3.2 to handle the dynamic graphs in the context of Algorithm 8.
>
> We also empirically studied the training and inference times of GraphANGEL against other baselines in Appendix 6.7. Results depicted in Figure A7 show that the running and inference time of our method is at an acceptable level comparing to the existing graph neural networks.
>
> > What about the overall time complexity of the proposed method? In my view, the method could cost more time than other baselines, such as those purely embedding-based methods for link prediction.
>
> We provide the overall time complexity analysis in Appendix 3.1. It is true that our method costs more time than those knowledge embedding-based methods such as TransE [1] and RotatE [2], as shown in Figure A7. However, we do note that the higher cost compared to traditional embedding-based methods also exist in other GNN-based methods such as R-GCN [3], as shown in Figure A7.
>
> > What is the standard for choosing baselines of link prediction? It seems that some recent strong methods are not included.
>
> The standard for choosing baselines is that the baseline methods are public and well-recognized. So, we follow the recent public work: RNNLogic [4], and include their baselines and their proposed method as the baseline methods in our paper. We are also aware of recent work [5]. However, as Reviewer y1cR pointed out, the publication time for this work is within 3 months, and we are unable to find the source code and detailed hyper-parameter settings such as the dimensions of embedding in the paper. So, we do not include this work as a baseline.
>
> [1] Translating Embeddings for Modeling Multi-relational Data. Bordes, et al., NIPS 2013.
>
> [2] RotatE: Knowledge Graph Embedding by Relational Rotation in Complex Space. Sun, et al., ICLR 2019.
>
> [3] Modeling Relational Data with Graph Convolutional Networks. Schlichtkrull, et al., ESWC 2018.
>
> [4] RNNLogic: Learning Logic Rules for Reasoning on Knowledge Graphs. Qu, et al., ICLR 2021.
>
> [5] Neural Bellman-Ford Networks: A General Graph Neural Network Framework for Link Prediction. Zhu, et al., Neurips 2021.

---

> > ### Comment · Reviewer_3Zrj · 2021-11-26
> > **Score raised**
> >
> > Thank you for the response to my concerns. I think this is very solid work. After reading all the comments and discussions, I would like to raise the score to 8.

---

### Author Response · Authors · 2021-11-23
**Repsonse to all reviewers**

We first summarize our response and the results of additional suggested experiments here. We have also responded to the specific concerns of each reviewer as individual comments below.

All reviewers agree that our idea is interesting and our contributions are valuable to the community. To summarize our contributions: (i) we propose a novel relation prediction framework named GraphANGEL that can generalize to relation types unseen during training without fine-tuning, and (ii) our experimental results verify that GraphANGEL can consistently outperform strong baselines and exhibit better generalizability to unseen relations (see Figure 4).

Here is a list of new experiments:
1. Comparison with [1] (named QuatE for the standard task in Tables 4, 5, A4, A13, QuatE$^*$ for the generalizability study in Tables 6, A5, A6, A7, A8, A9).
2. Ablation study on different graph embedding techniques. We evaluated GraphANGEL with different GNNs (with different number of layers) for representation computation (i.e., $\Phi$ in Eq. (1)).
3. Robustness study on noisy graphs. We perturbed the noisy graphs by modifying the source or target nodes with probability 5\% in the training dataset, and compared GraphANGEL to other baselines on the graphs. Please see the revised Appendix 6.5 for details.  This is primarily addressing Reviewer fant's question, but we believe this should be addressing other reviewers' concerns on robustness.

These new experiments show that GraphANGEL consistently outperforms the existing baselines (including QuatE and QuatE$^*$) on both unseen relations or noisy settings in terms of MRR, Hit@1, Hit@3, Hit@10.

Here is a list of new algorithms and discussions for the limitations of GraphANGEL:
1. An incremental updating algorithm for the buffer $\mathcal{B}$ for dynamic graphs in Algorithm 11 with corresponding analysis in revised Appendix 3.2. While this is primarily addressing Reviewer jqL2's concern on constantly evolving graphs, we believe that this could address the scalability concerns from other reviewers as well.
2. Further discussions about two limitations are mentioned in Section 3 in revised Appendix 4.1, which is supported by the statistics of the datasets reported in Table A3.

For your convenience, all modifications in the updated PDF are highlighted in blue color.

[1] Quaternion Knowledge Graph Embeddings. Zhang et al., NeurIPS 2019.

---

### Decision · Program_Chairs · 2022-01-20

**Decision:**

Accept (Poster)

**Comment:**

The paper presents an approach to predict relations between node pairs in heterogeneous graphs, with application to recommendation and knowledge-base completion.

The author's approach is to compute similarities between subgraphs that are neighborhoods of nodes where the relation holds or not to score a relation. The authors use graph neural networks to scores these subgraphs. The type of subgraphs that are considered are pairs of nodes, 3- and 4- cyles to make inference and training tractable. The paper lies in the stream of work that combines logical reasoning and neural network, even though in that particular instance it mostly combines graph mining techniques and neural networks.

The reviewers unanimously liked the presentation of the paper and the high empirical performance. The rebuttal addressed most of the remaining concerns.